# NMR and MS reveal characteristic metabolome atlas and optimize esophageal squamous cell carcinoma early detection

Yan Zhao [1,2,6], Changchun Ma[3,6], Rongzhi Cai[1], Lijing Xin [4], Yongsheng Li [5], Lixin Ke[1], Wei Ye[1], Ting Ouyang[1], Jiahao Liang[1], Renhua Wu [1,7] ✉ & Yan Lin [1,7] ✉

Metabolic changes precede malignant histology. However, it remains unclear whether detectable characteristic metabolome exists in esophageal squamous cell carcinoma (ESCC) tissues and biofluids for early diagnosis. Here, we conduct NMR- and MS-based metabolomics on 1,153 matched ESCC tissues, normal mucosae, pre- and one-week post-operative sera and urines from 560 participants across three hospitals, with machine learning and WGCNA. Aberrations in 'alanine, aspartate and glutamate metabolism' proved to be prevalent throughout the ESCC evolution, consistently identified by NMR and MS, and reflected in 16 serum and 10 urine metabolic signatures in both discovery and validation sets. NMR-based simplified panels of any five serum or urine metabolites outperform clinical serological tumor markers (AUC = 0.984 and 0.930, respectively), and are effective in distinguishing early-stage ESCC in test set (serum accuracy = 0.994, urine accuracy = 0.879). Collectively, NMR-based biofluid screening can reveal characteristic metabolic events of ESCC and be feasible for early detection (ChiCTR2300073613).

Esophageal cancer (EC) is a significant public health concern worldwide[1]. China accounts for over half of the global annual incidence and mortality, of which more than 90% are esophageal squamous cell carcinoma (ESCC), and the T0, Tis, and T1 stages account for 10.8% of the total cases[2,3]. The cure rate for early-stage ESCC exceeds 90%. However, due to the subtle nature of symptoms and the lack of biomarkers for early diagnosis, most patients are typically diagnosed in late-stage T3-T4, resulting in a 5-year survival rate of only approximately 21%[4]. Currently, the gold standard for diagnosing ESCC primarily relies on endoscopy coupled with histopathology, but its invasive nature reduces patient compliance[5]. Barium swallow and CT imaging techniques involve radiation exposure and tend to miss small lesions. Conventional serological tumor markers, such as Cytokeratin-19-fragment CYFRA21-1 (Crfr211), Squamous Cell Carcinoma Antigen (SCC), Carcinoembryonic Antigen (CEA), Carbohydrate Antigen 19-9

(CA 19-9) and Carbohydrate Antigen 15-3 (CA 15-3) have shown suboptimal accuracy in clinical practice. With the advancements in omics technologies, researchers have explored several novel biomarkers for ESCC diagnosis, including DNA methylation markers, serum miRNA, autoantibodies, somatic gene mutations, salivary exosomes, and artificial intelligence (AI)-assisted sponge cytology[6–11]. However, these approaches face limitations due to the requirement of advanced technology platforms, methodological instability, or high costs, hindering their translation into large-scale clinical applications. Consequently, there is an urgent need to develop reliable, non-invasive, accessible and affordable tools for ESCC early detection[12].

It takes years for ESCC to progress from squamous cell hyperplasia to atypical hyperplasia, carcinoma in situ, early-stage and invasive cancer[13]. In addition, metabolic phenotypic changes could precede malignant histological alterations, providing a significant

[1]Radiology Department, Second Affiliated Hospital of Shantou University Medical College, Shantou, Guangdong, China. [2]Central Laboratory, Clinical Research Center, Shantou Central Hospital, Shantou, Guangdong, China. [3]Radiation Oncology Department, Cancer Hospital of Shantou University Medical College, Shantou, Guangdong, China. [4]Animal Imaging and Technology Core, Center for Biomedical Imaging, Ecole Polytechnique Fédérale de Lausanne, Lausanne, Switzerland. [5]Department of Medical Oncology, Chongqing University Cancer Hospital, Chongqing, China. [6]These authors contributed equally: Yan Zhao, Changchun Ma. [7]These authors jointly supervised this work: Renhua Wu, Yan Lin. ✉e-mail: cjr.wurenhua@vip.163.com; 994809889@qq.com

opportunity for early detection and timely intervention[14]. However, it remains to be established whether there are characteristic metabolic changes during ESCC evolution and whether such metabolic changes can be detected. Metabolomics holds great promise for identifying disease-associated metabolites, highlighting its valuable insights into early diagnosis, treatment strategies, and mechanistic investigations[15]. Proton nuclear magnetic resonance ($^1$H-NMR) and mass spectrometry (MS) are the most mainstream technological platforms in metabolomics[16,17]. MS exhibits high sensitivity (SE) but requires expensive standard reagents. $^1$H-NMR has shown remarkable stability, excellent reproducibility, quantitative nature, non-invasive sample analysis, and has been well-suited for clinical multi-center, large-scale, and longitudinal monitoring studies for establishing successful clinical applications in countries such as the United Kingdom and Canada[18,19].

Most existing metabolomics studies of ESCC have mainly relied on analyzing biofluid samples, such as serum and urine. Our previous study showed that NMR-based biofluid metabolomic profiles can discriminate ESCC patients from healthy controls (HCs), suggesting the potential utility of biofluid metabolic fingerprinting as predictors for ESCC[20–22]. However, potential confounders, such as environmental factors, lifestyle habits, phenotypic variations and comorbidities, might influence biological fluid metabolism, leading to a gap between ESCC biofluids and characteristic molecular events of ESCC tissues. In this study, we employed a comprehensive research strategy incorporating 1,153 multi-dimensional matched specimens, NMR and targeted MS cross-platform testing, as well as multi-center validation. We aimed to investigate tumor tissue-specific metabolic biomarkers during ESCC evolution, and then leverage them as references to develop and optimize biofluid metabolic classifiers based on NMR (a means that can better achieve 'health equity'), which not only faithfully reflect the characteristics of tissue metabolic changes with high accuracy of tissue origin, but also has sufficient clinical SE for ESCC early diagnosis and screening.

## Results

### Clinical phenotypes of enrolled subjects

In this multi-center retrospective analysis, we included 560 participants with a total sample size of 1153 from three centers in southern China to explore metabolic alterations associated with ESCC. Figure 1 illustrated the overall study schema, and Supplementary Fig. 1 provided the sample distribution. As shown from the demographics and baseline characteristics summarized in Supplementary Table 1, clinical tumor markers, such as CEA, CA19-9, CA15-3, SCC and CYFRA21-1, did not present sufficient specificity (SP) and SE for diagnosing ESCC. Supplementary Table 2 presented the clinical phenotypes of the validation set. As seen from this table, early-stage ESCC patients had no apparent symptoms, and the incidence of thoracotomy and post-operative Intensive Care Unit (ICU) transfer rates were low. Additionally, six patients with ESCC were initially misdiagnosed with intraepithelial neoplasia due to sampling errors caused by intratumoral heterogeneity.

### Early-stage tissue metabolomic landscape of ESCC patients by NMR and MS

Given the potential influence of various confounders on biofluid biomarkers, it is crucial to identify cancer-specific biomarkers closely associated with ESCC evolution. In the discovery set, our three previous studies consistently identified metabolic biomarker candidates that constituted our initial potential biomarker pool in ESCC tissues using 400 MHz $^1$H-NMR, including Acetate, Alanine, beta-Glucose, etc., represented with directional arrows indicating their increase or decrease compared to normal controls (Supplementary Data 1, left panel)[20–22].

In the validation set, 600 MHz $^1$H-NMR was performed on matched ESCC patients and HC samples. Representative $^1$H-NMR spectra and metabolites resonance assignments were presented in Supplementary Fig. 2A–C and Supplementary Data 2, respectively. Uniform manifold approximation and projection (UMAP) and hierarchical clustering analysis (HCA) analysis divided tissue samples into two groups (Supplementary Fig. 3A, B). Orthogonal partial least squares discriminant analysis (OPLS-DA) revealed distinct metabolic differences between ESCC tumors vs. normal mucosa (Fig. 2A), early-stage ESCC tumors vs. normal mucosa (Fig. 2B), and early-stage ESCC vs. advanced-stage ESCC (Fig. 2C). External permutation tests validated that the models were suitable for data analysis (Fig. 2D–F). According to the screening criteria (Variable importance in the projection

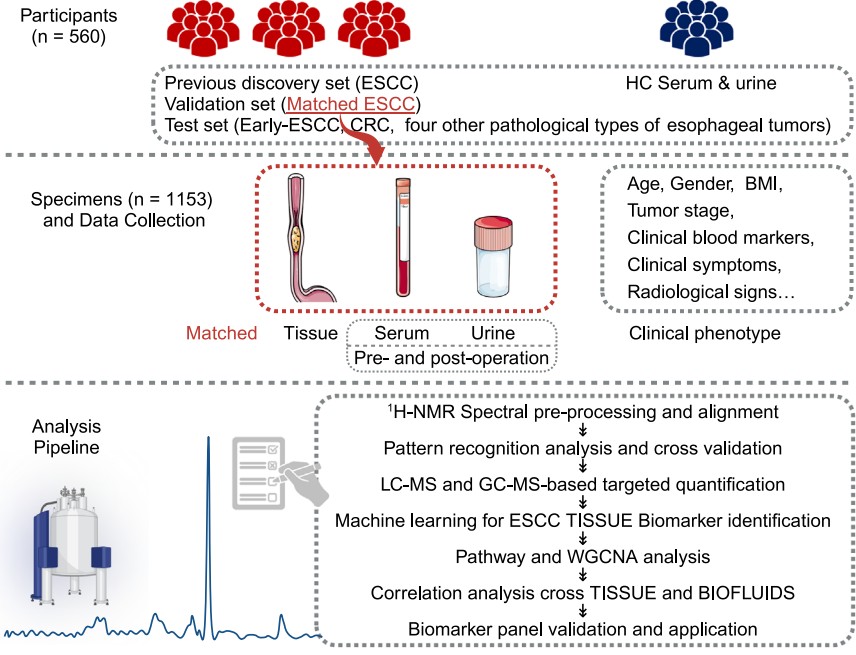

**Fig. 1 | Schema of overall study design.** A total of 560 participants from three centers were involved in this study. Tissue, serum and urine specimens were collected and subjected to $^1$H-NMR- and MS-based metabolomics analysis, followed by pattern recognition, machine learning and WGCNA analysis.

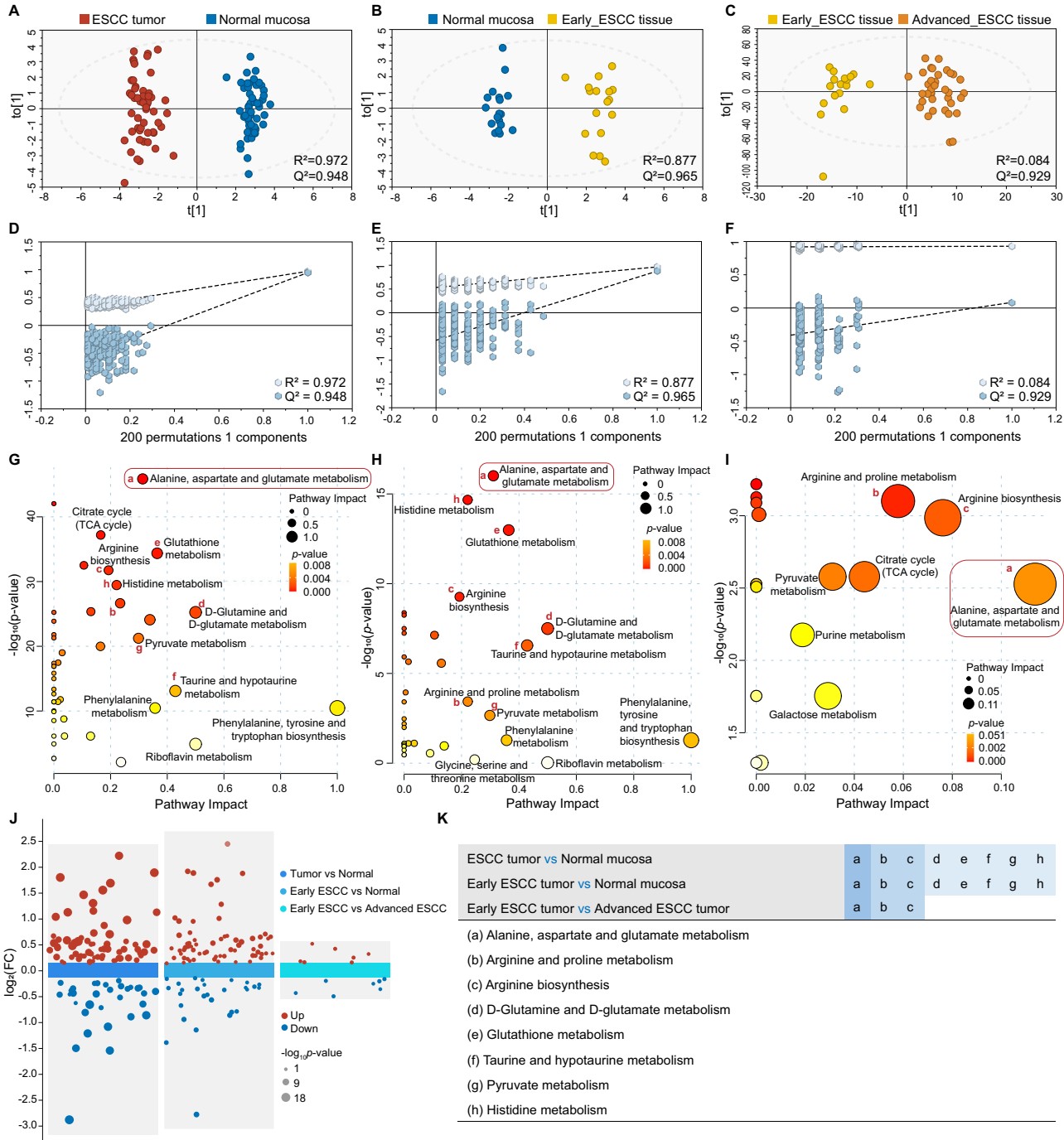

**Fig. 2 | Tissue metabolomic landscape of ESCC patients based on NMR-based metabolomics. A–C** OPLS-DA score plot based on $^1$H-NMR tissue spectra from ESCC patients at different stages. Red: ESCC tumor; Blue: Normal mucosa; Yellow: Early-stage ESCC tissue; Orange: Advanced-stage ESCC tissue. **D–F** Statistical validation of the corresponding model by permutation analysis (200 times). The x-axis represents the permutation retention rate of the permutation test, and the dots in the upper right corner represent the $R^2$ (light blue) and $Q^2$ (dark blue) values of the original model when the permutation retention rate is 1. $R^2$ measures the goodness of fit, while $Q^2$ measures the predictive ability of the model. Light blue dots represent the $R^2$ values obtained from the permutation test, while dark blue dots represent the $Q^2$ values obtained from the permutation test. The two dashed lines represent the regression lines of $R^2$ and $Q^2$, respectively. **G–I** Metabolic pathway analysis. Relative betweenness centrality was the selected node importance measure for pathway topological analysis. All pathways are represented as bubbles. The color and size of each bubble correspond to its *p*-value and pathway impact value,

respectively. In general, bubbles on the right side of the map have higher weights, while bubbles at the top have smaller *p*-values. The precise *p*-values for metabolic pathway analysis are provided in the Source Data without adjustments. **J** Multiple Volcano plot based on the same batch of samples, showing the comparison of differential metabolites between different groups (Tumor vs Normal; Early ESCC vs Normal; Early ESCC vs Advanced ESCC). *P*-values were determined by two-sided t-test without adjustments. Metabolites with *p* < 0.05 were visualized as solid circles on the plot, while those with *p* > 0.05 were not displayed. A $\log_{10}$ transformation was applied to the *p*-values of each significantly differential metabolite to visualize their significance levels. **K** Statistical analysis of principal metabolic pathway disturbances in the evolution of ESCC. A pathway impact greater than 0.1 and *p* < 0.05 was used as the cut-off value for the statistical significance. (**A, D, G**, left panel) ESCC vs. normal mucosa patients; (**B, E, H**, middle panel) early-ESCC vs. normal mucosa patients; and (**C, F, I**, right panel) early-ESCC vs. advanced -ESCC patients. Source data are provided as a Source Data file.

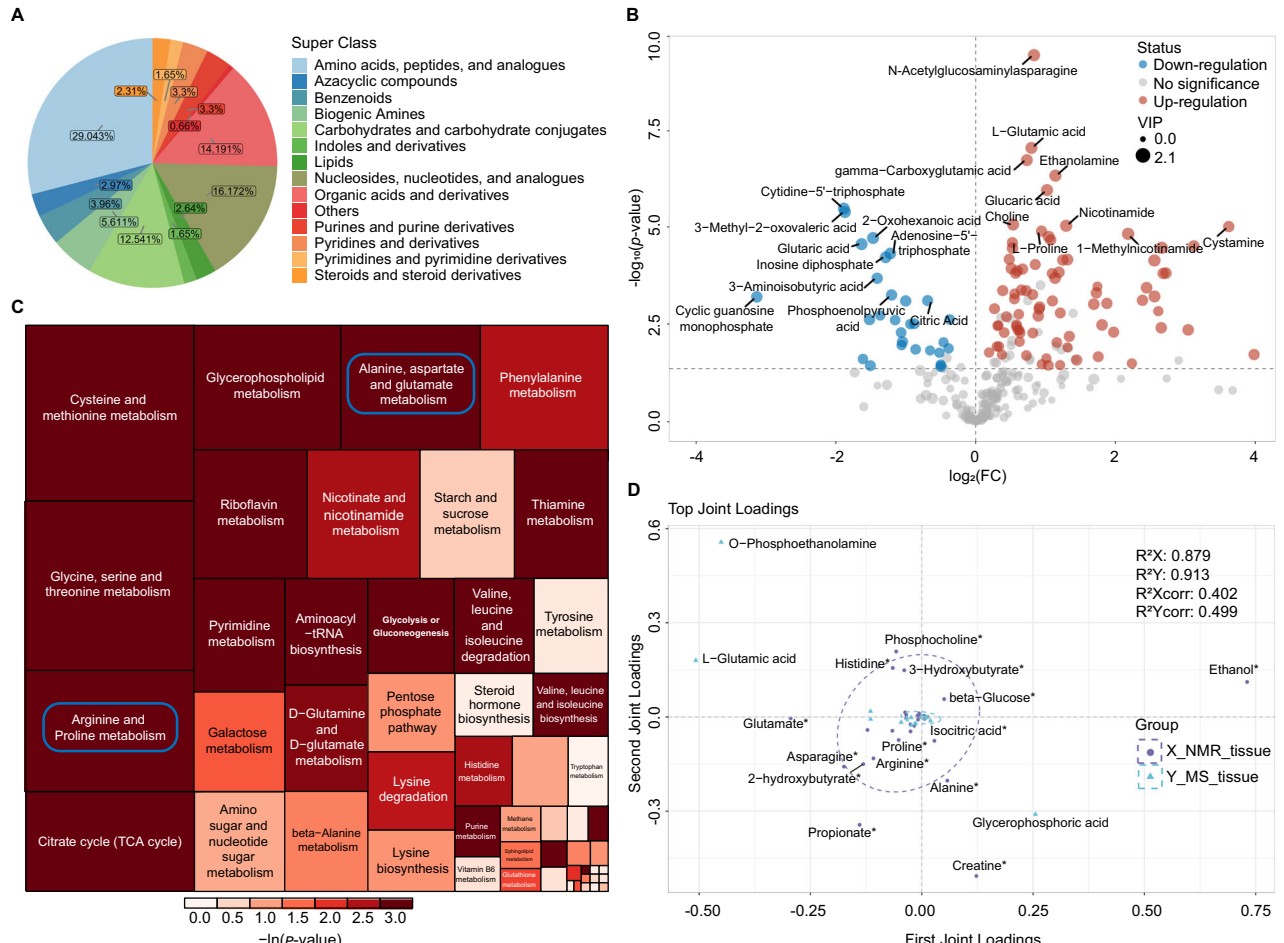

**Fig. 3 | Mass spectrometry-based targeted quantitative analysis of early stage ESCC tissue samples confirm the NMR results. A** Pie chart of quantified metabolite categories. Colors represent different compound super-classes; coloration follows the legend counterclockwise per each pie chart. **B** Volcano plot analysis performed on early-ESCC tissue samples vs. normal controls. *P*-values were determined by two-sided *t*-test without adjustments. Differentially-expressed metabolites are indicated by blue dots (down-regulation relative to normal controls) and red dots (up-regulation relative to normal controls), respectively. Gray dots indicate no significant difference. **C** Treemap of the most enriched KEGG pathways in early-ESCC tissue samples. Relative betweenness centrality was the selected node importance measure for pathway topological analysis. Each square represents a metabolic pathway; the square size represents the impact factor in the topological analysis; the color of the squares indicates the *p*-value of the enrichment analysis; and the darker the color, the more significant the enrichment. **D** O2PLS loading plots of the key differential metabolites analyzed by NMR-based and MS-based metabolomics. The top 30 metabolites in early-ESCC tissues detected by NMR and MS are labeled in purple and blue, respectively. Source data are provided as a Source Data file.

(VIP) > 1, adjusted *p* < 0.05 and significant p|corr| values), there were 43, 34, 12 and 45 potential metabolic biomarkers identified for ESCC tissues vs. normal mucosa, early-ESCC tissues vs. normal mucosa, early-ESCC tissue vs. advanced ESCC tissue, and advanced ESCC tissue vs. Normal mucosa, respectively, as shown in Supplementary Data 3, Supplementary Fig. 3C and multiple volcano plots (Fig. 2J). Given our study's emphasis on early detection, we primarily focused on metabolites already showing diagnostic efficacy in the early stage, and further screening included selecting those with high fold change (FC) and areas under the curve (AUC) values, recorded in Supplementary Data 1, middle panel. Kyoto Encyclopedia of Genes and Genomes (KEGG) pathway analysis revealed perturbations of 'Alanine, aspartate, and glutamate metabolism' and arginine-related metabolic pathways during ESCC progression, from normal mucosa to early and to advanced stages of tissues (Fig. 2G–I). Additionally, 'ᴅ-glutamine and ᴅ-glutamate metabolism', 'Glutathione metabolism', Taurine-, pyruvate-, and histidine-related metabolism' were found to be significantly disrupted during ESCC tumorigenesis (Fig. 2K).

Targeted quantitative MS were used to validate the NMR results of early-ESCC tissues on the same batch. Over 500 biochemicals were quantified by multiple reaction monitoring (MRM) mode, including

544 metabolites across 13 classes detected via Liquid chromatography-tandem mass spectrometry (LC/MS–MS) and 11 fatty acids (FAs) detected via gas chromatography-tandem mass spectrometer (GC/MS–MS), covering most of the substances we identified in NMR. Quality control (QC) and HCA analysis were carried out (Supplementary Fig. 3D, E). We identified 315 differential metabolites from various categories (Fig. 3A, Supplementary Data 3) and visualized them with volcano plots (Fig. 3B). Consistent expression trends and significant pathways were observed for the same differential metabolites detected across NMR and targeted MS platforms (Fig. 3C; Supplementary Data 1, middle panel). Random forest (RF) algorithm showed that Glutamate, a key molecule in the 'Alanine, aspartate and glutamate metabolism' pathway, was the most significant metabolite to distinguish early-stage ESCC from normal tissues (Supplementary Fig. 3F).

To pinpoint the unique metabolic biomarkers in early-ESCC tissues, we constructed a two-way orthogonal partial least squares (O2PLS) model by integrating NMR and MS data matrices. We then identified the top 30 metabolite variables with high correlation and weight in both datasets using the loading plot (Fig. 3D) and listed them in Supplementary Data 1, right panel. Combining the overall changes in key metabolic pathways in the tissue and considering them

**Table 1 | Potential biomarkers in early-stage ESCC tissues identified by integrated [1]H-NMR and MS-based metabolomics analysis**

| Potential biomarkers in early-ESCC tissue | | | |
|---|---|---|---|
| 2-Hydroxybutyrate | ↑ | Phosphocholine | ↑ |
| 3-Hydroxybutyrate | ↑ | Proline | ↑ |
| Acetate | ↑ | Propionate | ↑ |
| Arginine | ↑ | Threonine | ↑ |
| Asparagine | ↑ | Tyrosine | ↑ |
| Choline | ↑ | Uridine | ↑ |
| Flavin mononucleotide | ↑ | Valine | ↑ |
| Glutamate | ↑ | 3-Hydroxypropionic acid | ↓ |
| Glutathione | ↑ | Alanine | ↓ |
| Glycine | ↑ | beta-Glucose | ↓ |
| Histidine | ↑ | Creatine | ↓ |
| Isoleucine | ↑ | Glutamine | ↓ |
| Leucine | ↑ | Isocitric acid | ↓ |
| Lysine | ↑ | Sarcosine | ↓ |
| Ornithine | ↑ | Taurine | ↓ |
| Phenylalanine | ↑ | Trimethylamine N-oxide | ↓ |

↑: upregulation; ↓: downregulation.

comprehensively, we identified the cancer-specific metabolic biomarkers in early-ESCC tissues, including 2-hydroxybutyrate, 3-Hydroxybutyrate, 3-Hydroxypropionic acid, etc., shown in Table 1.

## Metabolic changes in biofluids reflect specific alterations in ESCC tissues

In the aforementioned validation set, we collected pre- and one-week post-operative serum and urine samples from ESCC patients and analyzed them by 600 MHz NMR spectroscopy. Pairwise comparison using OPLS-DA models showed significant differences in serum (Fig. 4A–C) and urine (Fig. 4G–I) profiles between pre-operation, post-operation, and HC groups. Following the same screening criteria for tissue analysis, we identified serum and urine metabolites that differed between groups (Supplementary Data 3). KEGG metabolic pathway enrichment analysis was further performed (Fig. 4D–F, J–L), and we found that 'Alanine, aspartate and glutamate metabolism' consistently exhibited the most significant changes from tumor-bearing to tumor-resected phase and to healthy status in biofluids (Supplementary Fig. 4A, B).

Using the Mantel test to assess the correlation between potential biomarkers in key metabolic pathways (such as 'Alanine, aspartate and glutamate metabolism' and 'Arginine and proline metabolism') in early-stage ESCC tissues, and the differential metabolites in serum and urine, we found that the correlation between tissue metabolome and serum metabolome was greater than urine metabolome, and 'Alanine, aspartate and glutamate metabolism' in tissues was more relevant to the biofluid metabolome than the 'Arginine and proline metabolism' (Fig. 4M, N). O2PLS model was performed to analyze the associations between 'tissue-specific biomarkers' and 'biofluid potential biomarkers', and we identified the top 20 serum (Supplementary Fig. 4C) and urine (Supplementary Fig. 4D) metabolites that correlated most with tissue biomarkers. After further screening, the serum signatures (including alpha-Ketoisovalerate, Arginine, Asparagine, etc), and the urine signatures (including 3-hydroxybutyrate, Ascorbate, Dimethylamine, etc), linked to early-stage ESCC, were summarized in Table 2.

## 'Alanine, aspartate and glutamate metabolism' is crucial in ESCC progression

Weighted Gene Co-Expression Network Analysis (WGCNA) was used to explore the critical driver metabolite modules in early ESCC. To compare early-ESCC tissues and normal mucosa, we divided metabolites into four modules (Fig. 5A, Supplementary Fig. 5A, B). Two modules were significant: the Turquoise module with 172 metabolites and the Blue module with 45 metabolites (Supplementary Data 4). We found correlations between module-membership scores and the standardized difference between ESCC and HC. Notably, the Turquoise module had the highest weight, and Glutamate and Asparagine levels were most associated with eigen-metabolite levels (Fig. 5B). Functional enrichment analysis of the Turquoise module showed the role of various amino acid metabolism in driving this module (Fig. 5C). To compare the pre- and post-operative serum, the metabolic changes were closely associated with 41 metabolites contained in the Turquoise module (Supplementary Fig. 5C, D, Supplementary Data 4). The networks of these metabolites were inferred using Cytoscape. Interestingly, key molecules from the 'Alanine, aspartate, and glutamate metabolism' pathway were consistently identified as hub metabolites, aligning with the earlier KEGG metabolic pathway analysis (Fig. 5D).

To further investigate the intricate details of this characteristic molecular event, we analyzed the expression profiles of relevant metabolic enzymes and transporters using the TCGA-GTEx-ESCA dataset (Fig. 5E). The changes in the expression levels of GLS (Glutaminase), ASNS (Asparagine synthetase), SLC1A5 (Solute carrier family 1 member 5) and GPT2 (Glutamic-pyruvic transaminase 2), were consistent with the alterations of 14 metabolites (including 2-Oxo-glutarate, 4-Aminobutanoate(GABA), Citrate, Fumarate, L-Alanine, L-Arginino-succinate, L-Asparagine, L-Aspartate, L-Glutamate, L-Glutamine, N-Acetyl-L-aspartate, N-Carbamoyl-L-aspartate, Pyruvate, Succinate) involved in 'alanine, aspartate and glutamate metabolism', quantified and visualized in Fig. 5F.

## Validations of serum and urine biomarkers for ESCC early diagnosis

As an internal validation, we continued our research to verify the diagnostic and predictive capabilities of the biofluid biomarkers for ESCC. Since patients usually provide only blood or urine samples in clinical practice, we therefore analyzed 16 serum signatures (Fig. 6, left panel) and 10 urine signatures (Fig. 6, right panel) screened in Table 2, respectively. STAMP analysis showed significant differences between ESCC and HCs (Fig. 6A, B). Receiver operating characteristic curve (ROC) analysis showed that most serum and urine metabolites had good diagnostic potential, with AUC above 0.80 (Fig. 6C–F). Linear support vector machine (SVM) algorithm was further applied to improve the diagnostic ability by combining all the serum or urine signatures. The serum biomarker panel achieved an AUC of 0.999 at 70% cross-validations (CV) and 0.996 at 30% hold-out data, with a predictive accuracy of 0.996 (Fig. 6G, H). The urine biomarker panel had an AUC of 0.975 for both 70% CV and 30% hold-out data, with a predictive accuracy of 0.909 (Fig. 6I, J).

Considering the simplicity of clinical application, we used SVM algorithms to try a panel with limited metabolites. Our results showed that any five metabolites could achieve good predictive accuracy for ESCC (Fig. 6K, M). Logistic regression was then used to construct a theoretically low-performing combined biofluid panel, using the five metabolites with the lowest AUC values from the screened serum and urine signatures (Fig. 6C–F, Table 2). Here, we established a serum panel (comprising arginine, beta-aminoisobutyrate, glutamine, histidine, low-density lipoprotein, Fig. 6L) and a urine panel (comprising 3-hydroxybutyrate, dimethylamine, malonate, para-hydrxoyphenylacetate, taurine, Fig. 6N). As hypothesized, these simplified panels showed good diagnostic ability (Table 3), and the serum panel had an AUC of 0.984 (95% CI 0.968–1.000) with a SE of 0.955 and SP of 0.948, while the urine panel had an AUC of 0.930 (95% CI 0.913–0.976) with a SE of 0.850 and SP of 0.949. We also compared the diagnostic performance of biofluid panels using XGBoost, Gaussian Naive Bayes, and K-Nearest Neighbors algorithms. While the results closely approximated those

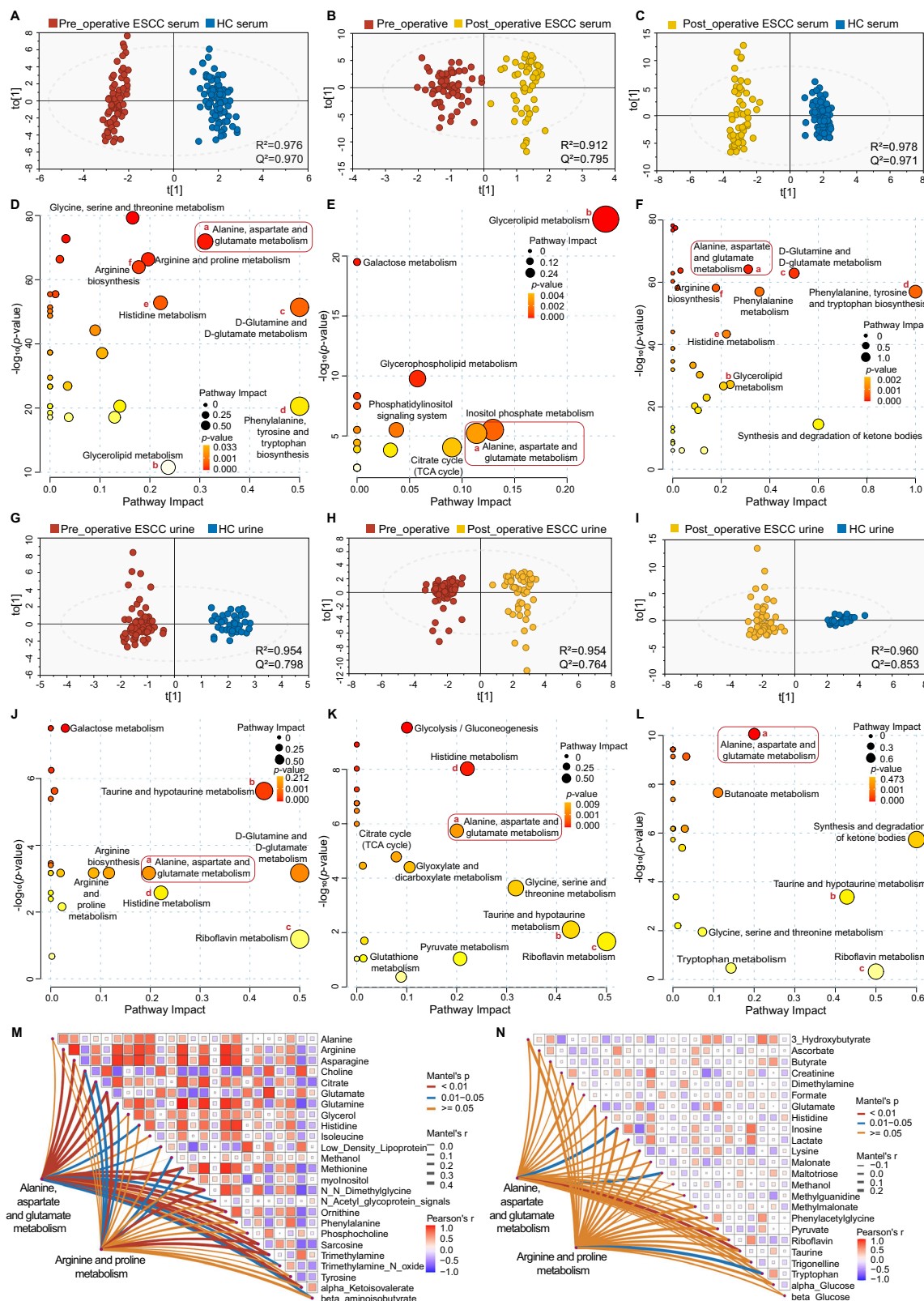

obtained through the classical logistic regression method, the efficacy of the logistic regression model remained better (Supplementary Table 3), indicating that the machine learning methods we used were robust. In other words, any five combinations of the 16 serum or 10 urine metabolites were effective in distinguishing between ESCC and HCs, and all were superior to the conventional clinical markers and even their combined models (Fig. 7A), although the diagnostic performance of the urine signatures was slightly inferior to the serum ones.

To further assess the cross-laboratory performance of the biofluid classifiers, we conducted external validation in the test set. We compared the metabolic differences between ESCC and colorectal cancer

**Fig. 4 | Changes in serum and urine metabolism in ESCC patients before and after surgery. A–L** OPLS-DA score plots of $^1$H-NMR serum spectra (**A–C**) and urine spectra (**G–I**) between experimental groups. Red: Pre-operation; Yellow: Post-operation; Blue: Healthy control (HC). Metabolic pathway analysis of distinguishing metabolites in serum (**D–F**) and urine (**J–L**) of ESCC patients. Relative betweenness centrality was the selected node importance measure for pathway topological analysis. The precise *p*-values for metabolic pathway analysis were provided in the Source Data without adjustments. (**A, D, G, J**, left panel) Pre-operation group vs. healthy group; (**B, E, H, K**, middle panel) Pre-operation group vs. post-operation group; and (**C, F, I, L**, right panel) post-operation group vs. healthy group. **M, N** Mantel test quantified the degree of correlation between tissue metabolome

and serum metabolome, together with the urine metabolome in early-ESCC patients. The key metabolites (potential biomarkers) in the alanine, aspartate and glutamate metabolism pathway, or arginine and proline metabolism pathway, from tissue profiles, were compared with serum and urine differential metabolites, respectively. Mantel statistics are provided in the right area in the plot. The network heatmap showed the correlation between the principal tissue biomarkers and biofluid metabolites (edge color denotes the statistical significance (*p*-values were determined by two-sided *t*-tests without adjustments for comparisons), and edge width corresponds to Mantel's *r* statistic for the corresponding distance correlations; the color gradient in the boxes represents Pearson's correlation coefficients). Source data are provided as a Source Data file.

**Table 2 | Serum and urine biomarkers are closely linked with the occurrence and progression of ESCC**

| Serum signatures | AUC (95% CI) | Urine signatures | AUC (95% CI) |
|---|---|---|---|
| alpha-Ketoisovalerate | 0.984 (0.954–1.000) | 3-Hydroxybutyrate | 0.673 (0.561–0.785) |
| Arginine | 0.887 (0.827–0.946) | Ascorbate | 0.764 (0.669–0.859) |
| Asparagine | 0.948 (0.916–0.980) | Dimethylamine | 0.670 (0.550–0.791) |
| beta-aminoisobutyrate | 0.893 (0.820–0.965) | Inosine | 0.776 (0.683–0.870) |
| Choline | 0.977 (0.959–0.995) | Malonate | 0.751 (0.650–0.852) |
| Citrate | 0.920 (0.875–0.966) | N-Methylnicotinamide | 0.834 (0.749–0.919) |
| Glutamate | 0.947 (0.906–0.989) | para-Hydrxoyphenylacetate | 0.735 (0.636–0.835) |
| Glutamine | 0.725 (0.638–0.813) | Riboflavin | 0.759 (0.660–0.857) |
| Histidine | 0.809 (0.736–0.883) | Taurine | 0.691 (0.585–0.798) |
| Low-density lipoprotein | 0.887 (0.834–0.940) | Trigonelline | 0.882 (0.816–0.949) |
| Methionine | 0.922 (0.880–0.964) | | |
| N,N-Dimethylglycine | 0.990 (0.976–1.000) | | |
| Ornithine | 0.893 (0.835–0.951) | | |
| Phenylalanine | 0.999 (0.997–1.000) | | |
| Sarcosine | 0.894 (0.841–0.947) | | |
| Trimethylamine | 0.928 (0.885–0.971) | | |

*AUC* areas under the curve, *95% CI* 95% CI confidence Interval.

(CRC), a common digestive tract tumor, and HCA showed distinct metabolic profiles between the two cancers (Fig. 7B). The serum joint model had a lower diagnostic efficiency for CRC, with an AUC of 0.733 (Fig. 7C). In addition, the metabolic profiles of four different types of esophageal tumors, including esophageal adenocarcinoma (EAC), gastroesophageal junction adenocarcinoma (GEJ), esophageal undifferentiated carcinoma and stromal tumors, differed from those of ESCC (Fig. 7D, E), further confirming the unique metabolomics signatures of ESCC. Notably, in clinical practice, distinguishing early-stage ESCC from healthy populations poses a big challenge. Therefore, we used the serum and urine panels to predict 18 new early-stage ESCC cases (with 18 serum and 18 urine samples) in the test set, with an accuracy of 0.994 and 0.879, respectively (Fig. 7F, Supplementary Table 4). The multi-center results confirmed that the biofluid classifiers captured the metabolic changes in early ESCC.

Logistic regression was used to analyze the metabolite characteristics to identify the risk and protective factors for early-stage ESCC. We made a nomogram (Supplementary Fig. 6A) and validated it with calibration plots and decision curve analysis (DCA) (Supplementary Fig. 6B, C). Notably, in the Nightingale dataset from prospective UK Biobank (UKB) cohort closer to real-world data, the pre-diagnosis levels of Clinical LDL-C, Glutamine, and 3-Hydroxybutyrate in plasma were associated with the risk of future esophageal malignancy onset[18]. For each 1-SD increment in the concentration of these three biomarkers, the median risk ratio decreased or increased by 0.90, 0.89, and 1.14 times, respectively (Supplementary Fig. 6D), which was consistent with the down-regulation of LDL (low-density lipoprotein) and glutamine, and the up-regulation of 3-hydroxybutyrate in the ESCC biofluids in this study.

## Discussion

Global predictions for ESCC across 185 countries or territories indicate an estimated 806,000 new cases by 2040 if incidence rates remain stable[1]. Despite significant advances in treatment over the past 20 years, the survival rate remains low due to its late diagnosis. Meanwhile, research has shown that ESCC screening emerges as the most cost-effective option based on ranking the cost per life-year saved to China's per capita GDP ratio[23]. Therefore, there is an urgent need to discover and establish biomarkers for ESCC screening or early diagnosis. It is well known that metabolic changes precede malignant histological alterations, however, it remains to be confirmed whether detectable characteristic metabolomics exists in tissues and biofluids for ESCC early diagnosis[14]. In this study, we characterize a comprehensive metabolome atlas of 1153 ESCC tissues, adjacent normal mucosae, pre- and post-operative sera and urines from 560 participants across three hospitals, using four platforms (including $^1$H-NMR and targeted MS) combined with machine learning and WGCNA. We demonstrated that changes in 'alanine, aspartate and glutamate metabolism' were prevalent throughout the progression of ESCC, from normal mucosae to early and to advanced stages of tissues, and from tumor-bearing to tumor-resected recovery phase and to healthy status in biofluids, suggesting that it is a characteristic molecular event in ESCC evolution. Based on the tissue metabolic characteristics consistently identified by NMR and targeted MS technique, we developed and optimized NMR-based 16 serum and 10 urine metabolic signatures that could reliably reflect the characteristic metabolic features of the ESCC tissues. Simplified NMR-based classifiers incorporating any five serum or urine metabolite signatures can effectively diagnose and predict early ESCC, making it suitable for clinical screening and thus

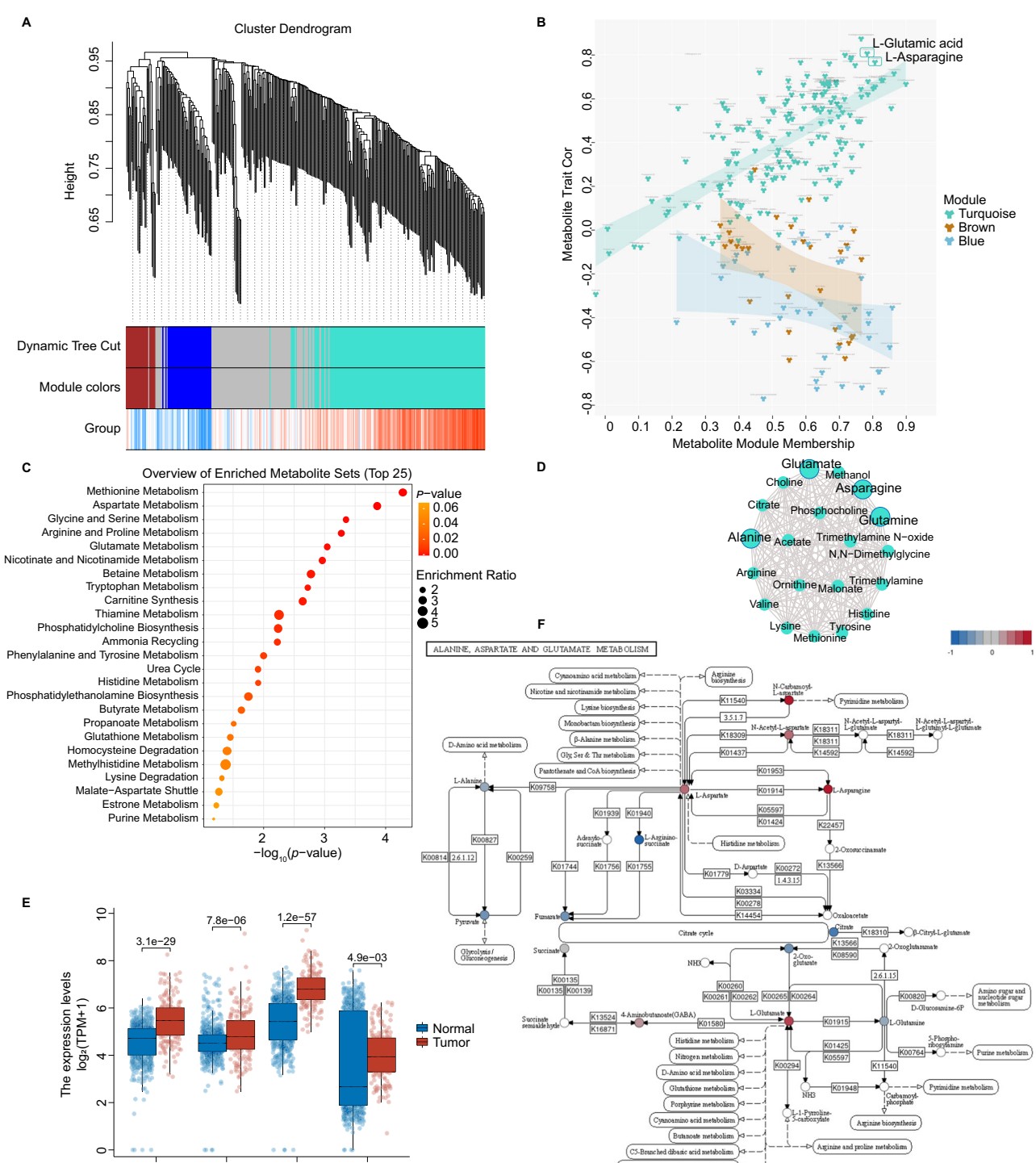

**Fig. 5 | WGCNA analysis, differential analysis of metabolic enzyme expression and critical metabolic pathways uncover the involved mechanisms. A** Early-ESCC tissue vs. normal mucosa: WGCNA cluster dendrogram groups differential metabolites into distinct metabolite modules (with different colors) defined by dendrogram branch cutting. Turquoise, blue and brown were the most strongly associated modules between tumor and normal tissue. **B** Early-ESCC tissue vs. normal mucosa: scatter plot showing the correlations between metabolite module-membership scores and trait. The turquoise, blue or brown hue denotes each module's metabolites of interest. **C** Early-ESCC tissue vs. normal mucosa: functional enrichment analysis of the metabolites in the turquoise module. The selected pathway enrichment analysis method is Globaltest. The dot plot summarizes the most significant metabolite sets identified during the enrichment analysis. The size of the dots per metabolite set represents the enrichment ratio, and the color

represents the *p*-value. **D** Pre-operative vs. post-operative serum groups: hub metabolites in the only crucial WGCNA module specific to the metabolic changes of ESCC patients before and after esophagectomy. **E** Differential expression (log2FC) of the principal metabolic enzymes in the alanine, aspartate, and glutamate metabolism pathway, using TCGA-ESCA data (red, *n* = 182 biologically independent tumor samples) and GTEx data (blue, *n* = 666 biologically independent normal samples). In the box plot, the central black line represents the data median, while the vertical lines correspond to the upper quartile and lower quartile. The data are presented as mean values ± standard error of the mean (SEM). Mann-Whitney U-test (Wilcoxon rank sum test) was performed for comparison, and the *p*-values were obtained. **F** Summary of the alanine, aspartate, and glutamate metabolism pathway including metabolites, enzymes and transporters where relevant (blue: down-regulation; red: up-regulation). Source data are provided as a Source Data file.

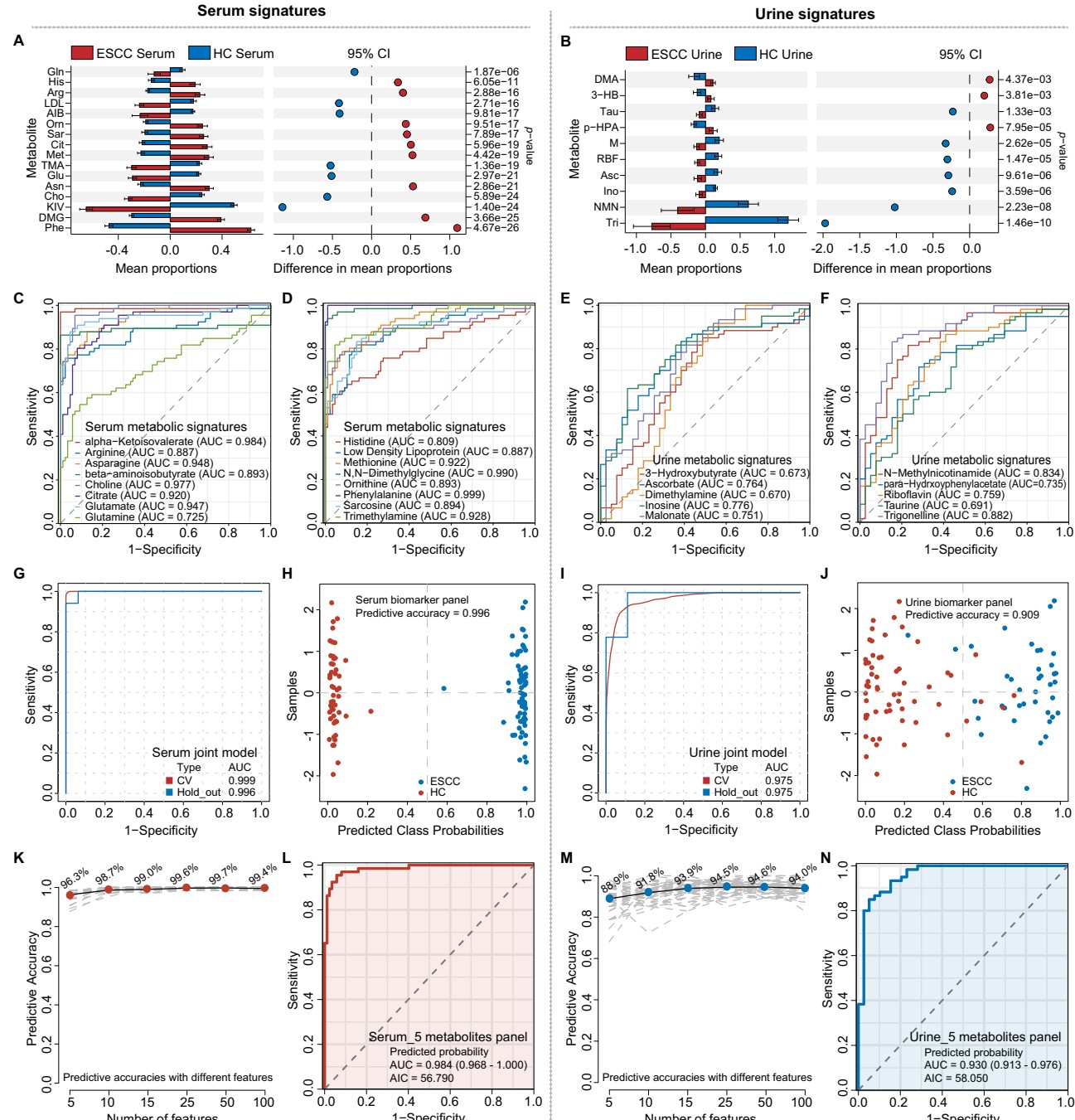

**Fig. 6 | Internal validation of serum and urine metabolite biomarkers.**
**A**, **B** STAMP analysis indicates expressional differences of serum (**A**) and urine (**B**) metabolite biomarkers between the ESCC ($n = 54$ biologically independent serum samples; $n = 54$ biologically independent urine samples) and HC groups ($n = 87$ biologically independent serum samples; $n = 39$ biologically independent urine samples). The results were divided into two parts: 1) Bar Chart on the left side: The red bars represent the ESCC group, while the blue bars represent the HC group. The length of each bar indicates the proportion of expression. The black lines on the bars represent the Standard Error of the Mean (SEM). 2) Dot Plot on the right side: When a differential metabolite has a higher abundance in the ESCC group than the HC group, it is represented by a red dot to the right of the dashed line. Conversely, if a metabolite's expression is lower in the ESCC group than in the HC group, it is marked with a blue dot to the left of the dashed line. The distance of the dot from the central dashed line represents the magnitude of the difference (95% CI). The vertical axis on the right displays the *p*-values for the differences (two-sided Welch t-test without adjustments for comparisons, FDR-adjusted), arranged from smallest to largest. The mean, standard deviation (SD), and SEM of the metabolite signatures in both groups have been provided in the source data. 95% CI: 95% Confidence Interval. **C**–**F** AUC of ROC curves for the 16 serum (**C**, **D**) and 10 urine (**E**, **F**) metabolic signatures discriminating ESCC from HC in the validation set, respectively. **G**–**N** ROC curves and AUCs from cross-validated serum (**G**) or urine (**I**) joint models (red), run on the 30% hold-out test dataset (blue). Prediction of class probabilities (average of the cross-validation) for each sample using the best classifier based on serum (**H**) or urine (**J**) AUC. Predictive accuracies of SVM models with different numbers of serum (**K**) or urine (**M**) features. Image shows the average of each sample's predicted class probabilities across 100 cross-validations. As the algorithm used a balanced sub-sampling approach, the classification boundary is located at the center ($x = 0.5$, the dotted line). ROC curve shows the efficacy of five serum (**L**) or five urine (**N**) metabolites (with the lowest AUC value) combined with the logistic regression model of ESCC. CV cross-validation. Source data are provided as a Source Data file.

**Table 3 | ROC information for serum or urine biomarker panels**

| Predicted probability panels | AUC (95% CI) | Cut-off | SE | SP | Accuracy | PPV | NPV | Youden |
|---|---|---|---|---|---|---|---|---|
| Serum panel (five metabolites) | 0.984 (0.968–1.000) | 0.502 | 0.955 | 0.943 | 0.948 | 0.926 | 0.965 | 0.897 |
| Urine panel (five metabolites) | 0.930 (0.913–0.976) | 0.833 | 0.850 | 0.949 | 0.889 | 0.962 | 0.804 | 0.799 |

*AUC* areas under the curve, *95% CI* 95% CI confidence interval, *SE* sensitivity, *SP* specificity, *PPV* positive predictive value, *NPV* negative predictive value.

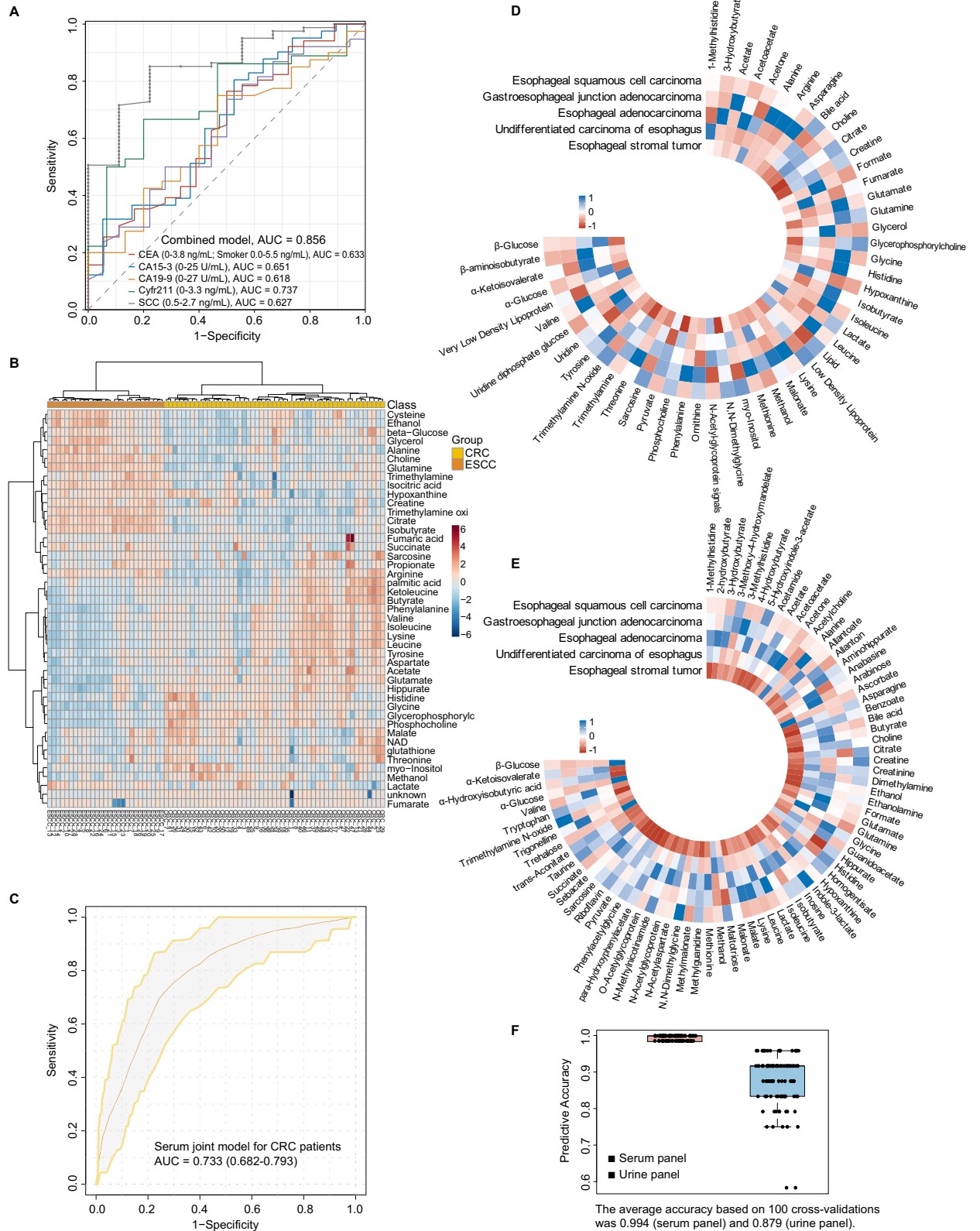

**Fig. 7 | External validation of the biofluid models using multi-center data.**
**A** ROC analysis for the five tumor biomarkers commonly used in the clinic (CEA, CA15-3, CA19-9, Crfr211, SCC) discriminating ESCC from HC. **B** Unsupervised hierarchical clustering of ESCC and CRC groups across all metabolites (Ward's method clustering). Yellow: CRC; Orange: ESCC. **C** ROC curve shows poor diagnostic efficacy of the serum joint model for CRC patients. Shaded areas represent the 95% CI of the corresponding ROC curves. **D**, **E** Circos heatmap was used to compare the serum (**D**) and urine (**E**) metabolic profiles of different pathological types of esophageal tumors, including ESCC, EAC, GEJ, undifferentiated carcinoma of the

esophagus and esophageal stromal tumors. **F** Performance of SVM-based classifiers was examined by ROC curves and evaluated by 100-fold cross-validation. The black dots in the box plot represent the predictive accuracy of the serum or urine panels in distinguishing early stage ESCC (red, $n = 18$ biologically independent early-stage serum samples; blue, $n = 18$ biologically independent early-stage urine samples) from HC groups. Notably, the serum panel data points exhibit proximity, while those of the urine panel are more dispersed. Source data are provided as a Source Data file.

---

providing a potential model for adoption and implementation in other laboratories.

Current clinical diagnostic methods for ESCC have limitations. Successfully developing a novel diagnostic approach requires new tools to surpass existing methods. The potential of liquid biopsies has been highlighted by studies showing that they can track the evolutionary dynamics and heterogeneity of tumors and detect the early emergence of diseases[24]. It can be used to examine cancer-derived circulating tumor cells (CTCs), circulating nucleic acids such as circulating tumor DNA (ctDNA) and cell-free DNA (cfDNA), extracellular vesicles (EVs), tumor-educated platelets, proteins, and metabolites[25]. Due to their abundance across biofluids, especially the cfDNA methylation-based technology, which can faithfully reflect tissue-SP with high tissue origin accuracy, liquid biopsy offers a promising strategy for the early detection of various human cancers, including ESCC[26–28]. However, the high cost and complex isolation procedures of the above-mentioned methods need to be considered. In the era of liquid biopsy, metabolomics is emerging, representing the clinical phenotype and metabolic reprogramming[15,24,29–31]. NMR-based metabolomics is robust, reproducible, cost-effective and sufficiently sensitive for clinical testing, no matter whether it involves a large sample size of tens of thousands, multi-center studies spanning many years, or cancer patients with only non-specific symptoms before onset[18,32–35]. Meanwhile, triple quadrupole (TQ) MS operated in MRM mode and coupled with LC or GC instruments are ideal complementary techniques to NMR due to their broad dynamic range and high SE[36,37]. Previous studies have proposed metabolic biomarkers for ESCC based on blood, urine, paired pre- and post-operative serum, or multiple serum-based analytical detection platforms[38–42]. However, these results were generally inconsistent or even contradictory between studies, due to potential factors, such as non-standardized sample collection processes, single and unmatched sample types, incomplete control designs, lack of validation or inconsistent detection methods. Tumor tissues contain global metabolic information at metabolic enzymes and metabolites levels. Investigations originating from tissue analysis are considered a more robust approach, offering explicit biochemical insights into disease pathogenesis and responses to stimuli[43]. Therefore, detecting cancer-specific metabolites in ESCC tissues first, and then measuring the metabolic phenotypes in biofluids that could reflect the characteristic metabolic features of the ESCC tissues, will be more likely to define specific and functional clinical biomarkers.

In this study, we first used NMR and MS to identify the potential biomarkers in early-stage ESCC tissues that play a central role in global metabolism homeostasis. 'Alanine, aspartate and glutamate metabolism' and arginine-related metabolic pathways were observed to be remarkably disrupted throughout the progression of ESCC from normal to early and to advanced stages. Interestingly, the expression trends of biomarkers and significant pathways in early-ESCC tissues in our study were highly consistent with those detected in ESCC tissues by Zeper Abliz's team using high-resolution MS imaging spatially resolved metabolomics, including downregulation of Glutamine and up-regulation of Glutamate, Proline, Uridine, Histidine, Short-chain FAs, Arginine, and Ornithine[44]. This further validated the reliability of our NMR-based results. We then used biofluid-based strategies to minimize the impact of tumor heterogeneity. By using a strictly

matched sample design and correlation analysis, we addressed the challenges of high-dimensional datasets, and identified NMR-based longitudinal blood and urine metabolites, which were closely associated with tissue metabolome in early-stage ESCC patients. Using WGCNA-based mechanistic exploration and quantitative depiction of KEGG metabolic pathways, we discovered that the most significant metabolic reprogramming in serum and urine was primarily involved in the 'Alanine, aspartate and glutamate metabolism' pathway, from tumor-bearing to tumor-resected phase and to healthy status in biofluids, consistent with the characteristics metabolic disorders in early-ESCC tissues. Therefore, we proposed that NMR-based biofluid metabolites can reflect the characteristic molecular events of early-ESCC tissues, and have great potential for ESCC early detection.

Our findings reveal varying degrees of disruption across most categories of metabolic pathways during ESCC progression, and this is because human metabolism is a complex, resilient and sophisticated regulatory ecosystem where a single change can affect the entire system. However, this study primarily focuses on observing the longitudinal metabolic changes specific to ESCC tissues. From this perspective, changes in amino acid metabolism appear notably active. Amino acids serve essential roles within tumors and their microenvironment. First, they are vital nutrients for all cell types, contributing substantially to cancer cell survival and growth. For example, glutamine is largely anaplerotic and relinquishes both amine groups to support the TCA cycle[45]. Second, enhanced biosynthetic activities are an essential feature of metabolic reprogramming in cancer, and amino acids play crucial roles in the synthesis of proteins, lipids, and nucleic acids[46]. Third, proliferating cancer cells accumulate reactive oxygen species, damaging macromolecules and eventually causing cell death. Cancer cells rely on glutamine, glycine, and cysteine to synthesize glutathione and regulate redox balance to address this problem[47,48]. In addition, amino acid derivatives contribute to epigenetic regulation and immune response associated with tumor occurrence and metastasis[49]. We also illustrate how the 'Alanine, aspartate and glutamate metabolism' pathway, a characteristic molecular event in ESCC, can inspire potential metabolic liabilities as therapeutic targets. This pathway involves key molecules such as GLS, SLC1A5, ASNS and Glutamine. GLS and SLC1A5 inhibitors have shown efficacy in preclinical models and are under trials; Glutamine can act as an intercellular metabolic checkpoint and modulate tumor immunity; and ASNS could be a vulnerability in gastric and liver cancer tumor cells, suggesting l-asparaginase as a potential therapy[50–52]. These collectively suggest that concurrent metabolome analysis offers valuable insights for early detection, biological significance studies and targeted therapies for ESCC.

Study design also plays a vital role in developing high-performance biomarker panels. On the basis of identifying characteristic molecular events of ESCC evolution, multiple phases of biomarker development and advanced data interpretation methods are also required[53]. First, we identified 16 serum and 10 urine metabolic signatures in the discovery and validation sets (internal validation). The identified metabolites encompass both common classic metabolites (such as glutamate, asparagine, etc.) and relatively niche but specific molecules associated with ESCC (such as alpha-Ketoisovalerate, 3-Hydroxybutyrate, etc.). These biofluid signatures were closely

associated with the occurrence of early-ESCC tissues, and can be stably detected by NMR. Considering the simplicity and flexibility of clinical promotion, we determined the optimal number of metabolites in the present model and its components by using machine learning algorithms, which has proven successful in the early detection of other diseases[54]. Any five combinations of 16 serum or 10 urine metabolic signatures showed high accuracy in clinical validation and test set (external validation). Combining 'classic molecules' and 'niche molecules' to build a model might be a better choice in clinical applications. Our approach is superior to conventional methods, has high SP for ESCC, is sensitive for early prediction, and advances the diagnostic window of ESCC to the early histological or even pre-cancerous stages, and could be used as an indicator to detect different digestive tract abnormalities. Therefore, we propose and emphasize that this could be a valuable contribution to liquid biopsy in the early detection of ESCC. If an individual is identified as 'high risk' through this screening strategy, further endoscopy and image-based examination can be recommended for personalized clinical management and intervention.

Our study has several strengths. First, it combined multi-platform metabolomics and matched study design to reveal ESCC-characteristic molecular events. Second, metabolomic analysis of tumor tissue is not susceptible to confounding effects, and we identified NMR-based serum and urine signatures that can be traced back to tumors through cross-talk of tissue and biofluid metabolic markers. Third, we successfully identified minimalistic metabolic biomarker panels with robust and generalizable performance in early-stage ESCC detection. This study also has limitations. First, it was limited to the southern Chinese population, further research is required to establish its generalizability to other populations. Second, due to the limited matched early-stage sample size, it used internal and external validation rather than multiple testing correction. Therefore, more large-scale prospective cohort studies in different regions are needed to confirm our findings under the control of confounding factors. Third, following the identification of tissue-specific biomarkers detectable in biofluids, we need to promptly validate them in high-risk populations (including low-grade intraepithelial neoplasia (LGIN), high-grade intraepithelial neoplasia (HGIN), gastroesophageal reflux disease (GERD), Barrett's, etc.) to refine the model further.

In conclusion, our findings demonstrate that there are NMR-detectable characteristic metabolomics in ESCC tissues and biofluids, and significant metabolic features in biofluid could faithfully reflect the metabolomics signatures in ESCC tissues. NMR-based simplified serum or urine metabolic classifiers exhibited high accuracy and SP in detecting ESCC, regardless of the histological stage of the disease, and can therefore be used as potential non-invasive ESCC early screening tools. Moving forward, we plan to expand the dataset with esophageal pre-cancerous lesions and follow-up data to build a metabolic timeline, while also incorporating metabolic data from different anatomical sites of the esophagus to construct spatiotemporal metabolomics of ESCC. Furthermore, we will conduct integrated multi-omics analyses, including metabolome, proteome and microbiome, to shed light on metabolites' origins and functional mechanisms. This work is ongoing.

## Methods

### Subject recruitment and clinical sample distribution
The study protocol has been approved by the Ethics Committee of Shantou University Medical College (#2021-92, #2022-103) and registered at the Chinese Clinical Trial Registry (Registration number: ChiCTR2300073613). Sample collection followed established biobanking protocols and ethical and legal standards, following informed consent. The report follows the Standards for Reporting of Diagnostic Accuracy Studies (STARD) reporting guideline.

We conducted a retrospective case-control study on 1153 samples from 560 participants across three medical centers in southern China. The diagnosis of patients was made using endoscopy combined with

biopsy, imaging, clinical symptoms, medical history, and further confirmed by histopathological examination of the surgically resected specimen. Healthy subjects from the high-incidence coastal area of ESCC were also included, and all clinical examinations were within normal ranges. The exclusion criteria were as follows: (1) subjects with a current or past history of other malignancies and history of gastrointestinal operations, (2) patients with neoadjuvant treatment before operation, (3) participants missing clinical information, (4) bacterial infection or use of antibiotics or probiotics within one month before surgery, and (5) presence of hypertension, diabetes, or other metabolic diseases. We rigorously applied inclusion and exclusion criteria to ensure that the study population is closely representative of the target population, thereby minimizing selection bias. The preliminary inclusion for this study involved 568 participants from three centers, with eight individuals excluded for the following reasons: patients with neoadjuvant treatment before operation ($n = 5$), participants missing clinical information ($n = 1$), and presence of hypertension, diabetes, or other metabolic diseases ($n = 2$).

The distribution of data from three clinical centers was as follows. Discovery set from center 1 ($n = 362$ biologically independent samples): 181 ESCC tissue samples and their corresponding 181 normal mucosae were collected between 2016 and 2020. Samples were detected using 400 MHz $^1$H-NMR and reanalyzed for the discovery set. The collection process and ethics statement can be found in the previously reported articles[20,21]. Validation set from center 2 ($n = 450$ biologically independent samples): 54 ESCC patients were included in this study, whose matched ESCC tissues, normal mucosae, pre- and one-week post-operative serum and urine samples, as well as their clinical phenotype data were obtained between 2021 and 2022, resulting in a total of 324 samples. 126 healthy subjects (including 87 serum and 39 urine samples) with no history of gastrointestinal problems were obtained as control groups. The metabolite analysis in this validation set was conducted using 600 MHz NMR and MS-based metabolomics techniques. Test set from center 3 ($n = 341$ biologically independent samples): 18 early-stage ESCC serum and urine specimens each; 24 matched tissue and longitudinal biofluids from 4 different pathological types of esophageal tumors patients, including esophageal adenocarcinoma (EAC), adenocarcinoma of the esophagogastric junction (GEJ), undifferentiated carcinoma of the esophagus, and esophageal stromal tumors; 104 colorectal cancer CRC tissues and their 104 normal mucosae; 42 control serum and 31 control urine specimens, were collected between 2021 and 2022. All specimens were subjected to 600 MHz NMR-based metabolomics.

To further investigate protective or risk factors within the EC metabolite signatures, we conducted a comprehensive literature search to identify publicly available datasets of NMR-based metabolomics for malignant neoplasm of the esophagus. As part of this investigation, we utilized data from a prospective UK Biobank (UKB) cohort. A total of 118,461 plasma samples were randomly selected from the full UKB prospective cohort, which included 502,543 participants[18]. Among them, 346 cases of malignant esophagus neoplasms (ICD-10 codes: C15; $p < 0.05$) occurred. In this study, biofluid samples were analyzed using NMR following a certified protocol, and 37 metabolic markers underwent rigorous clinical validation. The complete results of all 249 measured biomarkers can be accessed at https://biomarker-atlas.nightingale.cloud/.

### Statistics and reproducibility
Regarding the sample size estimation, we used the PASS (Power Analysis and Sample Size) software and relevant formulas to estimate the sample size based on the standard deviation, discrimination, test level, and test efficiency of metabolic biomarkers detection obtained in our previous work. When the sample size estimation parameters were set to SP of 90 ± 10%, SE of 80 ± 10%, significance level (α) = 0.05, confidence level (1-α) of 0.95, and two-sided test type, the required sample

size per group was estimated to be $n = 44$. We tried to make the sample size of multiple subgroups exceed this threshold in this project. The cut-off value for the correlation coefficient (r) was determined based on the sample size; a p|corr| ≥ 0.294 indicates a significant correlation between the two types of metabolites in the group. The $p$-value was obtained from the Mann–Whitney $U$-test of metabolite concentration, and the FC values represented the concentration ratio of each metabolite between the pairwise groups. Moreover, due to the limited sample size of early-stage ESCC patients, metabolomics studies can be conducted with as few as 16 cases per group, according to a previous study in *Nature Communications*, which showed that considering SE, reproducibility, detection limit, linearity and dynamic range, selectivity, identification, coverage, etc., the minimum optimal sample size for metabolomics and other omics studies was 16 cases, with an average power of at least 0.8, false discovery rate (FDR) of 0.05, and initial Cohen's d of 0.8[55].

The study participants were consecutively and randomly enrolled. Tissue samples were allocated to the NMR platform in random order in the discovery and validation sets. For paired analysis, the serum and urine samples from the same patient were subjected to corresponding NMR and targeted MS analyses, following the order of tissue samples. The study participants were consecutively and randomly enrolled. Tissue samples were allocated to the NMR platform in random order in the discovery and validation sets. For paired analysis, the serum and urine samples from the same patient were subjected to corresponding NMR and targeted MS analyses, following the order of tissue samples. The histopathological results, which serve as clinical reference testing data, were kept confidential to individuals responsible for processing and setting up the metabolomics testing (index tests). The investigators were blinded to group allocation when performing NMR or MS data acquisition. No adverse effects were related to using the reference and index tests.

## Sample preparation

We strictly followed the same detailed protocols and used the same brand of reagents and consumables for sample collection[16,56]. (1) Tissue samples, including tumor tissues and distant non-cancerous tissues (5 cm away from the edge of the tumor), were obtained under the guidance of an experienced pathologist without compromising the patients' pathological examinations. The collected tissues were rinsed with PBS to avoid contamination, as well as to remove excess water, and then quickly frozen in liquid nitrogen to arrest enzymatic or chemical reactions. Samples were subsequently stored at −80 °C until metabolite extraction. (2) Serum samples: Fasting blood was drawn into additive-free vacuum blood collection tubes and attention was paid to avoid hemolysis. Samples were coagulated naturally for 30 min and then centrifuged at $5000 \times g$ at 4 °C for 10 min (Thermo Scientific Sorvall ST 16R centrifuge, TX-400 rotor). Following centrifugation, the serum supernatant was aliquoted into storage tubes, immediately frozen in liquid nitrogen, and then stored at −80 °C until further analysis. (3) Urine samples: Patients were instructed to collect morning midstream urine. After mild centrifugation, the urine was aliquoted into storage tubes, promptly frozen in liquid nitrogen, and then stored at −80 °C until further analysis.

The pre-processing workflow for clinical samples in the NMR method was optimized using previous literature and practical considerations[16,57,58]. Tissue homogenate preparation: Tissue samples weighing 300 mg were ground using a 60 Hz grinder at 4 °C for 1 min in a mixture of 4 mL/g of $CH_3OH$ and 2 mL/g of ultrapure water. The resulting homogenate was then subjected to vortexing for 1 min after adding 4 mL/g of $CHCl_3$ and 4 mL/g of ultrapure water. The mixture was allowed to settle on ice for 15 min and subsequently centrifuged at $14,000 \times g$ for 10 min at 4 °C (Eppendorf centrifuge 5427R, FA-45-48-11 and FA-45-17 rotors). The supernatant was carefully transferred to a new 5 mL Eppendorf (EP) tube and treated with running nitrogen to remove the methanol. The resulting liquid was freeze-dried at −80 °C until further analysis. The freeze-dried powder was dissolved in 550 μL of PBS/$D_2O$ buffer (pH 7.4, 150 mM), which contained 0.05% TMSP-2,2,3,3-D4 (D, 98%) SODIUM-3-TRIMETHYLSILYLPROPIONATE (TSP, Cambridge Isotope Laboratories (CIL), Inc. #DLM-48, CAS #24493-21-8). After thorough mixing, the solution was centrifuged at $14,000 \times g$ for 10 min at 4 °C. Finally, 500 μL of the supernatant was transferred into a 5 mm NMR tube (NORELL, #S55 SECURE SERIES) for analysis. Serum preparation: A total of 400 μL of serum was combined with 200 μL of PBS/$D_2O$ buffer (pH 7.4, containing 0.9% NaCl) using vortexing to ensure thorough mixing. The resulting mixture was then centrifuged at $14,000 \times g$ for 10 min at 4 °C. Finally, the supernatant, which amounted to 550 μL, was carefully transferred into a 5 mm NMR tube for further analysis. Urine preparation: A total of 500 μL of urine was mixed with 50 μL of PBS/$D_2O$ buffer (pH 7.4, 1.5 M, containing 0.05% TSP) using vortexing to ensure thorough mixing. The mixture was then centrifuged at $14,000 \times g$ for 10 min at 4 °C. After centrifugation, 500 μL of the supernatant was transferred into a 5 mm NMR tube for further analysis.

Targeted quantitative MS-based metabolomics detection was conducted on early-stage ESCC tissues from the same batch in the validation set at another accredited diagnostic laboratory. For LC/MS-MS, 1000 μL of acetonitrile-methanol-$H_2O$ (2:2:1, containing isotope internal standards) was added to 50 mg samples. The samples were then homogenized, sonicated, and this process was repeated three times. After incubating the samples at −40 °C for 2 h, they were centrifuged at $17,000 \times g$ for 15 min at 4 °C. Subsequently, 800 μL of supernatant from each sample was transferred to a new EP tube and dried using a centrifugal concentrator. Next, 160 μL of 60% acetonitrile was added to reconstitute the dried samples. The mixture was vortexed for 30 s, and sonicated in an ice-water bath for 5 min, followed by centrifugation at $17,000 \times g$ for 15 min at 4 °C. Finally, 100 μL of supernatant from each sample was transferred into glass vials for LC–MS/MS analysis. For GC/MS-MS analysis, samples were collected in 2 mL EP tubes. Then, 1 mL of pure water was added and the mixture was vortexed for 10 s; the samples were homogenized and sonicated three times. Following centrifugation at $7000 \times g$ for 20 min at 4 °C, 800 μL of the supernatant was transferred into a new 2 mL EP tube. Subsequently, 100 μL of 50% $H_2SO_4$ and 800 μL of the extraction solution (containing 25 mg/L stock in methyl tert-butyl ether and 2-Methylvaleric acid as an internal standard) were added to the samples. The samples were then vortexed for 10 s, oscillated for 10 min, and sonicated in an ice-water bath for 10 min. After centrifugation, the supernatant was transferred into a vial for GC/MS–MS analysis.

## $^1$H-NMR and MS acquisition and data pre-processing

In brief, metabolites in tissue, serum and urine were measured using four platforms, including 400 MHz and 600 MHz $^1$H-NMR, together with LC/MS-MS and GC/MS–MS in MRM mode. The results obtained from NMR and MS approaches complement each other, enabling a comprehensive metabolome analysis.

The 400 MHz $^1$H-NMR spectrum analysis was conducted following the methodology outlined in our previously published article[57]. For the 600 MHz $^1$H-NMR spectra of the validation and test samples, a Bruker Advance NMR spectrometer (Bruker BioSpin, Germany) equipped with a triple resonance cryogenic probe operating at 600.13 MHz and 298.0 K was used. The NMR data acquisition was performed using TopSpin 3.2 software. Tissue extracts were subjected to a standard one-dimensional (1D) Nuclear Overhauser Effect Spectroscopy (NOESY, [RD-90°-t1-90°-tm-90°-ACQ]) presaturation pulse sequence with the following acquisition parameters: a spectral width of 12019.2 Hz, an acquisition time of 1.36 s, a relaxation delay (RD) of 2.0 s, a scan accumulation of 64 times, and a data point of 16 K. Serum samples were analyzed using a Car-Purcell-Meiboom-Gill (CPMG, [RD-90°-(t-180°-t)n-ACQ]) pulse sequence with a spectral width of

12,019.2 Hz, an acquisition time of 1.36 s, a RD of 4.0 s, a scan accumulation of 64 times, and a data point of 16 K. Urine samples were collected using a NOESY pulse sequence with a spectral width of 12,335.5 Hz. The acquisition time was 2.66 s, the RD was 4.0 s, the scan accumulation was 32 times, and the data point was 32 K.

The spectra were processed using MestReNova (version 14.0, Mestrelab Research, Spain). The chemical shifts were referenced to the singlet peak of TSP at δ 0.00 (for tissue and urine) or the doublet of endogenous lactate at δ1.33 (for serum) for spectral alignment. The spectral region containing residual water, urea, or methanol signals was removed to eliminate interference. The remaining spectral region (from δ 0.50 to δ 9.00) was then divided into discrete regions of 0.002 ppm. The resulting NMR spectral data were normalized to the total integral area to account for concentration differences between samples. The normalized data were then imported into SIMCA (version 14.1, Umetrics, Sweden) for pattern recognition analyses. The NMR signals were assigned to individual metabolites based on previous literature data, confirmed by the referenced chemical shift libraries on Human Metabolome Database (HMDB, http://www.hmdb.ca/) and Biological Magnetic Resonance Data Bank (BMRB, https://bmrb.io/)[57-60].

The targeted detection was performed by Biotree Biomedical Technology Co., Ltd. The core protocol of MS analysis was described in the literature[17]. For LC−MS Analysis, high metabolite coverage, including over 500 metabolites across 13 classes, was detected. Each substance has a standard, and absolute quantification was performed using isotope internal standard correction. Metabolic extracts were analyzed using an ACQUITY UPLC H-Class (Waters) ultra-high-performance liquid chromatography system, utilizing the Atlantis Premier BEH Z-HILIC Column (Waters, 1.7 μm, 2.1 mm × 150 mm) for chromatographic separation of target compounds. Mobile phase A consisted of ddH$_2$O: acetonitrile (8:2, v/v) with 10 mM ammonium acetate, while mobile phase B comprised acetonitrile: ddH$_2$O (9:1, v/v) with 10 mM ammonium acetate. Both phases were adjusted to pH 9 using aqueous ammonia. The auto-sampler temperature was maintained at 8 °C, and the injection volume was 1 μL. Additionally, AB SCIEX 6500 QTRAP+ triple quadrupole MS equipped with IonDrive Turbo V ESI ion source was used for MS analysis in MRM mode. The parameters of the ion source were as follows: Curtain Gas = 35 psi, IonSpray Voltage = +5000 V/−4500V, Temperature = 400 °C, Ion Source Gas 1 = 50 psi, and Ion Source Gas 2 = 50 psi.

GC-MS Analysis mainly detects seven short-chain and four medium-chain FAs. Qualitative and quantitative analyses were also performed using internal standard method. Metabolic extracts were analyzed using the SHIMADZU GC2030-QP2020 NX gas chromatography-mass spectrometer. The system employed an HP-FFAP capillary column, and a 1 μL aliquot of the analyte was injected in split mode (5:1, v/v). Helium was used as the carrier gas with a front inlet purge flow of 3 mL/min and a gas flow rate of 1 mL/min through the column. The initial temperature was maintained at 50 °C for 1 min, then increased to 150 °C at a rate of 50 °C/min for 1 min. Subsequently, it was raised to 170 °C at a rate of 10 °C/min for 1 min, further increased to 210 °C at a rate of 20 °C/min for 1 min, and finally raised to 240 °C at a rate of 40 °C/min for 1 min. The injection, transfer line, quad, and ion source temperatures were set at 220 °C, 240 °C, 150 °C, and 200 °C, respectively. The energy used was −70 eV in electron impact mode. MS data were acquired in Scan/SIM mode within the m/z range of 33-150 after a solvent delay of 3 min.

All MS data collection and quantitative analysis of target compounds were performed using SCIEX Analyst Work Station Software (version 1.7.2) and BIOTREE Bio Bud (version 2.1.4). Metabolite identification was performed using an in-house MS database. The preprocessing of MS raw data involved filtering individual metabolites to retain those with no more than 50% missing values. Missing values in the original data were simulated by multiplying the minimum value with a random number between 0.1 and 0.5.

## Metabolomic data processing and statistical analyses

In short, metabolomic data processing and statistical analyses (including univariate and multivariate analysis) were performed using SPSS software (version 26, IBM SPSS Statistics, USA), SIMCA program, R (version 4.2.1, https://www.r-project.org/) and MetaboAnalyst database (version 6.0, https://www.metaboanalyst.ca/).

Multivariate data analysis was conducted on the metabolomic data using the SIMCA program, following previously published methods[57,58]. Metabolite differences were determined using the Mann−Whitney $U$-test for non-normally distributed values and the student's $t$-test for normally distributed values. The statistical analysis was performed with a Benjamini−Hochberg-based FDR, setting the adjusted $p$-value < 0.05 as the significance level. SPSS software and R were used for this analysis.

Unsupervised principal component analysis (PCA), UMAP, and HCA were used to visualize the distribution and relationship between samples, such as clusters or outliers. Pareto scaling (Par), which involves mean-centering and dividing by the square root of the standard deviation of each variable, was optimized for supervised multivariate statistical analyses. This included OPLS-DA to extract and maximize the metabolome differences between different groups in the pattern recognition models. To confirm the results of the established models and avoid overfitting, we performed 200 iterations of the permutation test and CV-ANOVA. The Permutation Plot helps assess the risk that the current model is spurious, i.e., the model fits the training set well but does not predict Y well for new observations. The idea of this validation is to compare the goodness of fit ($R^2$ and $Q^2$) of the original model with the goodness of fit of several models based on data where the order of the Y-observations has been randomly permuted, while the X-matrix has been kept intact. In addition, we used various statistical measures including the correlation coefficient ($r$), VIP from the OPLS-DA model, the adjusted $p$-value, and FC from the univariate statistical analysis to identify significant differential metabolites.

In this study, we utilized O2PLS and Mantel test to analyze the correlation between NMR metabolomic and MS metabolomic data in early-stage ESCC tissue, as well as the potential biomarkers that match between tissue and biofluid samples[61,62]. Our aim was to identify more rigorous potential biomarkers. To analyze the individual metabolite signatures obtained, we performed differential analysis using STAMP software and ROC curve analysis[63]. We then used machine learning algorithms for signature selection. Before conducting further analysis, the data were standardized by sum and log-transformed. For the individual metabolite, we performed univariate ROC analysis using the pROC R package to calculate the AUC. The results were visualized using the ggplot2 R package. Additionally, we generated multivariate exploratory ROC curves using balanced sub-sampling by Monte-Carlo cross-validation (MCCV) in the MetaboAnalyst 6.0 database[64]. Two-thirds of the samples in each MCCV were used to evaluate the feature importance, while the remaining one-third was used to validate the classification models. This process was repeated 50 times to calculate the performance and 95% confidence interval (CI) for each model and determine the optimal number of features for maximum accuracy. The linear SVM and RF machine learning algorithms were employed for sample classification, and the mean importance measure of the SVM and RF was used for metabolite feature selection. The average accuracy of the model was based on 100 cross-validations. The selected metabolic signatures were then used to construct serum and urine joint models, with 70% of the samples used for model building and 30% reserved for internal validation. The simplified biofluid panels were evaluated based on the AUC (95% CI), SE, SP, and accuracy (probability

of correct classification). The optimal probability cut-off value was determined using the model's maximum Youden Index (SE + SP-1). Logistic regression, XGBoost, Gaussian Naive Bayes, and K-Nearest Neighbors algorithms were used to compared the diagnostic performance of biofluid panels. Samples predicted as early-stage ESCC or HCs were classified based on whether their predicted probability was less or greater than the cut-off value, respectively.

A multivariable logistic regression analysis was performed using the glm function to analyze the biofluid metabolite signatures. Backward step-wise selection was used with the likelihood ratio test and Akaike's information criterion as the stopping rule. In order to provide clinicians with a quantitative tool to predict the risk of early-stage ESCC, a metabolomics nomogram was developed based on the logistic model, utilizing the rms R package for visualization. Calibration curves were generated to assess the accuracy of the nomogram, and the Hosmer-Lemeshow test was conducted. DCA was also performed to evaluate the clinical usefulness of the nomogram, measuring the net benefits at different threshold probabilities using the rmda R package for visualization.

### Metabolic module and pathway analyses

To explore the potential mechanisms underlying the progression of ESCC, we utilized WGCNA and KEGG pathway analysis to evaluate differential metabolic modules and pathways in ESCC patients and HCs, as well as in pre- and post-operative ESCC patients.

The main objective of WGCNA analysis is to identify modules consisting of interconnected nodes, such as metabolites in this study[65]. These modules can then be used for summary measurements in subsequent analysis. Additionally, WGCNA can identify hub nodes (metabolites) that are highly connected and located centrally within the module. Previous gene expression studies have demonstrated that these hub nodes are more likely to be biologically relevant markers. The principles and formulas for applying WGCNA to analyze metabolomics data have been thoroughly described in existing literature[66,67]. The WGCNA R package was used to perform computations[68]. Initially, a correlation network of metabolites was constructed and eigen-metabolite scores (representing the first principal component) were derived from the identified modules. These scores were then utilized to assess the differential association between each module and the 'ESCC patients vs HCs' grouping, as well as the 'pre- vs post-operation' grouping. Furthermore, we identified potential hub metabolites within each module based on their high intramodular importance, which was determined by a strong correlation between the eigen-metabolite score for a given module and the individual metabolite level.

KEGG pathway analysis was conducted to analyze metabolites, including the contents within the driver modules and potential metabolite biomarkers[69]. The analysis was performed using MetaboAnalyst. Additionally, metabolites involved in the key pathways were quantified on the MS in MRM mode. The metabolic enzymes and transporters in the pathways were extracted from TCGA-ESCA and GTEx databases. The expression data were obtained from RNA-seq data in TPM format from TCGA and corresponding normal tissue data from GTEx, which were uniformly processed through the Toil pipeline. The UCSC XENA database (https://xenabrowser.net/datapages/) was used to access the data. Data analysis was performed using the stats and car R packages. Data pre-processing involved $log_2(value + 1)$, and statistical analysis was conducted using the Wilcoxon sum rank test.

### Reporting summary

Further information on research design is available in the Nature Portfolio Reporting Summary linked to this article.

## Data availability

The matched NMR and MS metabolomics data generated in this study have been deposited in the NIH Common Fund's National Metabolomics Data Repository (NMDR) website, the Metabolomics Workbench database under accession code Project ID PR001876: https://doi.org/10.21228/M87426. The NMR data from the previous study and the smaller subset of the test set are not available due to intellectual property agreements with different hospitals. Still, should collaboration be established, these data can be obtained from the corresponding author, Yan Lin (email: 994809889@qq.com). The processed metabolomics data are provided in the Supplementary Information/Source Data file. The RNA-seq data from TCGA-ESCA and GTEx databases can be downloaded from https://xenabrowser.net/datapages/. Source data are provided with this paper.

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

## Acknowledgements

We thank Professor Jianghua Feng of Xiamen University for providing valuable insights into NMR detection and data analysis. We thank Director Chunpeng Zheng of Shantou Central Hospital and Director Shaobin Chen of Cancer Hospital of Shantou University Medical College for supporting our multi-center validation efforts. We also thank Stanley Li Lin of Shantou University Medical College for contributing to language polishing. This study was supported by grants from the National Natural Science Foundation of China (82071973 to Y.L., 82020108016 to R.H.W.), Guangdong Basic and Applied Basic Research Foundation (2023A1515010326 to Y.L., 2022A1515220020 to C.C.M., 2020A1515011022 to Y.L.) and Key Research Platform and Project of Guangdong University (2022ZDZX2020 to Y.L.).

## Author contributions

Y.L. and C.C.M. supervised and directed this project. Y.Z. and C.C.M. are joint first authors. Y.L. and R.H.W. are the study guarantors. Y.Z., C.C.M., W.Y., T.OY. and J.H.L. collected the data. Y.Z., R.Z.C and L.X.K. were involved in data cleaning and verification. Y.Z. analyzed the data, and drafted the manuscript. Y.L., Y.Z., C.C.M., R.H.W., L.J.X. and Y.S.L. contributed to the interpretation of the results and critical revision of the manuscript for important intellectual content and approved the final version of the manuscript. All authors have read and approved the final manuscript.

## Competing interests

The authors declare no competing interests.
