## [Peer Review File · Nature Communications]

NMR and MS reveal characteristic metabolome atlas and optimize esophageal squamous cell carcinoma early detectionREVIEWER COMMENTS

Reviewer #1 (Remarks to the Author): expertise in machine learning classifier development

Well-written and detailed article, with no methodological weaknesses.
A few suggestions/comments:

It would be interesting to present the objective of the study more clearly.

Even though the performance is very good, it would have been interesting to compare these results with those obtained using other machine learning methods.

Reviewer #2 (Remarks to the Author): technical expertise in NMR MS-based metabolomics

The authors describe an interesting study of tissue, blood and urine metabolomics with the goal of identifying early biomarkers for esophageal cancer (EC). They have done a lot of nice work here, but I am also concerned that the results may well be "too good to be true," as they don't appear to line up with previous work. Therefore I am on the fence as to whether this should be published in Nature Communications. Nevertheless, the authors should address the following major issues:
Major:

1) The statistical analysis procedure should be described more clearly. While there are 3 centers, it seems that the first center was used to identify biomarkers which were then verified in center 2 and maybe a model was developed that was tested in center 3. Do I have that process described correctly? A better description and a figure, perhaps in SI would clear that up.

2) If my assumption on the process is correct, why not use the first data set to both detect the markers and develop the algorithm? Then the test set would be larger. Right now, the test set is very small, with only 18 early stage cases.

3) Sample collection details are critical for metabolomics studies as incorrectly collected samples cause bias. The authors need to describe in this manuscript how each of the sample sets were acquire, and if any of the patients were under anesthesia or had prior knowledge of their surgery before blood collection.

4) The incidence of EC is quite low and therefore a screening test that uses healthy controls is unlikely to be translated because the false positive rate would be too high. The translatable test would actually be one that compares EC patients with at risk patients, such as those with Barrett's or even GERD or acid reflux. The authors should at a minimum acknowledge this situation in their limitations section.

5) The results here are much better than previous papers which tended to be in the 80-90% AUC range. The authors detected some pretty common metabolites here, how do those metabolites perform in the other papers? Why, for example are glutamate and asparagine such strong markers here, but not in other's studies. It doesn't seem plausible.

6) Related to the point above, I appreciate that the authors included the data from Nightingale for comparison. It does seem that some of the metabolites that were identified here as biomarkers appeared in the Nightingale dataset. However, the odds ratio for that work is vastly different than what I can surmise here. For example, in the Nightingale data, the odds ratios are in the range of 0.9 to 1.1, they are more like 100 for some of the best performing metabolites in the current study, which seems like a disconnect. The authors need to explain why the serum metabolites measured in their study have low or no odds ratios in the Nightingale study, which was also an NMR based analysis.

Minor:

7) The text is too small in many figures and the description of what is in those figures is sometimes too limited.

8) Please provide more information on the targeted LC-MS data acquisition. Were the samples analyzed by the Asara lab, or were they run elsewhere? How many metabolites were targeted and how many were measured for each sample type? Similar questions about the GC-MS data acquisition.

Reviewer #4 (Remarks to the Author): expertise in esophageal cancer classification

This manuscript by Zhao et al. performed comprehensive metabolome profiling of 1,153 ESCC tissues, adjacent normal mucosae, pre- and post-operative sera and urines from 560 participants across three hospitals. The authors used different metabolomic platforms (including NMR and targeted MS). Data was analyzed by machine learning, logistic regression as well as WGCNA. They found that changes in alanine, aspartate and glutamate metabolism were prevalent throughout the progression of ESCC, from normal mucosae to early and to advanced cancers. The authors further developed NMR-based 16 serum and 10 urine metabolic signatures, which reflected the characteristic metabolic features of the ESCC tissues. Lastly, the authors showed that NMR-based classifiers incorporating any five serum or urine metabolite signatures can effectively diagnose and predict early ESCC.

Overall, this is a tour de force metabolic work and represents perhaps the largest scale metabolomic profiling in ESCC (if not any other cancer), therefore representing a huge effort and highly valuable resource for the research community. The data analysis was comprehensive and justified (with some weaknesses listed below) and the data interpretation was generally appropriate. However, I do have a list of questions/concerns which need to be improved in order for the audience to digest and understand better the work.

- 1) I did not find "Data availability" section in the main article. All of the raw and processed metabolomic data would need to be publicly available, which is a standard for Nature journals.
- 2) Early ESCC biomarker is a strong focus for this paper. However, only 18 early ESCC patients were profiled in the Test set from center 3 (Supplementary Table 1). Is this number actually sufficiently powered for the discovery? The authors need to show statistical analysis, particularly power analysis to demonstrate that with this low number, what fold change will be needed for the discovery and at what p value.
- 3) Relatedly, throughout the manuscript, no sample number was clearly given in the result section, which makes it very difficult to review because there are multiple cohorts, multiple stages of cancers, multiple types of samples, as well as multiple centers. Thus, the authors need to explicitly describe sample numbers and types for each experiment and each comparison in the result section.
- 4) What does it mean by "External permutation tests validated that the models were suitable for data analysis (Fig. 2D-F)?" it is unclear what do the dots represent and what are Q or R values.
- 5) Fig. 2J does not show p value, which is important. This can be plotted along X axis.
- 6) Fig.2G-I, the color and size of each bubble correspond to its p-value and pathway impact value, respectively. However, there is no color scale or size scale.
- 7) Any overlapped metabolites between early and late ESCCs? Those shared could be important ones to follow up.
- 8) Regarding the validation of the NMR results of early ESCC tissues using LC-MS/MS and GC/MS-MS, the description and presentation is difficult to follow (Figure 3). 1) Were LC-MS/MS and GC/MS-MS both used for the validation? If so, were LC-MS/MS and GC/MS-MS using the same customized panels? 2) How many of the 34 hits from the NMR results on early ESCC (Fig.2J) were validated by LC-MS/MS and GC/MS-MS? Were they validated by both LC-MS/MS and GC/MS-MS or just one of these two methods? 3) Line 133 "We then identified the top 30 metabolite variables with high correlation and weight in both datasets using the loading plot (Fig. 3D) and listed them in Supplementary Table 3 (right panel)." Are these 30 from the 34 hits from the NMR results on early ESCC (Fig.2J)?
- 9) The NMR profiles the global metabolomics, including amino acids, lipids, nucleotides, carbohydrates, etc. however, the most significant changes were seen mostly in amino acids (e.g, Fig.2K, 3D, Supplementary Fig. 4A-B). Indeed, one of the conclusion is "Alanine, aspartate and glutamate metabolism is a key functional pathway in ESCC progression", which are all amino acids. Can the authors discuss why amino acid metabolism seems to dominate these metabolic changes?
- 10) In the introduction, the authors described novel biomarkers for ESCC diagnosis, including serum miRNA, autoantibodies, somatic gene mutations, salivary exosomes, and artificial intelligence (AI)-assisted sponge cytology. DNA methylation markers is an important group and should be discussed.
- 11) The paper reads fairly cumbersome, and I recommend it to be edited by professionals.

De-Chen Lin, Ph.D.
University of Southern California

RESPONSE TO REVIEWERS' COMMENTS

Reviewer #1 (Remarks to the Author): expertise in machine learning classifier development

Well-written and detailed article, with no methodological weaknesses.

Response: We are very pleased with your strong support for our work.

A few suggestions/comments:

1) It would be interesting to present the objective of the study more clearly.

Response: We concur with this comment, and have added more detailed descriptions in the Introduction and Discussion section to clarify the objective of the study.

Measures taken:

In the Document "Article File",

1) "We aimed to investigate tumor tissue-specific metabolic biomarkers during the ESCC evolution, and then leverage them as references to develop and optimize NMR-based biofluid metabolic classifiers, that not only can faithfully reflect the characteristics metabolic changes in tissues, but also has high specificity and sensitivity for ESCC early diagnosis and screening." in Line (L) 78-81 **has been revised to** "We aimed to investigate tumor tissue-specific metabolic biomarkers during ESCC evolution, and then leverage them as references to develop and optimize biofluid metabolic classifiers based on NMR (a means that can better achieve "health equity"), which not only faithfully reflect the characteristics of tissue metabolic changes with high accuracy of tissue origin, but also has sufficient clinical sensitivity for ESCC early diagnosis and screening.", highlighted in red in the revised manuscript in **L 80-84**.

2) "ESCC continues to be a heavy global health burden with a high mortality due to its late diagnosis, despite considerable advances in traditional treatments." in L 358-359 **has been revised to** "Global predictions for ESCC across 185 countries or territories indicate an estimated 806,000 new cases by 2040 if incidence rates remain stable¹. Despite significant advances in treatment over the past 20 years, the survival rate remains low due to its late diagnosis. Meanwhile, research has shown that ESCC screening emerges as the most cost-effective option based on ranking the cost per life-year saved to China's per capita GDP ratio²", highlighted in red in the revised manuscript in **L 354-358**.

3) "Investigations originating from tissue analysis are considered a more robust approach, offering explicit biochemical insights into disease pathogenesis and responses to stimuli³." **has been added to** the revised manuscript, highlighted in red in **L 395-396**.

2) Even though the performance is very good, it would have been interesting to compare these results with those obtained using other machine learning methods.

Response: Thank you for the constructive suggestions. As requested, we compared our current results with other new machine learning methods, including XGBoost, Gaussian Naive Bayes and K-Nearest Neighbors to determine whether the panels constructed by serum and urine signatures in this study performed well. Considering the theoretically low-performing combined serum and urine

panels mentioned in L 282-286 of document "Article File", the obtained Model Metrics are presented in the **New Supplementary Table 12** below:

New Supplementary Table 12. Comparison of the theoretically low-performing simplified biofluid panels using different machine learning methods.

Model Metrics	Efficacy validation of serum panel				Efficacy validation of urine panel			
	Logistic regression	XGboost	Gaussian Naive Bayes	K-Nearest Neighbors	Logistic regression	XGboost	Gaussian Naive Bayes	K-Nearest Neighbors
AUC (95% CI)	0.984 (0.968 - 1.000)	0.979 (0.905-1.000)	0.977 (0.892-1.000)	0.944 (0.867-1.000)	0.930 (0.913 - 0.976)	0.842 (0.715-1.000)	0.874 (0.840-0.908)	0.826 (0.789-0.963)
SE	0.955	0.947	0.948	0.931	0.850	0.848	0.831	0.729
SP	0.943	0.939	0.929	0.819	0.949	0.894	0.863	0.933
Accuracy	0.948	0.929	0.922	0.907	0.889	0.932	0.848	0.841
PPV	0.926	0.893	0.889	0.965	0.962	0.923	0.846	0.706
NPV	0.965	0.948	0.948	0.919	0.804	0.747	0.833	0.725

Abbreviations: AUC: Areas Under the Curve; SE: Sensitivity; SP: Specificity; PPV: Positive predictive value; NPV: Negative predictive value

The comparative analysis demonstrates that various machine learning methods yield consistently similar results. Furthermore, the performance of the serum panel continues to outperform the urine panel slightly. We believe these additional data strengthen our findings.

Measures taken:

- 1) We have added **Supplementary Table 12** to Document "Supplementary Tables".
- 2) In the Document "Article File", "We also compared the diagnostic performance of biofluid panels using XGBoost, Gaussian Naive Bayes, and K-Nearest Neighbors algorithms. While the results closely approximated those obtained through the classical logistic regression method, the efficacy of the logistic regression model remained better (Supplementary Table 12), indicating that the machine learning methods we used were robust." **has been added to** the revised manuscript, highlighted in red in **L 289-293**.

Reviewer #2 (Remarks to the Author): technical expertise in NMR MS-based metabolomics

The authors describe an interesting study of tissue, blood and urine metabolomics with the goal of identifying early biomarkers for esophageal cancer (EC). They have done a lot of nice work here, but I am also concerned that the results may well be "too good to be true," as they don't appear to line up with previous work. Therefore I am on the fence as to whether this should be published in Nature Communications. Nevertheless, the authors should address the following major issues:

Response: Thank you so much for your insightful and constructive evaluation of our work. We have also observed the inconsistencies in previous metabolomic assays between different laboratories, especially biofluid metabolomics, which is sensitive to many potential confounders, such as environmental factors, lifestyle habits, phenotypic variations and comorbidities, etc. As highlighted in this authoritative review (*Nat Rev Mol Cell Biol*, 2016, 17, 451-459, doi:10.1038/nrm.2016.25), tissue-based investigations are deemed more reliable, offering explicit biochemical insight into disease pathogenesis and responses to stimuli³.

In light of this, based on our previous metabolome studies and the technical characteristics of NMR and MS, we proposed two scientific questions: 1) Are there ESCC-specific NMR detectable and MS verifiable metabolome alterations? 2) Can the metabolic changes in serum and urine reflect the characteristic metabolic changes in ESCC tissues? Therefore, after identifying the potential biomarker pool for ESCC, we further strictly design and collect the patient-matched ESCC tissues and adjacent normal esophageal tissues, pre-operative (with tumor burden) serum and urine, and one-week post-operative (recovery phase after tumor removal) serum and urine. We aim to minimize individual variances' biases and discern relatively robust, reproducible 'characteristic molecular events', and the corresponding reliable metabolic classifiers in biofluids.

The uncovered 'characteristic metabolic event' in ESCC tissues and biofluids and the corresponding differential metabolites reflected in biofluids did exhibit good diagnostic performance. These encouraging results lead us to believe that NMR-based biofluid metabolic fingerprint is feasible for ESCC early detection, which could be a novel contribution to liquid biopsy in early cancer screening.

We appreciate the opportunity to provide further clarification on this important point. We assure you that we have taken your feedback seriously and made every effort to address the issues raised. Please find below our point-to-point response:

Major:

1) The statistical analysis procedure should be described more clearly. While there are 3 centers, it seems that the first center was used to identify biomarkers which were then verified in center 2 and maybe a model was developed that was tested in center 3. Do I have that process described correctly? A better description and a figure, perhaps in SI would clear that up.

Response: Thank you for your careful review and good suggestion. Your description of the overall study design aligns with our intentions. We apologize for not clearly describing the statistical analysis procedure in our previous manuscript. In the revised version, we have provided a more detailed explanation of the workflow and have updated Supplementary Figure 1 accordingly,

which we believe could illustrate the process more clearly and intuitively. As you can see in the updated Supplementary Fig. 1, our study consisted of three phases: **discovery set from center 1**, **matched validation set from center 2**, and **test set from center 3**. Details are as follows:

Discovery set: 181 ESCC tissue samples and their corresponding 181 normal mucosae were collected from center 1 between 2016 and 2020. These samples were divided into three batches and underwent 400 MHz NMR detection. The reported VIP > 1 and p -value < 0.05 were used as screening criteria, and the consistently identified biomarker candidates constituted our initial potential biomarker pool. The expression changes (up- or down-regulation) of these metabolites were recorded based on their Fold change (FC) values and were presented in Supplementary Table 3, left panel.

Validation set: 54 ESCC patients were included in this study, whose matched ESCC tissues, normal mucosae, pre-operative and one-week post-operative serum and urine samples, as well as their clinical phenotype data were obtained from center 2 between 2021 and 2022, resulting in a total of 324 samples. 126 healthy subjects (including 87 serum and 39 urine samples) with no history of gastrointestinal problems were obtained as control groups.

1) We first performed higher-resolution 600 MHz NMR detection on the tissue samples. Specifically, we subdivided the samples for comparing normal, early-stage, and advanced-stage tissue samples in pairs. We used stricter screening criteria (including adjusted p -value, $p(|\text{corr}|)$, VIP value, machine learning RF, SVM algorithms) to obtain potential biomarkers for early-stage and overall ESCC. Given our study's emphasis on early detection, we primarily focused on metabolites already showing diagnostic efficacy in the early stage, and further screened for those with high FC and AUC values, which were recorded in Supplementary Table 3, middle panel.

2) Subsequently, metabolites from early-stage tissues underwent absolute quantification, including 544 metabolites across 13 classes detected via LC/MS-MS and 11 fatty acids detected via GC/MS-MS, covering most of the substances we identified in NMR. The expression trends of the metabolites that matched the MS and NMR results were also recorded in Supplementary Table 3, middle panel. To pinpoint the unique metabolic biomarkers in early ESCC tissues, we constructed an O2PLS model by integrating NMR and MS data matrices and identified the top 30 metabolite variables most associated between these two methods, documented in Supplementary Table 3, right panel. Combining the overall changes in key metabolic pathways in the tissue and considering them comprehensively, we identified the cancer-specific metabolic biomarkers in early ESCC tissues (Table 1).

3) We also collected pre- and one-week post-operative serum and urine samples from ESCC patients mentioned above and analyzed them using 600 MHz NMR spectroscopy. Specifically, we subdivided the samples for comparison: pre-operative, post-operative, and healthy control samples in pairs. On the one hand, we identified diagnostic biomarkers for serum and urine closely associated with the early-stage tissue (Table 2). On the other hand, we also observed the enriched metabolic pathways from significantly differentiated metabolites during the progression from pre- to post-operative to healthy states. This helps us to summarize the 'general direction' of metabolic changes in biofluids and verify whether they match the changes in tissue metabolism.

Test set: 18 early-stage ESCC serum and urine specimens each; 24 matched tissue and longitudinal biofluids from 4 different pathological types of esophageal tumors patients; 104 colorectal cancer (CRC) tissues and their 104 normal mucosae; as well as 42 control serum and 31 control urine

specimens, were collected from center 3 between 2021 and 2022. By comparing the metabolic profiles of ESCC with those of 4 other different pathological types of esophageal tumors and CRC (lower digestive tract tumors), we demonstrated the specificity of the ESCC metabolome atlas. We also conducted predictive analyses on 36 independent early-stage ESCC serum and urine samples to illustrate the sensitivity of the biofluid classifiers.

Updated Supplementary Figure 1: Flowchart and sample distribution in the discovery, validation and test sets for developing biofluid classifiers for ESCC early detection. A total of 560 participants with 1153 specimens were recruited from 3 multi-centers.

Measures taken:

In the Document "Article File",

1) "In the discovery set, our three previous studies consistently identified potential metabolic biomarkers in ESCC tissues using 400 MHz ¹H-NMR, including Acetate, Alanine, beta-Glucose, ... etc." in Line (L) 103-105 **has been revised to** "In the discovery set, our three previous studies consistently identified metabolic biomarker candidates that constituted our initial potential biomarker pool in ESCC tissues using 400 MHz ¹H-NMR, including Acetate, Alanine, beta-Glucose,

etc", which has been highlighted in red in the revised manuscript in **L 106-108**.

2) "Given our study's emphasis on early detection, we primarily focused on metabolites already showing diagnostic efficacy in the early stage, and further screening included selecting those with high fold change (FC) and areas under the curve (AUC) values, recorded in Supplementary Table 3, middle panel." **has been added to** the revised manuscript, highlighted in red in **L 123-126**.

3) "Liquid chromatography-tandem mass spectrometry (LC-MS/MS) and gas chromatography-tandem mass spectrometer (GC/MS-MS) were used to validate the NMR results of early ESCC tissues on the same batch. 600 biochemicals were quantified by multiple reaction monitoring (MRM) mode" in Lines (L) 123-125 **has been revised to** "Targeted quantitative MS were used to validate the NMR results of early ESCC tissues on the same batch. Over 500 biochemicals were quantified by multiple reaction monitoring (MRM) mode, including 544 metabolites across 13 classes detected via Liquid chromatography-tandem mass spectrometry (LC/MS-MS) and 11 fatty acids detected via gas chromatography-tandem mass spectrometer (GC/MS-MS), covering most of the substances we identified in NMR.", which has been highlighted in red in the revised manuscript in **L 132-136**.

4) "We screened these variables rigorously and meticulously (Supplementary Table 3) to identify the cancer-specific metabolic biomarkers in early ESCC tissues, including 2-hydroxybutyrate, 3-Hydroxybutyrate, 3-Hydroxypropionic acid, ... etc., shown in Table 1." in L 135-137 **has been revised to** "Combining the overall changes in key metabolic pathways in the tissue and considering them comprehensively, we identified the cancer-specific metabolic biomarkers in early ESCC tissues, including 2-hydroxybutyrate, 3-Hydroxybutyrate, 3-Hydroxypropionic acid, etc., shown in Table 1.", which has been highlighted in red in the revised manuscript in **L 147-150**.

5) "Investigations originating from tissue analysis are considered a more robust approach, offering explicit biochemical insights into disease pathogenesis and responses to stimuli³." **has been added to** the revised manuscript, highlighted in red in **L 395-396**.

6) "The initial stage of tumorigenesis occurs within the metabolic constraints of the native tissue. Therefore, understanding tissue-specific metabolic phenotypes is an important foundation for understanding cancer metabolism." in L 392-394 **has been deleted**.

7) In the Document "Supplementary Figures", on the first page, **Supplementary Figure 1** has been modified, and a figure legend **has been added to L 5-6**: "Flowchart and sample distribution in the discovery, validation and test sets for developing biofluid classifiers for ESCC early detection.", which has been highlighted in red in the revised manuscript.

2) If my assumption on the process is correct, why not use the first data set to both detect the markers and develop the algorithm? Then the test set would be larger. Right now, the test set is very small, with only 18 early stage cases.

Response: We appreciate the reviewer's question and valuable suggestions. Your assumption on the process is correct.

1) Regarding the **sample size**, the scarcity of clinical specimens for early-stage ESCC presents a challenge. Due to the lack of obvious clinical symptoms in the early stages, most of the patients we encounter in clinical work are already in the advanced stage. In addition, about 60% of ESCC patients receive neoadjuvant therapy initially. Therefore, the 18 treatment-naïve early-stage ESCC

cases already represent the annual number of cases in the largest hospital in the coastal area with high ESCC incidence in our country. According to a study in *Nature Communications*, considering sensitivity, reproducibility, detection limit, linearity and dynamic range, selectivity, identification, coverage, etc., the minimum optimal sample size for metabolomics and other omics studies is 16 cases, with an average power of at least 0.8, FDR of 0.05, and initial Cohen's d of 0.8 (*Nat Commun* 11, 3092, doi:10.1038/s41467-020-16937-8 (2020))⁴. We agree that a larger sample size may be preferable, but the current number of cases in our test set is sufficient to support subsequent analyses.

2) Regarding the **model development**, we used **discovery sets** to identify potential metabolic biomarkers, validated them using **validation sets**, and tested their efficacy in predicting early-stage ESCC using the **test sets**, as explained in detail in the previous response. These three datasets serve their individual roles, and overall, the model was robust, regardless of whether clinical samples from stage I-IV were included initially or only 18 pairs of early-stage ESCC samples were used later.

3) Let's take a closer look at the specific performance of these independent 18 pairs of early-stage serum and urines. When faced with the clinical challenge of identifying early-stage ESCC patients in the population, we used the Biomarker Analysis module of the MetaboAnalyst platform (<https://www.metaboanalyst.ca/MetaboAnalyst/upload/RocUploadView.xhtml>) to "blindly" test 30% of the 36 early-stage biofluids and their corresponding healthy control samples, using signatures from Table 2 for diagnostic predictions. For the sake of brevity, the specific results of these predictions were not fully presented in the previous manuscript, and we apologize for this. The prediction results for the early-stage biofluids in the **New Supplementary Table 13** have been added to the revised manuscript.

New Supplementary Table 13. Prediction probability values and outcomes of the blinded cases by the biofluid metabolic classifiers.

Blinded case	Probability scores	Predicted class
Serum – 1	0.85158	Early_ESCC
Serum – 2	0.95892	Early_ESCC
Serum – 3	0.95902	Early_ESCC
Serum – 4	0.89986	Early_ESCC
Serum – 5	0.95378	Early_ESCC
Serum – 6	0.97922	Early_ESCC
Urine – 1	0.57677	Early_ESCC
Urine – 2	0.90475	Early_ESCC
Urine – 3	0.9571	Early_ESCC
Urine – 4	0.79	Early_ESCC
Urine – 5	0.81501	Healthy control (prediction error)
Urine – 6	0.87018	Early_ESCC

4) We also conducted a trace-back analysis of one urine sample misclassified by our urine classifier. We found that the patient who provided the sample reported a mild sense of swallowing obstruction and sought timely medical attention. The individual denied having a history of smoking or drinking, and there was no post-operative pathological metastasis. We inferred that the incorrect classification

may be due to the early stages of the tumors, indicating that there is room for improvement in urine classification. However, the classification was correct when serum predictions were made for the same patient. At present, it is clear from the prediction results that the predictive accuracy of serum is slightly better than that of urine, consistent with the diagnostic efficacy described earlier. We plan to improve the classifier by expanding the sample size of the validation, so as to make our research outcomes more robust.

Measures taken:

1) We have added **Supplementary Table 13** to Document "Supplementary Tables" and marked it on **L 308** in Document "Article File".

2) In Document "Methods", "The cut-off value for the correlation coefficient (r) was determined based on the sample size of 44, which was calculated using PASS software (version 11.0, NCSS statistical software, USA). The sample size of 44 was chosen to achieve an FDR-corrected p -value of 0.05 and expected sensitivity and specificity of 0.80 to 0.90. Therefore, a $|p|_{\text{corr}} \geq 0.294$ indicates a significant correlation between two types of metabolites in the group. The p -value was obtained from the Mann-Whitney U-test of metabolite concentration, and the FC values represent the concentration ratio of each metabolite between the pair-wise groups." in L 186-193 **has been revised to** "Regarding the sample size estimation, we used the PASS (Power Analysis and Sample Size) software and relevant formulas to estimate the sample size based on the standard deviation, discrimination, test level, and test efficiency of metabolic biomarkers detection obtained in our previous work. When the sample size estimation parameters were set to specificity of $90\% \pm 10\%$, sensitivity of $80\% \pm 10\%$, significance level (α) = 0.05, confidence level ($1-\alpha$) of 0.95, and two-sided test type, the required sample size per group was estimated to be $n = 44$. We tried to make the sample size of multiple subgroups exceed this threshold in this project. The cut-off value for the correlation coefficient (r) was determined based on the sample size; a $|p|_{\text{corr}} \geq 0.294$ indicates a significant correlation between the two types of metabolites in the group. The p -value was obtained from the Mann-Whitney U-test of metabolite concentration, and the FC values represent the concentration ratio of each metabolite between the pair-wise groups. Moreover, given the limited sample size of early-stage ESCC patients, metabolomics studies can be conducted with as few as 16 cases per group. A study in Nature Communications showed that, considering sensitivity, reproducibility, detection limit, linearity and dynamic range, selectivity, identification, coverage, etc., the minimum optimal sample size for metabolomics and other omics studies was 16 cases, with an average power of at least 0.8, FDR of 0.05, and initial Cohen's d of 0.8⁴.", which has been highlighted in red in the revised manuscript in **L 210-227**.

3) Sample collection details are critical for metabolomics studies as incorrectly collected samples cause bias. The authors need to describe in this manuscript how each of the sample sets were acquire, and if any of the patients were under anesthesia or had prior knowledge of their surgery before blood collection.

Response: Thank you for your insightful feedback. We apologize for the inadequate description of our previous manuscript. We fully agree that sample collection details are critical for metabolomics studies. In fact, this was how we conducted our experiments. Before sample collection, we mainly refer to literatures (*Nat Protoc* 2, 2692-2703, doi:10.1038/nprot.2007.376 (2007); *Clin Chem* 64, 1158-1182, doi:10.1373/clinchem.2018.287045 (2018))^{5,6} to determine our protocol. We strictly

followed the same detailed protocols and used the same brand of reagents and consumables for sample collection. The collection of biofluid samples was performed with the patient's informed consent while in a conscious and cooperative state. There was no blood collection under anesthesia. Details are as follows:

1) **Tissue samples**, including tumor tissues and distant non-cancerous tissues (5 cm away from the edge of the tumor), were obtained under the guidance of an experienced pathologist without compromising the patients' pathological examinations. The collected tissues were rinsed with PBS to avoid contamination, as well as to remove excess water, and then quickly frozen in liquid nitrogen to arrest enzymatic or chemical reactions. Samples were subsequently stored at -80°C until metabolite extraction.

2) **Serum samples**: Fasting blood was drawn into additive-free vacuum blood collection tubes and attention was paid to avoid hemolysis. Samples were coagulated naturally for 30 minutes and then centrifuged at 3000 rpm at 4°C for 10 minutes. Following centrifugation, the serum supernatant was aliquoted into storage tubes, immediately frozen in liquid nitrogen, and then stored at -80°C until further analysis.

3) **Urine samples**: Patients were instructed to collect morning midstream urine. After mild centrifugation, the urine was aliquoted into storage tubes, promptly frozen in liquid nitrogen, and then stored at -80°C until further analysis.

Measures taken:

In Document "Methods.", **L 26-40 has been added** with the sentence: "We strictly followed the same detailed protocols and used the same brand of reagents and consumables for sample collection^{5,6}. 1) Tissue samples, including tumor tissues and distant non-cancerous tissues (5 cm away from the edge of the tumor), were obtained under the guidance of an experienced pathologist without compromising the patients' pathological examinations. The collected tissues were rinsed with PBS to avoid contamination, as well as to remove excess water, and then quickly frozen in liquid nitrogen to arrest enzymatic or chemical reactions. Samples were subsequently stored at -80°C until metabolite extraction. 2) Serum samples: Fasting blood was drawn into additive-free vacuum blood collection tubes and attention was paid to avoid hemolysis. Samples were coagulated naturally for 30 minutes and then centrifuged at 3000 rpm at 4°C for 10 minutes. Following centrifugation, the serum supernatant was aliquoted into storage tubes, immediately frozen in liquid nitrogen, and then stored at -80°C until further analysis. 3) Urine samples: Patients were instructed to collect morning midstream urine. After mild centrifugation, the urine was aliquoted into storage tubes, promptly frozen in liquid nitrogen, and then stored at -80°C until further analysis.

4) *The incidence of EC is quite low and therefore a screening test that uses healthy controls is unlikely to be translated because the false positive rate would be too high. The translatable test would actually be one that compares EC patients with at risk patients, such as those with Barrett's or even GERD or acid reflux. The authors should at a minimum acknowledge this situation in their limitations section.*

Response: Thank you so much for your incisive comment and valuable suggestion. We also recognize our limitation of the lack of ESCC pre-cancerous lesion control samples, which is indeed an important step for improving our model. In the process of revising the manuscript, we contacted

the person in charge of the "National Key R&D Program for Precision Medicine – Esophageal Cancer Special Cohort Study", the "Esophageal Cancer Screening and Early Diagnosis and Treatment Base in Nanao County, Shantou City", and Xuzhou Cancer Hospital (China Academy of Medical Sciences Cancer Hospital closely linked medical consortium hospital). They strongly support our future collection of high-risk population samples (including Low-Grade Intraepithelial Neoplasia (LGIN), High-Grade Intraepithelial Neoplasia (HGIN), Barrett's, GERD, acid reflux, etc.). We look forward to addressing this issue in our subsequent work.

Measures taken:

In Document "Article File", "Third, following the identification of tissue-specific biomarkers detectable in biofluids, we need to promptly validate them in high-risk populations (including low-grade intraepithelial neoplasia (LGIN), high-grade intraepithelial neoplasia (HGIN), gastroesophageal reflux disease (GERD), Barrett's, etc.) to refine the model further." **has been added to** the revised manuscript in **L 467-470**, which has been highlighted in red. We have also accelerated the collection of samples from high-risk populations, including those with pre-cancerous lesions.

5) The results here are much better than previous papers which tended to be in the 80-90% AUC range. The authors detected some pretty common metabolites here, how do those metabolites perform in the other papers? Why, for example are glutamate and asparagine such strong markers here, but not in other's studies. It doesn't seem plausible.

Response: Thank you so much for your helpful inquiry. We would like to provide supportive evidence using literature findings.

1) The majority of identified studies on ESCC metabolomics have employed non-targeted approaches. Here are two published quantitative studies. **For ESCC tissues**, as outlined in our Discussion section in L 403-407: "The expression trends of biomarkers and significant pathways in early ESCC tissues in our study were highly consistent with those detected in ESCC tissues by Zeper Abliz's team using **high-resolution MS imaging spatially resolved metabolomics**, including down-regulation of Glutamine and up-regulation of Glutamate, Proline, Uridine, Histidine, Short-chain fatty acids (FAs), Arginine, and Ornithine (*Proc Natl Acad Sci U S A* 116, 52-57, doi:10.1073/pnas.1808950116 (2019))⁷." **For ESCC serum studies**, we refer to a particular article where authors performed **targeted metabolomics** analysis on serum samples from 132 early-stage ESCC and 107 healthy controls, using multiple detection platforms and multiple screening (*Anal Chim Acta* 1220, 340038, doi:10.1016/j.aca.2022.340038 (2022))⁸. The results showed that amino acid metabolites like Asparagine, Glutamic acid, Isoleucine, Leucine, Phenylalanine, Threonine, and Valine showed similar trends to our NMR metabolomics results. Specifically, Asparagine and Glutamic acid were significantly up-regulated in early-stage ESCC serum, as shown in **Table 2 of this publication**:

Table 2

The information on the differential metabolites obtained from targeted metabolomics analysis in the discovery set.

Compound	log2FC	FC	VIP	Adjusted P value a	AUC
EFA(C14:0)	-0.919		1.081	0.001*	0.770
EFA(C18:3 γ)	-0.659		1.155	<0.001**	0.831
EFA(C20:2)	-2.353		1.565	<0.001**	0.920
FFA(C18:2)	-0.805		1.294	<0.001**	0.850
FFA(C18:3 α)	-0.667		1.321	<0.001**	0.869
FFA(C20:2)	-1.947		1.396	<0.001**	0.885
FFA(C20:4)	-0.650		1.169	<0.001**	0.803
FFA(C20:5)	-1.474		1.330	<0.001**	0.811
Threonine	0.711	1.64	1.481	<0.001**	0.913
Leucine	1.088	2.13	1.791	<0.001**	0.988
Valine	0.731	1.66	1.723	<0.001**	0.971
Isoleucine	1.373	2.59	1.869	<0.001**	0.995
Phenylalanine	0.673	1.59	1.299	<0.001**	0.850
Tryptophan	1.554		1.949	<0.001**	0.989
Serine	0.622		1.548	<0.001**	0.932
Lysine	-0.747		1.340	<0.001**	0.864
Glutamic acid	0.702	1.63	1.103	<0.001**	0.792
Asparagine	1.724	3.30	1.991	<0.001**	0.998
4-hydroxyproline	1.379		1.526	<0.001**	0.902
Acetyl-carnitine	1.590		1.660	<0.001**	0.960
Citrulline	-0.787		1.482	<0.001**	0.919
Adipic acid	1.023		1.680	<0.001**	0.999

2) The interconnectedness of various metabolic pathways within the body means an intricate network where a single change can affect the entire system. Concerning some common metabolites, their association may span across multiple diseases (*Nat Commun* 14, 604, doi:10.1038/s41467-023-36231-7 (2023); *Nat Med* 28, 2309-2320, doi:10.1038/s41591-022-01980-3 (2022))^{9,10}. One potential explanation for these common biomarkers could be that they reflect general frailty, low-grade systemic inflammation, and impaired immune response to various diseases. However, further research is needed to elucidate the underlying mechanisms. We hope our study can provide a valuable starting point for such research.

3) Regarding the practical application of metabolic biomarkers (e.g., in diagnostics), if diagnostic biomarker panels are composed of common metabolites, it may pose challenges for certain clinical applications requiring high disease specificity. However, from another perspective, lacking specificity might be advantageous when evaluating the risk of multiple diseases and overall health status based on a single measurement using a biomarker panel. This idea is also mentioned in an article: non-targeted metabolomics provides a diagnostic benefit six times higher than traditional screening methods (*JAMA Netw Open* 4, e2114155, doi:10.1001/jamanetworkopen.2021.14155 (2021))¹¹. This is consistent with the statement in our manuscript that "it could be used as an indicator to detect different digestive tract abnormalities" in L454 in the revised manuscript.

4) Among the metabolic features we've currently identified (Table 2), we encompass both common classic metabolites (such as glutamate, asparagine, etc.) and relatively niche but previously unreported specific molecules associated with ESCC (such as alpha-Ketoisovalerate, 3-Hydroxybutyrate, etc.). Although the current results show that any five serum or urine metabolic features can form an effective diagnostic model, it may be a better choice to combine 'classic

molecules' and 'specific molecules' to build a model. This modelling pattern is reflected in the three national invention patents we have authorized (CN116465920, CN116430049, CN116359272), which can be found on the China National Intellectual Property Administration website: <https://pss-system.cponline.cnipa.gov.cn/conventionalSearch>.

Measures taken:

In the Document "Article File",

1) "The identified metabolites encompass both common classic metabolites (such as glutamate, asparagine, etc.) and relatively niche but previously unreported specific molecules associated with ESCC (such as alpha-Ketoisovalerate, 3-Hydroxybutyrate, etc.)." **has been added** and highlighted in red in the revised manuscript in L 442-445.

2) "Combining 'classic molecules' and 'niche molecules' to build a model might be a better choice in clinical applications." **has been added** and highlighted in red in the revised manuscript in L 450-451.

6) Related to the point above, I appreciate that the authors included the data from Nightingale for comparison. It does seem that some of the metabolites that were identified here as biomarkers appeared in the Nightingale dataset. However, the odds ratio for that work is vastly different than what I can surmise here. For example, in the Nightingale data, the odds ratios are in the range of 0.9 to 1.1, they are more like 100 for some of the best performing metabolites in the current study, which seems like a disconnect. The authors need to explain why the serum metabolites measured in their study have low or no odds ratios in the Nightingale study, which was also an NMR based analysis.

Response: Thank you for your insightful observation. I will try to answer your question by explaining a few concepts.

1) When screening for clinical diagnostic biomarkers, we need to ensure that the expression levels of the metabolite can differentiate between the ESCC and non-ESCC control groups. Here, we use the concepts of **fold change (FC)** and **area under the curve (AUC)** values. For instance, let's assume that there is a metabolite A. The FC value describes the concentration difference of A between ESCC patients and the control group. Meanwhile, the AUC value describes the accuracy of A as a biomarker to distinguish ESCC patients and controls.

2) A *JAMA* article explained that **odds ratios (OR)** does not reflect the actual multiple relationship between the occurrence of outcomes between groups, but can reflect the strength of the association (*JAMA* 320, 84-85, doi:10.1001/jama.2018.6971 (2018))¹². The OR of risk factors that affect clinical outcome can be interpreted as whether people with risk factors are more or less likely to experience an outcome of interest than someone without risk factors. For example, suppose the metabolite A has an OR value of 1.2; in that case, individuals with elevated A levels are more likely to develop ESCC in the future than those without elevated A levels. Moreover, this article mentions that "Nor can the magnitude of the odds ratio from one study be compared with the magnitude of the odds ratio from another study, because different samples and different model specifications will have different arbitrary scaling factors." Hence, the above indicators cannot be straightforwardly converted or directly compared. In our manuscript, we avoid overstating their significance, and instead we present them as supplementary information that indirectly supported our findings.

3) However, the Nightingale dataset from the UK Biobank represents a prospective cohort that is closer to real-world data. The findings demonstrated an association between pre-diagnosis levels of LDL, glutamine, and 3-hydroxybutyrate and future incidence of esophageal malignancies, suggesting these metabolites carry more information. Considering their significant expression differences between ESCC cases and controls, it would be advantageous to integrate these molecules into our biomarker panel. Perhaps leveraging OR values could optimize the diagnostic efficacy of ESCC.

Minor:

7) *The text is too small in many figures and the description of what is in those figures is sometimes too limited.*

Response: Thank you for the kind suggestion. We apologize for not uploading the high-resolution version of the Supplementary Figures earlier. This time, we have attached the high-resolution version of all Figures and revised the legends to include enough information to understand the figures.

8) *Please provide more information on the targeted LC-MS data acquisition. Were the samples analyzed by the Asara lab, or were they run elsewhere? How many metabolites were targeted and how many were measured for each sample type? Similar questions about the GC-MS data acquisition.*

Response: Thank you for your helpful question. We are now providing more detailed information on the targeted MS data acquisition.

The targeted detection was performed by Biotree Biomedical Technology Co., Ltd. (<https://www.biotree.cn/>), which holds certifications, including BSL-2 accreditation for pathogenic microorganism experiments, ISO9001, and CMA (China Metrology Accreditation), and has previously published a series of case studies on the targeted metabolomics product (*Cell Host Microbe* 31, 1820-1836 e1810, doi:10.1016/j.chom.2023.09.010 (2023); *Microbiome* 10, 245, doi:10.1186/s40168-022-01437-2 (2022))^{13,14}.

Early-stage tissues underwent absolute quantification via MS analysis, including 544 metabolites across 13 classes detected through LC/MS-MS and 11 fatty acids via GC/MS-MS, covering most of the substances we identified in NMR. Please find the detailed information list of targeted metabolites at the end of this "Point-by-point response" document (**Page 37-53**). Specifically,

1) **Targeted LC-MS:** High metabolite coverage, including 544 metabolites across 13 classes. Each substance has a standard, and absolute quantification is performed using isotope internal standard correction. The MRM detection mode of the high-sensitivity triple quadrupole instrument achieves ultra-high sensitivity and cross-level detection with a wide linear range of 0.01-500 μM .

2) **Targeted GC-MS:** The Shimadzu QP2020NX platform mainly detects seven short-chain and four medium-chain fatty acids. Qualitative and quantitative analyses were also performed using internal standard method.

This study quantified early-stage tissues for 307 metabolites using the LC-MS platform and eight fatty acids using the GC-MS platform. Absolute concentrations were available in **Updated**

Supplementary Table 7.

Measures taken:

In the Document "Methods",

1) "The core protocol of MS analysis was described in the literature. For LC-MS Analysis, metabolic extracts were analyzed using an ACQUITY UPLC System (H-Class, Waters) equipped with a Waters Atlantis Premier BEH Z-HILIC Column." in L 133-135 **has been revised to** "The targeted detection was performed by Biotree Biomedical Technology Co., Ltd. The core protocol of MS analysis was described in the literature. For LC-MS Analysis, high metabolite coverage, including 544 metabolites across 13 classes, was detected. Each substance has a standard, and absolute quantification is performed using isotope internal standard correction. Metabolic extracts were analyzed using an ACQUITY UPLC System (H-Class, Waters) equipped with a Waters Atlantis Premier BEH Z-HILIC Column in MRM mode, which achieves ultra-high sensitivity and cross-level detection with a wide linear range of 0.01-500 μM .", which has been highlighted in red in the revised manuscript in **L 147-154**.

2) "GC-MS Analysis mainly detects seven short-chain and four medium-chain fatty acids. Qualitative and quantitative analyses were also performed using internal standard method.", which **has been added** and highlighted in red in the revised manuscript in **L161-163**.

3) We have updated Supplementary Table 7 to Document "Supplementary Tables".

Reviewer #4 (Remarks to the Author): expertise in esophageal cancer classification

This manuscript by Zhao et al. performed comprehensive metabolome profiling of 1,153 ESCC tissues, adjacent normal mucosae, pre- and post-operative sera and urines from 560 participants across three hospitals. The authors used different metabolomic platforms (including NMR and targeted MS). Data was analyzed by machine learning, logistic regression as well as WGCNA. They found that changes in alanine, aspartate and glutamate metabolism were prevalent throughout the progression of ESCC, from normal mucosae to early and to advanced cancers. The authors further developed NMR-based 16 serum and 10 urine metabolic signatures, which reflected the characteristic metabolic features of the ESCC tissues. Lastly, the authors showed that NMR-based classifiers incorporating any five serum or urine metabolite signatures can effectively diagnose and predict early ESCC.

Overall, this is a tour de force metabolic work and represents perhaps the largest scale metabolomic profiling in ESCC (if not any other cancer), therefore representing a huge effort and highly valuable resource for the research community. The data analysis was comprehensive and justified (with some weaknesses listed below) and the data interpretation was generally appropriate. However, I do have a list of questions/concerns which need to be improved in order for the audience to digest and understand better the work.

Response: Thank you so much for your positive comments and recognition of our manuscript. We appreciate your summary and understand the importance of addressing your list of questions and concerns. We have carefully considered each point and made the necessary revisions to improve the clarity and readability of our work for the audience. Kindly find our point-to-point response below.

1) I did not find "Data availability" section in the main article. All of the raw and processed metabolomic data would need to be publicly available, which is a standard for Nature journals.

Response: Thank you for your kind reminder. We have carefully read the journal policy (<https://www.nature.com/nature-portfolio/editorial-policies/reporting-standards#availability-of-data>) and understand that sequencing or protein MS technologies are the types of data that need to be submitted. The metabolomics data involved in our study might not fall within this category. In fact, at the time of our initial submission, we sought guidance on this matter from the esteemed editor, Dr. Kathryn McGinnis.

Meanwhile, we have searched the current mainstream metabolomics data platforms, such as Metabolomics Workbench or MetaboLights, and found that there is currently no publicly available raw data for ESCC metabolomics. In the spirit of collaborative research and advancing knowledge, we would like to share the critical matched data from our study earnestly, including raw and processed data (on the Metabolomics Workbench database) and resonance assignments of metabolites (provided in the SI). Additionally, due to intellectual property agreements with different hospitals, the 400 MHz data from the previous study and the smaller subset of the Test set are not suitable for public release. However, they can be obtained from the corresponding author upon reasonable request.

The data can be accessed directly through the Project DOI: <http://dx.doi.org/10.21228/M87426>, which will be released publicly after a year. But we would like to provide you with the database account in advance for your review: <http://dev.metabolomicsworkbench.org:22222/data/DRCCMetadata.php?Mode=Project&ProjectID=PR001876>; Username: Linyan; Password: Linyan@1115.

Measures taken:

In the Document "Article File", "**Data availability:** These data are available at the NIH Common Fund's National Metabolomics Data Repository (NMDR) website, the Metabolomics Workbench, <https://www.metabolomicsworkbench.org>, where it has been assigned a Project ID PR001876. The data can be accessed directly through the Project DOI: <http://dx.doi.org/10.21228/M87426>." **has been added to** the revised manuscript, highlighted in red in **Line (L) 482-485**.

2) Early ESCC biomarker is a strong focus for this paper. However, only 18 early ESCC patients were profiled in the Test set from center 3 (Supplementary Table 1). Is this number actually sufficiently powered for the discovery? The authors need to show statistical analysis, particularly power analysis to demonstrate that with this low number, what fold change will be needed for the discovery and at what p value.

Response: Thank you for your professional questions. A similar point was also raised by Referee #2 in point 2. We'll address your questions regarding sample size, power analysis, and the predictive performance achieved with these independent 18 pairs of sera and urines.

1) Regarding the **sample size**, the scarcity of clinical specimens for early-stage ESCC presents a challenge. Due to the lack of obvious clinical symptoms in the early stages, most of the patients we encounter in clinical work are already in the advanced stage. In addition, about 60% of ESCC

patients receive neoadjuvant therapy initially. Therefore, the 18 treatment-naïve early-stage ESCC cases already represent the annual number of cases in the largest hospital in the coastal area with high ESCC incidence in our country. According to a study in *Nature Communications*, considering sensitivity, reproducibility, detection limit, linearity and dynamic range, selectivity, identification, coverage, etc., the minimum optimal sample size for metabolomics and other omics studies is 16 cases, with an average power of at least 0.8, FDR of 0.05, and initial Cohen's d of 0.8 (*Nat Commun* 11, 3092, doi:10.1038/s41467-020-16937-8 (2020))⁴. We agree that a larger sample size may be preferable, but the current number of cases in our test set is sufficient to support subsequent analyses.

2) Regarding the **model development**, we used **discovery sets** to identify potential metabolic biomarkers, then validated them using **validation sets**, and tested their efficacy in predicting early-stage ESCC using the **test sets**, as will be explained in detail in the next response. In terms of the datasets, both the discovery and validation sets contained samples from clinical stages I-IV (including early-stage samples), while the test set contained 18 pairs of early-stage ESCC samples. These three datasets serve their individual roles, and overall, the model was robust, regardless of whether clinical samples from stage I-IV were included initially or whether only 18 pairs of early-stage ESCC samples were used later.

3) Let's take a closer look at the specific performance of these independent 18 pairs of early-stage serum and urines. When faced with the clinical challenge of identifying early-stage ESCC patients in the population, we used the Biomarker Analysis module of the MetaboAnalyst platform (<https://www.metaboanalyst.ca/MetaboAnalyst/upload/RocUploadView.xhtml>) to "blindly" test 30% of the 36 early-stage biofluids and their corresponding healthy control samples, using signatures from Table 2 for diagnostic predictions. For the sake of brevity, the specific results of these predictions were not fully presented in the previous manuscript, and we apologize for this. The prediction results for the early-stage biofluids in the **New Supplementary Table 13** have been added to the revised manuscript.

New Supplementary Table 13. Prediction probability values and outcomes of the blinded cases by the biofluid metabolic classifiers.

Blinded case	Probability scores	Predicted class
Serum – 1	0.85158	Early_ESCC
Serum – 2	0.95892	Early_ESCC
Serum – 3	0.95902	Early_ESCC
Serum – 4	0.89986	Early_ESCC
Serum – 5	0.95378	Early_ESCC
Serum – 6	0.97922	Early_ESCC
Urine – 1	0.57677	Early_ESCC
Urine – 2	0.90475	Early_ESCC
Urine – 3	0.9571	Early_ESCC
Urine – 4	0.79	Early_ESCC
Urine – 5	0.81501	Healthy control (prediction error)
Urine – 6	0.87018	Early_ESCC

4) We also conducted a trace-back analysis of one urine sample misclassified by our urine classifier. We found that the patient who provided the sample reported a mild sense of swallowing obstruction and sought timely medical attention. The individual denied having a history of smoking or drinking, and there was no post-operative pathological metastasis. We inferred that the incorrect classification may be due to the early stages of the tumors, indicating that there is room for improvement in urine classification. However, the classification was correct when serum predictions were made for the same patient. At present, it is clear from the prediction results that the predictive accuracy of serum is slightly better than that of urine, consistent with the diagnostic efficacy described earlier. We plan to improve the classifier by expanding the sample size of the validation, so as to make our research outcomes more robust.

Measures taken:

1) We have added **Supplementary Table 13** to Document "Supplementary Tables" and marked it on **L 308** in Document "Article File".

2) In Document "Methods", "The cut-off value for the correlation coefficient (r) was determined based on the sample size of 44, which was calculated using PASS software (version 11.0, NCSS statistical software, USA). The sample size of 44 was chosen to achieve an FDR-corrected p -value of 0.05 and expected sensitivity and specificity of 0.80 to 0.90. Therefore, a $p|corr| \geq 0.294$ indicates a significant correlation between two types of metabolites in the group. The p -value was obtained from the Mann-Whitney U-test of metabolite concentration, and the FC values represent the concentration ratio of each metabolite between the pair-wise groups." in L 186-193 **has been revised to** "Regarding the sample size estimation, we used the PASS (Power Analysis and Sample Size) software and relevant formulas to estimate the sample size based on the standard deviation, discrimination, test level, and test efficiency of metabolic biomarkers detection obtained in our previous work. When the sample size estimation parameters were set to specificity of 90%±10%, sensitivity of 80%±10%, significance level (α) = 0.05, confidence level (1- α) of 0.95, and two-sided test type, the required sample size per group was estimated to be $n = 44$. We tried to make the sample size of multiple subgroups exceed this threshold in this project. The cut-off value for the correlation coefficient (r) was determined based on the sample size; a $p|corr| \geq 0.294$ indicates a significant correlation between the two types of metabolites in the group. The p -value was obtained from the Mann-Whitney U-test of metabolite concentration, and the FC values represented the concentration ratio of each metabolite between the pair-wise groups. Moreover, due to the limited sample size of early-stage ESCC patients, metabolomics studies can be conducted with as few as 16 cases per group, according to a previous study in *Nature Communications*, which showed that considering sensitivity, reproducibility, detection limit, linearity and dynamic range, selectivity, identification, coverage, etc., the minimum optimal sample size for metabolomics and other omics studies was 16 cases, with an average power of at least 0.8, FDR of 0.05, and initial Cohen's d of 0.8⁴.", which has been highlighted in red in the revised manuscript in **L 210-227**.

3) Relatedly, throughout the manuscript, no sample number was clearly given in the result section, which makes it very difficult to review because there are multiple cohorts, multiple stages of cancers, multiple types of samples, as well as multiple centers. Thus, the authors need to explicitly describe sample numbers and types for each experiment and each comparison in the result section.

Response: Thank you for your careful review and good suggestion. A similar point was also raised by Referee #2 in point 1. We apologize for not clearly describing the statistical analysis procedure in our previous manuscript. In the revised version, we have provided a more detailed explanation of the workflow and have updated Supplementary Figure 1 accordingly, which we believe could illustrate the process more clearly and intuitively. As you can see in the updated Supplementary Fig. 1, our study consisted of three phases: **discovery set from center 1, matched validation set from center 2, and test set from center 3**. Details are as follows:

Discovery set: 181 ESCC tissue samples and their corresponding 181 normal mucosae were collected from center 1 between 2016 and 2020. These samples were divided into three batches and underwent 400 MHz NMR detection. The reported VIP > 1 and p -value < 0.05 were used as screening criteria, and the consistently identified biomarker candidates constituted our initial potential biomarker pool. The expression changes (up- or down-regulation) of these metabolites were recorded based on their Fold change (FC) values and were presented in Supplementary Table 3, left panel.

Validation set: 54 ESCC patients were included in this study, whose matched ESCC tissues, normal mucosae, pre-operative and one-week post-operative serum and urine samples, as well as their clinical phenotype data were obtained from center 2 between 2021 and 2022, resulting in a total of 324 samples. 126 healthy subjects (including 87 serum and 39 urine samples) with no history of gastrointestinal problems were obtained as control groups.

1) We first performed higher-resolution 600 MHz NMR detection on the tissue samples. Specifically, we subdivided the samples for comparing normal, early-stage, and advanced-stage tissue samples in pairs. We used stricter screening criteria (including adjusted p -value, $p(|\text{corr}|)$, VIP value, machine learning RF, SVM algorithms) to obtain potential biomarkers for early-stage and overall ESCC. Given our study's emphasis on early detection, we primarily focused on metabolites already showing diagnostic efficacy in the early stage, and further screened for those with high FC and AUC values, which were recorded in Supplementary Table 3, middle panel.

2) Subsequently, metabolites from early-stage tissues underwent absolute quantification, including 544 metabolites across 13 classes detected via LC/MS-MS and 11 fatty acids detected via GC/MS-MS, covering most of the substances we identified in NMR. The expression trends of the metabolites that matched the MS and NMR results were also recorded in Supplementary Table 3, middle panel. To pinpoint the unique metabolic biomarkers in early ESCC tissues, we constructed an O2PLS model by integrating NMR and MS data matrices and identified the top 30 metabolite variables most associated between these two methods, documented in Supplementary Table 3, right panel. Combining the overall changes in key metabolic pathways in the tissue and considering them comprehensively, we identified the cancer-specific metabolic biomarkers in early ESCC tissues (Table 1).

3) We also collected pre- and one-week post-operative serum and urine samples from ESCC patients mentioned above and analyzed them using 600 MHz NMR spectroscopy. Specifically, we subdivided the samples for comparison: pre-operative, post-operative, and healthy control samples in pairs. On the one hand, we identified diagnostic biomarkers for serum and urine closely associated with the early-stage tissue (Table 2). On the other hand, we also observed the enriched metabolic pathways from significantly differentiated metabolites during the progression from pre- to post-

operative to healthy states. This helps us to summarize the 'general direction' of metabolic changes in biofluids and verify whether they match the changes in tissue metabolism.

Test set: 18 early-stage ESCC serum and urine specimens each; 24 matched tissue and longitudinal biofluids from 4 different pathological types of esophageal tumors patients; 104 colorectal cancer (CRC) tissues and their 104 normal mucosae; as well as 42 control serum and 31 control urine specimens, were collected from center 3 between 2021 and 2022. By comparing the metabolic profiles of ESCC with those of 4 other different pathological types of esophageal tumors and CRC (lower digestive tract tumors), we demonstrated the specificity of the ESCC metabolome atlas. We also conducted predictive analyses on 36 independent early-stage ESCC serum and urine samples to illustrate the sensitivity of the biofluid classifiers.

Updated Supplementary Figure 1: Flowchart and sample distribution in the discovery, validation and test sets for developing biofluid classifiers for ESCC early detection. A total of 560 participants with 1153 specimens were recruited from 3 multi-centers.

Measures taken:

In the Document "Article File",

- 1) "In the discovery set, our three previous studies consistently identified potential metabolic biomarkers in ESCC tissues using 400 MHz ¹H-NMR, including Acetate, Alanine, beta-Glucose, ... etc." in Lines (L) 103-105 **has been revised to** "In the discovery set, our three previous studies consistently identified metabolic biomarker candidates that constituted our initial potential biomarker pool in ESCC tissues using 400 MHz ¹H-NMR, including Acetate, Alanine, beta-Glucose, etc", which has been highlighted in red in the revised manuscript in **L 106-108**.
- 2) "Given our study's emphasis on early detection, we primarily focused on metabolites already showing diagnostic efficacy in the early stage, and further screening included selecting those with high fold change (FC) and areas under the curve (AUC) values, recorded in Supplementary Table 3, middle panel." **has been added to** the revised manuscript, highlighted in red in **L 123-126**.
- 3) "Liquid chromatography-tandem mass spectrometry (LC-MS/MS) and gas chromatography-tandem mass spectrometer (GC/MS-MS) were used to validate the NMR results of early ESCC tissues on the same batch. 600 biochemicals were quantified by multiple reaction monitoring (MRM) mode" in L 123-125 **has been revised to** "Targeted quantitative MS were used to validate the NMR results of early ESCC tissues on the same batch. Over 500 biochemicals were quantified by multiple reaction monitoring (MRM) mode, including 544 metabolites across 13 classes detected via Liquid chromatography-tandem mass spectrometry (LC/MS-MS) and 11 fatty acids detected via gas chromatography-tandem mass spectrometer (GC/MS-MS), covering most of the substances we identified in NMR.", which has been highlighted in red in the revised manuscript in **L 132-136**.
- 4) "We screened these variables rigorously and meticulously (Supplementary Table 3) to identify the cancer-specific metabolic biomarkers in early ESCC tissues, including 2-hydroxybutyrate, 3-Hydroxybutyrate, 3-Hydroxypropionic acid, ... etc., shown in Table 1." in L 135-137 **has been revised to** "Combining the overall changes in key metabolic pathways in the tissue and considering them comprehensively, we identified the cancer-specific metabolic biomarkers in early ESCC tissues, including 2-hydroxybutyrate, 3-Hydroxybutyrate, 3-Hydroxypropionic acid, etc., shown in Table 1.", which has been highlighted in red in the revised manuscript in **L 147-150**.
- 5) "Investigations originating from tissue analysis are considered a more robust approach, offering explicit biochemical insights into disease pathogenesis and responses to stimuli³." **has been added to** the revised manuscript, highlighted in red in **L 395-396**.
- 6) "The initial stage of tumorigenesis occurs within the metabolic constraints of the native tissue. Therefore, understanding tissue-specific metabolic phenotypes is an important foundation for understanding cancer metabolism." in L 392-394 **has been deleted**.
- 7) In the Document "Supplementary Figures", on the first page, **Supplementary Figure 1** has been modified, and a figure legend has been added to **L 5-6**: "Flowchart and sample distribution in the discovery, validation and test sets for developing biofluid classifiers for ESCC early detection.", which has been highlighted in red in the revised manuscript.

4) What does it mean by "External permutation tests validated that the models were suitable for data analysis (Fig. 2D-F)?" it is unclear what do the dots represent and what are Q or R values.

Response: Thank you for your valuable suggestion, and we apologize for any previous lack of clarity. The Permutation Plot helps assess the risk that the current model is spurious, i.e., the model fits the training set well but does not predict Y well for new observations. The idea of this validation

is to compare the goodness of fit (R^2 and Q^2) of the original model with the goodness of fit of several models based on data where the order of the Y-observations has been randomly permuted, while the X-matrix has been kept intact.

Taking Fig. 2D as an example: the x-axis represents the permutation retention rate of the permutation test, and the dots in the upper right corner represent the R^2 and Q^2 values of the original model when the permutation retention rate is 1. R^2 measures the goodness of fit, while Q^2 measures the predictive ability of the model. Light blue dots represent the R^2 values obtained from the permutation test, while dark blue dots represent the Q^2 values obtained from the permutation test. The two dashed lines represent the regression lines of R^2 and Q^2 , respectively. It can be observed from the regression lines that the R^2 and Q^2 values obtained through the permutation test are smaller than the model's original values, indicating that the model is robust. The smaller the intercepts of the R^2 and Q^2 fitting regression lines, the better the model fit.

Measures taken:

1) In the Document "Article File", "The x-axis represents the permutation retention rate of the permutation test, and the dots in the upper right corner represent the R^2 and Q^2 values of the original model when the permutation retention rate is 1. R^2 measures the goodness of fit, while Q^2 measures the predictive ability of the model. Light blue dots represent the R^2 values obtained from the permutation test, while dark blue dots represent the Q^2 values obtained from the permutation test. The two dashed lines represent the regression lines of R^2 and Q^2 , respectively." **has been added to** the revised manuscript, highlighted in red in L 155-160.

2) In the Document "Methods", "The Permutation Plot helps assess the risk that the current model is spurious, i.e., the model fits the training set well but does not predict Y well for new observations. The idea of this validation is to compare the goodness of fit (R^2 and Q^2) of the original model with the goodness of fit of several models based on data where the order of the Y-observations has been randomly permuted, while the X-matrix has been kept intact." **has been added to** the revised manuscript, highlighted in red in L 201-206.

5) *Fig. 2J does not show p value, which is important. This can be plotted along X axis.*

Response: Thank you for your helpful suggestion. As requested, we have revised the Multiple Volcano plot accordingly: Metabolites with $p < 0.05$ are visualized as solid circles on the plot, while those with $p > 0.05$ are not displayed. For example, the p -value of Asparagine between Early and

Advanced stages was > 0.05 , hence it remains invisible in the right panel, shown in Fig. 2J. A \log_{10} transformation was applied to the p -values of each significantly differential metabolite to visualize their significance levels.

Fig. 2J. Multiple Volcano plot based on the same batch of samples.

Measures taken:

In the Document "Article File", **Fig. 2J has been modified**, and a figure legend **has been added to L 164-166**: "(J) Multiple Volcano plot based on the same batch of samples. Metabolites with $p < 0.05$ are visualized as solid circles on the plot, while those with $p > 0.05$ are not displayed. A \log_{10} transformation was applied to the p -values of each significantly differential metabolite to visualize their significance levels."

6) Fig.2G-I, the color and size of each bubble correspond to its p -value and pathway impact value, respectively. However, there is no color scale or size scale.

Response: Thank you for your helpful suggestion. As requested, we have incorporated color and size scales into the metabolic pathway maps. Specifically, the metabolic pathway enrichment results were generated using the Pathway Analysis module of MetaboAnalyst (<https://www.metaboanalyst.ca/>). The results are presented in a table containing the p -value and Impact value of each pathway. By default, the p -value was converted to $-\log_{10}$ and combined with the impact value, which is visualized by bubbles of different colors and sizes, and the scale is no longer displayed. In general, bubbles on the right side of the map have higher weights, while bubbles at the top have smaller p -values. Here is an example of the pathway results of Fig. 2G:

ESCC tumors vs. normal mucosa			
Metabolic pathways	p -value	$-\log_{10}(p)$	Impact
Alanine, aspartate and glutamate metabolism	1.36E-46	45.868	0.3133
Aminoacyl-tRNA biosynthesis	9.04E-43	42.044	0
Citrate cycle (TCA cycle)	6.34E-38	37.198	0.16525
Glutathione metabolism	4.24E-35	34.373	0.36435
Glyoxylate and dicarboxylate metabolism	3.10E-33	32.509	0.10582
Arginine biosynthesis	1.80E-32	31.745	0.19289

Histidine metabolism	3.45E-30	29.462	0.22131
Arginine and proline metabolism	2.26E-27	26.647	0.23378
Glycolysis / Gluconeogenesis	4.43E-26	25.354	0.13055
D-Glutamine and D-glutamate metabolism	5.70E-26	25.244	0.5
Nitrogen metabolism	5.70E-26	25.244	0
Glycine, serine and threonine metabolism	7.93E-25	24.1	0.3387
Propanoate metabolism	1.34E-24	23.872	0
Porphyrin and chlorophyll metabolism	1.59E-22	21.8	0
Butanoate metabolism	5.10E-22	21.292	0
Pyruvate metabolism	5.81E-22	21.236	0.29859
Tyrosine metabolism	1.03E-20	19.988	0.16435
Galactose metabolism	1.00E-19	18.999	0.02924
Primary bile acid biosynthesis	3.40E-18	17.469	0.01516
beta-Alanine metabolism	4.22E-18	17.374	0
Fructose and mannose metabolism	1.26E-17	16.898	0
Amino sugar and nucleotide sugar metabolism	1.26E-17	16.898	0
Valine, leucine and isoleucine biosynthesis	1.45E-16	15.837	0
Valine, leucine and isoleucine degradation	1.02E-15	14.99	0
Pyrimidine metabolism	4.31E-14	13.365	0
Taurine and hypotaurine metabolism	7.96E-14	13.099	0.42857
Cysteine and methionine metabolism	2.32E-13	12.635	0
Purine metabolism	1.68E-12	11.774	0.02305
Fatty acid biosynthesis	3.83E-12	11.417	0.01473
Fatty acid elongation	3.83E-12	11.417	0
Fatty acid degradation	3.83E-12	11.417	0
Biosynthesis of unsaturated fatty acids	3.83E-12	11.417	0
Phenylalanine, tyrosine and tryptophan biosynthesis	3.51E-11	10.455	1
Phenylalanine metabolism	3.51E-11	10.455	0.35714
Synthesis and degradation of ketone bodies	9.46E-11	10.024	0
Pantothenate and CoA biosynthesis	1.28E-10	9.8915	0
Glycerophospholipid metabolism	1.71E-09	8.7661	0.03519
Ubiquinone and other terpenoid-quinone biosynthesis	3.03E-09	8.518	0
Inositol phosphate metabolism	7.58E-07	6.1202	0.12939
Phosphatidylinositol signaling system	7.58E-07	6.1202	0.03736
Ascorbate and aldarate metabolism	7.58E-07	6.1202	0
Selenocompound metabolism	1.21E-05	4.9178	0
Riboflavin metabolism	1.33E-05	4.8777	0.5
Lysine degradation	2.13E-03	2.6713	0
Biotin metabolism	2.13E-03	2.6713	0
Glycerolipid metabolism	8.07E-03	2.0933	0.23676

Based on the table, we manually added the Scale visualization to each pathway analysis result. This makes it easier for readers to understand:

g**Measures taken:**

In the Document "Article File", **Fig. 2G-I, Fig. 4D-F, Fig. 4J-L** have been modified, and a figure legend "In general, bubbles on the right side of the map have higher weights, while bubbles at the top have smaller p -values." has been added to L 162-163.

7) Any overlapped metabolites between early and late ESCCs? Those shared could be important ones to follow up.

Response: Thank you for your valuable suggestions, which have provided new insights for our analysis. We have analyzed the biomarkers of late-stage ESCC and updated the results in the right panel of Supplementary Table 6: Advanced ESCC tissue vs. Normal mucosa:

Part of the Supplementary Table 6. Statistical analysis of the differential metabolites between the early- and late-stage ESCC tissues and normal mucosa groups.

Metabolites	Early ESCC vs. Normal tissue				Advanced ESCC vs. Normal tissue			
	VIP	adjusted P	p(corr)	FC	VIP	adjusted P	p(corr)	FC
2-hydroxybutyrate	2.645	1.71E-03	0.628	1.658	3.437	6.57E-12	0.686	2.347
3-Hydroxybutyrate	1.193	1.90E-02	0.683	1.315	1.023	6.50E-09	0.662	1.212
3-Hydroxypropionic acid	1.094	3.33E-03	0.611	0.79	1.027	9.84E-09	0.788	0.765
3-Methylhistidine	2.473	3.60E-05	0.599	0.145	2.251	9.57E-14	0.675	0.148
Acetate	2.871	2.18E-04	0.614	1.565	2.594	7.55E-11	0.577	1.739
Acetoin	1.504	1.82E-01	0.666	1.256	1.156	5.11E-02	0.623	1.106
Adenine	1.071	1.76E-02	0.775	1.502	0.936	2.94E-04	0.774	1.395
ADP	0.884	4.89E-01	0.048	0.799	1.088	2.08E-03	0.440	0.595
Alanine	4.233	4.91E-02	0.365	0.826	4.543	2.79E-05	0.576	0.793
alpha-Glucose	1.795	2.18E-04	0.334	0.451	1.872	3.27E-13	0.682	0.284
alpha-Ketoisovalerate	0.993	3.04E-01	0.109	0.867	0.776	3.74E-01	0.277	0.917
AMP	0.819	1.46E-01	0.082	0.546	0.639	1.34E-02	0.246	0.581
Arginine	1.678	5.33E-04	0.746	1.629	2.162	4.28E-13	0.777	2.179
Asparagine	2.882	3.60E-05	0.855	3.366	2.780	1.48E-13	0.852	3.400
Aspartate	1.145	6.40E-01	0.532	0.932	1.154	1.41E-02	0.755	0.918
beta-Glucose	2.238	1.15E-04	0.548	0.77	2.751	3.36E-13	0.849	0.384
Bile acid	0.677	2.38E-03	0.745	1.523	0.624	9.65E-07	0.759	1.441
Butyrate	0.91	6.70E-03	0.515	1.219	0.824	7.82E-05	0.530	1.201
Choline	1.618	4.10E-03	0.482	1.153	1.436	4.40E-03	0.371	1.182
Citrate	1.902	4.64E-01	0.743	0.962	1.707	4.19E-02	0.752	0.941
Creatine	7.098	3.84E-02	0.365	0.792	7.184	1.01E-08	0.667	0.723
Dimethylamine	0.654	7.19E-02	0.375	1.119	0.990	1.76E-03	0.471	1.155
Ethanol	9.108	2.39E-04	0.28	0.572	8.344	5.53E-13	0.574	0.548
Flavin mononucleotide	1.27	2.05E-02	0.637	1.166	1.071	2.46E-03	0.645	1.152
Formate	0.195	5.71E-02	0.447	1.377	0.179	2.24E-03	0.462	1.548
Fumarate	0.979	3.03E-03	0.021	0.578	1.146	3.75E-10	0.751	0.503
Glutamate	3.255	5.21E-05	0.772	1.502	3.118	8.78E-13	0.770	1.519
Glutamine	2.748	3.03E-03	0.367	0.805	3.184	1.65E-10	0.746	0.683
Glutathione	1.745	3.62E-05	0.84	3.169	1.676	9.57E-14	0.862	3.154
Glycerol	1.461	2.05E-01	0.517	1.1	1.882	3.32E-03	0.535	1.178
Glycerophosphorylcholine	1.559	3.36E-01	0.732	1.403	1.245	7.49E-02	0.168	0.969
Glycine	5.096	1.76E-02	0.485	1.251	5.029	7.02E-09	0.502	1.412
Histidine	1.606	3.62E-05	0.824	1.535	1.550	3.65E-12	0.784	1.544
Hypoxanthine	1.33	6.40E-01	0.476	1.155	1.330	2.05E-01	0.560	1.183
Inosine	3.044	7.65E-02	0.69	1.35	2.375	3.01E-01	0.681	1.097
Isobutyrate	1.051	1.43E-03	0.756	1.438	0.957	6.82E-07	0.709	1.480
Isocitric acid	1.269	6.72E-04	0.131	0.781	1.311	1.31E-09	0.748	0.740
Isoleucine	1.692	5.46E-05	0.718	1.442	1.707	1.47E-12	0.759	1.522
Itaconic acid	0.878	8.60E-01	0.681	1.016	0.818	6.28E-01	0.678	1.037
Lactate	10.332	7.45E-01	0.63	0.984	11.368	8.38E-02	0.625	1.098
Leucine	2.376	1.98E-03	0.617	1.318	2.642	4.54E-10	0.669	1.403
Lipid	2.266	9.31E-01	0.158	0.625	2.416	1.29E-10	0.681	0.520
Low Density Lipoprotein	0.564	5.39E-01	0.354	1.013	0.470	5.46E-01	0.434	1.002
Lysine	1.043	2.21E-02	0.673	1.125	0.951	6.95E-04	0.519	1.111
Malate	1.845	1.98E-03	0.716	1.244	1.370	6.20E-04	0.650	1.122
Malonate	0.679	5.55E-01	0.67	1.022	0.601	2.11E-01	0.611	1.051

Methylamine	1.215	3.62E-05	0.761	1.603	1.121	8.70E-13	0.744	1.596
Methylmalonate	0.871	4.80E-01	0.203	0.967	0.880	2.51E-01	0.628	0.973
myo-Inositol	3.016	1.62E-01	0.377	0.836	3.219	5.57E-08	0.640	0.722
N-Acetyl-aspartate	0.947	8.43E-01	0.379	0.995	0.889	5.78E-01	0.520	0.995
NADP+	1.496	2.92E-01	0.069	0.381	0.342	5.97E-01	0.464	1.064
NADPH	0.542	6.79E-05	0.732	5.441	0.468	1.94E-12	0.702	4.157
Ornithine	0.762	2.62E-02	0.532	1.101	0.782	1.78E-02	0.565	1.109
Palmitic acid	1.684	3.33E-04	0.742	1.358	1.550	1.48E-08	0.761	1.303
Phenylalanine	1.314	3.03E-03	0.826	1.429	1.296	4.08E-09	0.865	1.521
Phosphocholine	3.297	2.60E-03	0.469	1.57	3.393	5.42E-09	0.455	1.604
Proline	1.576	1.98E-03	0.738	1.211	1.485	1.36E-05	0.732	1.163
Propionate	2.578	1.84E-04	0.846	1.321	2.517	1.11E-07	0.752	1.314
Pyruvate	2.318	1.57E-03	0.618	1.281	2.205	1.89E-10	0.655	1.351
Sarcosine	1.533	8.49E-05	0.433	0.549	1.459	4.28E-13	0.785	0.498
Succinate	1.657	4.98E-01	0.185	1.137	1.353	9.07E-01	0.403	1.040
Taurine	7.711	2.62E-04	0.536	0.71	6.853	2.29E-10	0.814	0.720
Theophylline	1.278	1.69E-04	0.884	1.385	1.200	1.79E-12	0.858	1.511
Threonine	1.191	3.33E-02	0.616	1.135	1.004	1.05E-01	0.608	1.071
Trigonelline	1.043	4.04E-01	0.463	1.075	0.757	2.94E-01	0.460	0.981
Trimethylamine	0.639	4.89E-01	0.541	1.056	0.423	3.00E-01	0.403	1.057
Trimethylamine N-oxide	2.045	8.15E-03	0.582	0.894	1.935	2.86E-05	0.637	0.866
Tryptophan	0.818	3.62E-05	0.888	3.771	0.816	2.96E-13	0.893	3.939
Tyrosine	1.473	1.22E-02	0.824	1.344	1.381	3.67E-08	0.864	1.380
Uracil	0.947	3.60E-05	0.91	3.687	0.989	1.14E-13	0.913	3.849
Uridine	1.203	7.36E-01	0.857	1.576	1.078	1.05E-05	0.858	1.426
Uridine diphosphate glucose	0.492	2.17E-03	0.686	1.264	0.399	5.01E-04	0.579	1.293
Valine	1.7	5.03E-03	0.73	1.329	1.881	1.69E-08	0.675	1.372
Very Low Density Lipoprotein	1.395	3.69E-01	0.628	1.044	1.865	6.47E-06	0.711	1.211

Abbreviations: **VIP:** Variable Importance in the Projection; **FC:** Fold Change; **AUC:** Areas Under the Curve.

We then identified the potential biomarkers of early and late-stage ESCCs based on the criteria of VIP, adjusted p and $|p(\text{corr})|$. These independent or overlapping metabolites were visualized using a Venn diagram for better comprehension:

Supplementary Figure 3C. Independent or overlapping potential biomarkers between early and late ESCCs.

Interestingly, we observed a considerable overlap in metabolites shared between early and late ESCCs. This suggests that such metabolites might not only work in the early detection of ESCC, but also hold the potential for monitoring tumor progression, warranting continued focus in our future research efforts.

Measures taken:

- 1) The analysis of advanced-stage ESCC biomarkers was updated in **Supplementary Table 6** in the Document "Supplementary Tables".
- 2) The Venn diagram has been added to **Supplementary Figure 3** in the Document "Supplementary Figures".
- 3) In the Document "Article File", "Potential biomarkers included 43, 34, and 12 metabolites for ESCC tissues vs. normal mucosa, early ESCC tissues vs. normal mucosa, and early ESCC tissue vs. advanced ESCC tissue, respectively" in L 114-116 **has been revised to** "According to the screening criteria (Variable importance in the projection (VIP) > 1, adjusted $p < 0.05$ and significant $|p(\text{corr})|$ values), there were 43, 34, 12 and 45 potential metabolic biomarkers identified for ESCC tissues vs. normal mucosa, early ESCC tissues vs. normal mucosa, early ESCC tissue vs. advanced ESCC

tissue, and advanced ESCC tissue vs. Normal mucosa, respectively", which has been highlighted in red in the revised manuscript in L 118-122.

8) Regarding the validation of the NMR results of early ESCC tissues using LC-MS/MS and GC/MS-MS, the description and presentation is difficult to follow (Figure 3). 1) Were LC-MS/MS and GC/MS-MS both used for the validation? If so, were LC-MS/MS and GC/MS-MS using the same customized panels? 2) How many of the 34 hits from the NMR results on early ESCC (Fig.2J) were validated by LC-MS/MS and GC/MS-MS? Were they validated by both LC-MS/MS and GC/MS-MS or just one of these two methods? 3) Line 133 "We then identified the top 30 metabolite variables with high correlation and weight in both datasets using the loading plot (Fig. 3D) and listed them in Supplementary Table 3 (right panel)." Are these 30 from the 34 hits from the NMR results on early ESCC (Fig.2J)?

Response: Thank you for your careful review and questions. We apologize for the lack of clarity in our previous descriptions. Here, we address each of your questions and improve the manuscript accordingly:

1) Early-stage tissues underwent absolute quantification via MS analysis, including 544 metabolites across 13 classes detected through LC/MS-MS and 11 fatty acids via GC/MS-MS, covering most of the substances we identified in NMR. Please find the detailed information list of targeted metabolites at the end of this "Point-by-point response" document (Page 37-53).

2) 34 early differential metabolites from the NMR results on early ESCC (Fig. 2J) are visualized in the upper panel of Supplementary Figure 3C (identified by red circles, as indicated in the previous response). Here, we extracted information from Supplementary Tables 3, 6, and 7 and organized it into a table for your review:

Validation Status of 34 NMR-Based Biomarkers in Early-Stage ESCC via MS Analysis.

NMR-based biomarkers of Early-stage ESCC	Fold Change	AUC	Target MS-based validation (GC-MS, µg/g; LC-MS, nmol/g)	Early ESCC tissue	Normal tissue	Verification platform	MS validation Coverage	Consistency	O2PLS
2-hydroxybutyrate	1.658	0.858	2-Hydroxybutyric acid	32.743	32.052	LC-MS/MS	√	√	√
3-Hydroxybutyrate	1.315	0.869	3-Hydroxybutyric acid	116.753	100.351	LC-MS/MS	√	√	√
3-Hydroxypropionic acid	0.79	0.82	3-Hydroxypropionic acid	29.574	56.51	LC-MS/MS	√	√	√
3-Methylhistidine	0.145	0.986	3-Methylhistidine	2.949	2.714	LC-MS/MS	√		
Acetate	1.565	0.913	Acetic acid	15.62	7.406	GC/MS-MS	√	√	
Adenine	1.502	0.768	Adenine	1.991	1.85	LC-MS/MS	√	√	√
Arginine	1.629	0.893	L-Arginine	444.938	360.606	LC-MS/MS	√	√	√
Asparagine	3.366	0.99	L-Asparagine	317.513	153.752	LC-MS/MS	√	√	√
beta-Glucose	0.77	0.931	D-Glucose	276.119	613.372	LC-MS/MS	√	√	√
Choline	1.153	0.851	Choline	477.495	329.466	LC-MS/MS	√	√	
Flavin mononucleotide	1.166	0.775	Flavin mononucleotide	0.757	0.5	LC-MS/MS	√	√	√
Glutamate	1.502	0.969	L-Glutamic acid	3742.348	2153.195	LC-MS/MS	√	√	√
Glutamine	0.805	0.817	L-Glutamine	1084.284	1085.869	LC-MS/MS	√	√	√
Glutathione	3.169	0.969	Glutathione	6714.039	4725.835	LC-MS/MS	√	√	√
Glycine	1.251	0.772	L-Glycine	514.68	393.479	LC-MS/MS	√	√	
Histidine	1.535	0.976	L-Histidine	–	–	LC-MS/MS			√
Isobutyrate	1.438	0.869	Isobutyric acid	–	–	GC/MS-MS			√
Isoleucine	1.442	0.955							√
Leucine	1.318	0.851	L-Leucine	607.886	322.439	LC-MS/MS	√	√	√
Lysine	1.125	0.782	L-Lysine	205.858	186.214	LC-MS/MS	√	√	
Malate	1.244	0.841	L-Malic acid	234.576	306.851	LC-MS/MS	√		√
Methylamine	1.603	0.979							√
Palmitic acid	1.358	0.903							
Phenylalanine	1.429	0.83	L-Phenylalanine	296.179	190.67	LC-MS/MS	√	√	√
Phosphocholine	1.57	0.848							√
Proline	1.211	0.851	L-Proline	681.169	354.256	LC-MS/MS	√	√	√
Propionate	1.321	0.931	Propionic acid	6.239	1.309	GC/MS-MS	√	√	√
Pyruvate	1.281	0.872	Pyruvic acid	884.299	1209.432	LC-MS/MS	√		√
Taurine	0.71	0.879	Taurine	775.959	804.521	LC-MS/MS	√	√	√

Theophylline	1.385	0.934	Theophylline	–	–	LC-MS/MS			
Threonine	1.135	0.765	L-Threonine	361.769	246.422	LC-MS/MS	√	√	
Trimethylamine N-oxide	0.894	0.751	Trimethylamine N-oxide	7.621	8.03	LC-MS/MS	√	√	√
Tyrosine	1.344	0.806	L-Tyrosine	354.841	237.234	LC-MS/MS	√	√	√
Valine	1.329	0.83	L-Valine	855.831	554.316	LC-MS/MS	√	√	

3) From the table content, it can be seen that MS-targeted quantitative detection (using either LC-MS/MS or GC-MS/MS) covers 80% (27/34) of NMR-based potential biomarkers in early ESCC (refer to "MS validation Coverage" column). Among these, 89% (24/27) of the metabolites showed consistent expression trends between NMR and MS results (refer to the "Consistency" column). In the O2PLS model integrating NMR and MS data matrices, 74% (25/34) of NMR metabolites overlapped with the top 30 metabolite variables, indicating a close overall correlation between NMR and MS data (refer to the "O2PLS" column).

4) Most of the potential biomarkers in early ESCC tissue presented in Table 1 have undergone concurrent validation through NMR and MS, at least confirmed to be closely associated with the targeted MS matrix through O2PLS analysis.

Measures taken:

In the Document "Article File", "Liquid chromatography-tandem mass spectrometry (LC-MS/MS) and gas chromatography-tandem mass spectrometer (GC/MS-MS) were used to validate the NMR results of early ESCC tissues on the same batch. 600 biochemicals were quantified by multiple reaction monitoring (MRM) mode" in Lines (L) 123-125 **has been revised to** "Targeted quantitative MS were used to validate the NMR results of early ESCC tissues on the same batch. Over 500 biochemicals were quantified by multiple reaction monitoring (MRM) mode, including 544 metabolites across 13 classes detected via Liquid chromatography-tandem mass spectrometry (LC/MS-MS) and 11 fatty acids detected via gas chromatography-tandem mass spectrometer (GC/MS-MS), covering most of the substances we identified in NMR.", which has been highlighted in red in the revised manuscript in **L 132-136**.

9) *The NMR profiles the global metabolomics, including amino acids, lipids, nucleotides, carbohydrates, etc. however, the most significant changes were seen mostly in amino acids (e.g, Fig.2K, 3D, Supplementary Fig. 4A-B). Indeed, one of the conclusion is "Alanine, aspartate and glutamate metabolism is a key functional pathway in ESCC progression", which are all amino acids. Can the authors discuss why amino acid metabolism seems to dominate these metabolic changes?*

Response: Thank you for your valuable suggestion. Our findings indicate varying degrees of disruption across most metabolic pathways during ESCC progression. However, this study primarily focuses on observing the longitudinal metabolic changes specific to ESCC tissues; from this perspective, changes in amino acid metabolism appear notably active. We further elaborate on amino acid metabolism in the Discussion section.

Measures taken:

In the Document "Article File", "Our findings reveal varying degrees of disruption across most categories of metabolic pathways during ESCC progression, and this is because human metabolism is a complex, resilient and sophisticated regulatory ecosystem where a single change can affect the entire system. However, this study primarily focuses on observing the longitudinal metabolic changes specific to ESCC tissues. From this perspective, changes in amino acid metabolism appear notably active. Amino acids serve essential roles within tumors and their microenvironment. First, they are vital nutrients for all cell types, contributing substantially to cancer cell survival and growth. For example, glutamine is largely anaplerotic and relinquishes both amine groups to support the TCA cycle¹⁵. Second, enhanced biosynthetic activities are an essential feature of metabolic reprogramming in cancer, and amino acids play crucial roles in the synthesis of proteins, lipids, and nucleic acids¹⁶. Third, proliferating cancer cells accumulate reactive oxygen species, damaging macromolecules and eventually causing cell death. Cancer cells rely on glutamine, glycine, and cysteine to synthesize glutathione and regulate redox balance to address this problem¹⁷. In addition, amino acid derivatives contribute to epigenetic regulation and immune response associated with tumor occurrence and metastasis¹⁸. We also illustrate how the 'Alanine, aspartate and glutamate metabolism' pathway, a characteristic molecular event in ESCC, can inspire potential metabolic liabilities as therapeutic targets." **has been added to** the revised manuscript, highlighted in red in **L 418-432**.

10) *In the introduction, the authors described novel biomarkers for ESCC diagnosis, including serum miRNA, autoantibodies, somatic gene mutations, salivary exosomes, and artificial intelligence (AI)-assisted sponge cytology. DNA methylation markers is an important group and should be discussed.*

Response: Thank you for your valuable advice. We agree with the importance of DNA methylation markers in the progress of ESCC and have made the following modifications to the Introduction and Discussion sections.

Measures taken:

In Document "Article File":

1) Introduction Section: "DNA methylation markers (*Genome Biol* 24, 193, doi:10.1186/s13059-023-03035-3 (2023))¹⁹" **has been added to** the revised manuscript, highlighted in red in the revised

manuscript in L 54.

2) Discussion Section: "Liquid biopsy assays, including circulating tumor cells (CTCs), circulating tumor DNA (ctDNA), and extracellular vesicles (EVs), provide a promising strategy for the early detection of various human cancers, including ESCC, primarily due to their abundance in various biofluids and their capability to faithfully reflect tissue specificity." in L 378-381 **has been revised to** "It can be used to examine cancer-derived circulating tumor cells (CTCs), circulating nucleic acids such as circulating tumor DNA (ctDNA) and cell-free DNA (cfDNA), extracellular vesicles (EVs), tumor-educated platelets, proteins, and metabolites (*Cell Rep Med* 4, 101198, doi:10.1016/j.xcrm.2023.101198 (2023))²⁰. Due to their abundance across biofluids, especially the cfDNA methylation-based technology, which can faithfully reflect tissue-specificity with high tissue origin accuracy, liquid biopsy offers a promising strategy for the early detection of various human cancers, including ESCC (*Nat Commun* 14, 893, doi:10.1038/s41467-023-36557-2 (2023); *Signal Transduct Target Ther* 7, 53, doi:10.1038/s41392-022-00873-8 (2022); *Ann Oncol* 34, 486-495, doi:10.1016/j.annonc.2023.02.010 (2023))²¹⁻²³.", which has been highlighted in red in the revised manuscript in L 377-382.

11) The paper reads fairly cumbersome, and I recommend it to be edited by professionals.

Response: Many thanks for the kind suggestion. As requested, the manuscript has been edited by an English-speaking person and appropriate grammatical and typographical corrections have been made.

De-Chen Lin, Ph.D.

University of Southern California

References:

- 1 Morgan, E. *et al.* The Global Landscape of Esophageal Squamous Cell Carcinoma and Esophageal Adenocarcinoma Incidence and Mortality in 2020 and Projections to 2040: New Estimates From GLOBOCAN 2020. *Gastroenterology* **163**, 649-658 e642, doi:10.1053/j.gastro.2022.05.054 (2022).
- 2 Shi, J. F. *et al.* [Priority setting in scaled-up cancer screening in China: an systematic review of economic evaluation evidences]. *Zhonghua Yu Fang Yi Xue Za Zhi* **54**, 306-313, doi:10.3760/cma.j.issn.0253-9624.2020.03.012 (2020).
- 3 Johnson, C. H., Ivanisevic, J. & Siuzdak, G. Metabolomics: beyond biomarkers and towards mechanisms. *Nat Rev Mol Cell Biol* **17**, 451-459, doi:10.1038/nrm.2016.25 (2016).
- 4 Tarazona, S. *et al.* Harmonization of quality metrics and power calculation in multi-omic studies. *Nat Commun* **11**, 3092, doi:10.1038/s41467-020-16937-8 (2020).
- 5 Beckonert, O. *et al.* Metabolic profiling, metabolomic and metabonomic procedures for NMR spectroscopy of urine, plasma, serum and tissue extracts. *Nat Protoc* **2**, 2692-2703, doi:10.1038/nprot.2007.376 (2007).
- 6 Kirwan, J. A. *et al.* Preanalytical Processing and Biobanking Procedures of Biological

- Samples for Metabolomics Research: A White Paper, Community Perspective (for "Precision Medicine and Pharmacometabolomics Task Group"-The Metabolomics Society Initiative). *Clin Chem* **64**, 1158-1182, doi:10.1373/clinchem.2018.287045 (2018).
- 7 Sun, C. *et al.* Spatially resolved metabolomics to discover tumor-associated metabolic alterations. *Proc Natl Acad Sci U S A* **116**, 52-57, doi:10.1073/pnas.1808950116 (2019).
- 8 Zhao, J. *et al.* A multi-platform metabolomics reveals possible biomarkers for the early-stage esophageal squamous cell carcinoma. *Anal Chim Acta* **1220**, 340038, doi:10.1016/j.aca.2022.340038 (2022).
- 9 Julkunen, H. *et al.* Atlas of plasma NMR biomarkers for health and disease in 118,461 individuals from the UK Biobank. *Nat Commun* **14**, 604, doi:10.1038/s41467-023-36231-7 (2023).
- 10 Buerger, T. *et al.* Metabolomic profiles predict individual multidisease outcomes. *Nat Med* **28**, 2309-2320, doi:10.1038/s41591-022-01980-3 (2022).
- 11 Liu, N. *et al.* Comparison of Untargeted Metabolomic Profiling vs Traditional Metabolic Screening to Identify Inborn Errors of Metabolism. *JAMA Netw Open* **4**, e2114155, doi:10.1001/jamanetworkopen.2021.14155 (2021).
- 12 Norton, E. C., Dowd, B. E. & Maciejewski, M. L. Odds Ratios-Current Best Practice and Use. *JAMA* **320**, 84-85, doi:10.1001/jama.2018.6971 (2018).
- 13 Liu, S. *et al.* Mycobacterium tuberculosis suppresses host DNA repair to boost its intracellular survival. *Cell Host Microbe* **31**, 1820-1836 e1810, doi:10.1016/j.chom.2023.09.010 (2023).
- 14 Huang, Y. *et al.* Mapping the early life gut microbiome in neonates with critical congenital heart disease: multiomics insights and implications for host metabolic and immunological health. *Microbiome* **10**, 245, doi:10.1186/s40168-022-01437-2 (2022).
- 15 Hensley, C. T., Wasti, A. T. & DeBerardinis, R. J. Glutamine and cancer: cell biology, physiology, and clinical opportunities. *J Clin Invest* **123**, 3678-3684, doi:10.1172/JCI69600 (2013).
- 16 Lieu, E. L., Nguyen, T., Rhyne, S. & Kim, J. Amino acids in cancer. *Exp Mol Med* **52**, 15-30, doi:10.1038/s12276-020-0375-3 (2020).
- 17 Li, X. & Zhang, H. S. Amino acid metabolism, redox balance and epigenetic regulation in cancer. *FEBS J*, doi:10.1111/febs.16803 (2023).
- 18 Kelly, B. & Pearce, E. L. Amino Assets: How Amino Acids Support Immunity. *Cell Metab* **32**, 154-175, doi:10.1016/j.cmet.2020.06.010 (2020).
- 19 Zheng, Y. *et al.* Comprehensive analyses of partially methylated domains and differentially methylated regions in esophageal cancer reveal both cell-type- and cancer-specific epigenetic regulation. *Genome Biol* **24**, 193, doi:10.1186/s13059-023-03035-3 (2023).
- 20 Batool, S. M. *et al.* The Liquid Biopsy Consortium: Challenges and opportunities for early cancer detection and monitoring. *Cell Rep Med* **4**, 101198, doi:10.1016/j.xcrm.2023.101198 (2023).
- 21 Lin, D. C. Large-scale genomic analyses reveal alterations and mechanisms underlying clonal evolution and immune evasion in esophageal cancer. *Nat Commun* **14**, 893, doi:10.1038/s41467-023-36557-2 (2023).
- 22 Xi, Y. *et al.* Multi-omic characterization of genome-wide abnormal DNA methylation reveals diagnostic and prognostic markers for esophageal squamous-cell carcinoma.

Signal Transduct Target Ther **7**, 53, doi:10.1038/s41392-022-00873-8 (2022).

- 23 Gao, Q. *et al.* Unintrusive multi-cancer detection by circulating cell-free DNA methylation sequencing (THUNDER): development and independent validation studies. *Ann Oncol* **34**, 486-495, doi:10.1016/j.annonc.2023.02.010 (2023).

Detailed information on targeted metabolites.

No.	Compound Name	Formula	CAS	HMDB	KEGG	Class
1	(S)-beta-Aminoisobutyric acid	C4H9NO2	4249-19-8	HMDB0002166	C03284	Amino acids, peptides, and analogues
2	1,1-Dimethyl-prolinium	C7H14NO2+	4136-37-2	HMDB0004827	C10172	Amino acids, peptides, and analogues
3	1,3,7-Trimethyluric acid	C8H10N4O3	5415-44-1	HMDB0002123	C16361	Purines and purine derivatives
4	1,3-Dimethyluric-Acid	C7H8N4O3	944-73-0	HMDB0001857		Purines and purine derivatives
5	1,4-Dihyronicotinamide adenine dinucleotide	C21H29N7O14P2	58-68-4	HMDB0001487	C00004	Nucleosides, nucleotides, and analogues
6	1,5-Anhydro-D-Glucitol	C6H12O5	154-58-5	HMDB0002712	C07326	Carbohydrates and carbohydrate conjugates
7	1,7-Dimethyluric acid	C7H8N4O3	33868-03-0	HMDB0011103	C16356	Purines and purine derivatives
8	11-Deoxycortisol	C21H30O4	152-58-9	HMDB0000015	C05488	Steroids and steroid derivatives
9	17alpha-Hydroxyprogesterone	C21H30O3	68-96-2	HMDB0000374	C01176	Steroids and steroid derivatives
10	1-Aminopropan-2-ol	C3H9NO	78-96-6	HMDB0012136	C03194	Biogenic Amines
11	1-Kestose	C18H32O16	470-69-9	HMDB0011729	C03661	Carbohydrates and carbohydrate conjugates
12	1-Methyladenosine	C11H15N5O4	15763-06-1	HMDB0003331	C02494	Nucleosides, nucleotides, and analogues
13	1-Methyl-L-histidine	C7H11N3O2	332-80-9	HMDB0000001	C01152	Amino acids, peptides, and analogues
14	1-Methyluric acid	C6H6N4O3	708-79-2	HMDB0003099	C16359	Purines and purine derivatives
15	1-Methylxanthine	C6H6N4O2	6136-37-4	HMDB0010738	C16358	Purines and purine derivatives
16	2,3-Dihydroxybenzoic acid	C7H6O4	303-38-8	HMDB0000397	C00196	Benzenoids
17	2,4-Dihydroxybenzoic acid	C7H6O4	89-86-1	HMDB0029666		Benzenoids
18	2,4-Dihydroxypyrimidine-5-carboxylic acid	C5H4N2O4	23945-44-0		C03030	Pyrimidines and pyrimidine derivatives
19	2-Amino-2-deoxy-D-gluconic acid	C6H13NO6	3646-68-2	HMDB0250755	C03752	Carbohydrates and carbohydrate conjugates
20	2-Aminoisobutyric acid	C4H9NO2	62-57-7	HMDB0001906	C03665	Amino acids, peptides, and analogues
21	2-Aminooctanoic acid	C8H17NO2	644-90-6	HMDB0000991		Amino acids, peptides, and analogues
22	2-dehydro-D-gluconic acid	C6H10O7	669-90-9		C06473	Carbohydrates and carbohydrate conjugates
23	2'-Deoxyadenosine	C10H13N5O3	958-09-8	HMDB0000101	C00559	Nucleosides, nucleotides, and analogues
24	2'-Deoxyadenosine-5'-monophosphate	C10H14N5O6P	653-63-4	HMDB0000905	C00360	Nucleosides, nucleotides, and analogues
25	2'-Deoxycytidine diphosphate	C9H15N3O10P2	800-73-7	HMDB0001245	C00705	Nucleosides, nucleotides, and analogues
26	2'-Deoxycytidine-5'-monophosphate	C9H14N3O7P	1032-65-1	HMDB0001202	C00239	Nucleosides, nucleotides, and analogues
27	2'-Deoxyguanosine 5'-monophosphate	C10H14N5O7P	902-04-5	HMDB0001044	C00362	Nucleosides, nucleotides, and analogues
28	2'-Deoxyguanosine-5'-diphosphate	C10H15N5O10P2	3493-09-2	HMDB0000960	C00361	Nucleosides, nucleotides, and analogues
29	2'-Deoxyuridine	C9H12N2O5	951-78-0	HMDB0000012	C00526	Nucleosides, nucleotides, and analogues
30	2'-Deoxyuridine 5'-triphosphate	C9H15N2O14P3	1173-82-6	HMDB0001191	C00460	Nucleosides, nucleotides, and analogues
31	2-Furoic acid	C5H4O3	88-14-2	HMDB0000617	C01546	Organic acids and derivatives

32	2-Hydroxy-2-methylbutyric acid	C5H10O3	3739-30-8	HMDB0001987		Organic acids and derivatives
33	2-Hydroxy-3-methylbutyric acid	C5H10O3	4026-18-0	HMDB0000407		Organic acids and derivatives
34	2-Hydroxyadipic acid	C6H10O5	18294-85-4	HMDB0000321	C02360	Organic acids and derivatives
35	2-Hydroxybutyric acid	C4H8O3	600-15-7	HMDB0000008	C05984	Organic acids and derivatives
36	2-Hydroxyglutaric acid	C5H8O5	2889-31-8	HMDB0059655	C20267	Organic acids and derivatives
37	2-Hydroxyisobutyric acid	C4H8O3	594-61-6	HMDB0000729		Organic acids and derivatives
38	2-Hydroxyphenylacetic acid	C8H8O3	614-75-5	HMDB0000669	C05852	Benzenoids
39	2-indolecarboxylic acid	C9H7NO2	1477-50-5	HMDB0002285		Indoles and derivatives
40	2-Isopropylmalic acid	C7H12O5	49601-06-1	HMDB0000402	C02504	Organic acids and derivatives
41	2-Ketoglutaric Acid	C5H6O5	328-50-7	HMDB0000208	C00026	Organic acids and derivatives
42	2-Methoxybenzoic acid	C8H8O3	579-75-9	HMDB0032604		Benzenoids
43	2-Methylbenzoic acid	C8H8O2	118-90-1	HMDB0002340	C07215	Benzenoids
44	2-Methylcitric acid	C7H10O7	6061-96-7	HMDB0000379	C02225	Organic acids and derivatives
45	2-Methylglutaric acid	C6H10O4	617-62-9	HMDB0000422		Organic acids and derivatives
46	2-Methylhippuric acid	C10H11NO3	42013-20-7	HMDB0011723	C01586	Benzenoids
47	2'-O-methylguanosine	C11H15N5O5	2140-71-8			Nucleosides, nucleotides, and analogues
48	2-Oxobutanoic acid	C4H6O3	600-18-0	HMDB0000005	C00109	Organic acids and derivatives
49	2-Oxohexanoic acid	C6H10O3	2492-75-3	HMDB0001864	C00902	Organic acids and derivatives
50	2-Phenylacetamide	C8H9NO	103-81-1	HMDB0010715	C02505	Benzenoids
51	2-Phenylglycine	C8H9NO2	2835-06-5	HMDB0002210		Amino acids, peptides, and analogues
52	2-Piperidone	C5H9NO	675-20-7	HMDB0011749		Azacyclic compounds
53	2-Thiocytidine	C9H13N3O4S	13239-97-9			Nucleosides, nucleotides, and analogues
54	3-(3-Hydroxyphenyl)propanoic acid	C9H10O3	621-54-5	HMDB0000375	C11457	Benzenoids
55	3-(Methylthio)propionate	C4H8O2S	646-01-5	HMDB0001527	C08276	Lipids
56	3,4-Dihydroxybenzoic acid	C7H6O4	99-50-3	HMDB0001856	C00230	Benzenoids
57	3,4-Dihydroxyhydrocinnamic acid	C9H10O4	1078-61-1	HMDB0000423	C10447	Benzenoids
58	3,4-Dihydroxymandelic acid	C8H8O5	775-01-9	HMDB0001866	C05580	Benzenoids
59	3,4-Dihydroxyphenylacetic acid	C8H8O4	102-32-9	HMDB0001336	C01161	Benzenoids
60	3,5-Dihydroxybenzoic acid	C7H6O4	99-10-5	HMDB0013677		Benzenoids
61	3,5-Diiodo-L-thyronine	C15H13I2NO4	1041-01-6	HMDB0000582		Amino acids, peptides, and analogues
62	3,7-Dimethyluric acid	C7H8N4O3	13087-49-5	HMDB0001982	C16360	Purines and purine derivatives
63	3-Aminoisobutyric acid	C4H8NO2-	144-90-1	HMDB0003911	C05145	Amino acids, peptides, and analogues
64	3-Guanidinopropionic acid	C4H9N3O2	353-09-3	HMDB0013222	C03065	Organic acids and derivatives

65	3-Hydroxy-3-methylglutaric acid	C6H10O5	503-49-1	HMDB0000355	C03761	Organic acids and derivatives
66	3-Hydroxyanthranilic acid	C7H7NO3	548-93-6	HMDB0001476	C00632	Benzenoids
67	3-Hydroxybutyric acid	C4H8O3	300-85-6	HMDB0000011	C01089	Organic acids and derivatives
68	3-Hydroxycinnamic acid	C9H8O3	14755-02-3	HMDB0001713	C12621	Organic acids and derivatives
69	3-Hydroxydecanoic acid	C10H20O3	14292-26-3	HMDB0002203		Organic acids and derivatives
70	3-Hydroxyglutaric acid	C5H8O5	638-18-6	HMDB0000428		Organic acids and derivatives
71	3-Hydroxyisobutyric acid	C4H8O3	2068-83-9	HMDB0000023	C06001	Organic acids and derivatives
72	3-Hydroxykynurenine	C10H12N2O4	2147-61-7	HMDB0000732	C02794	Amino acids, peptides, and analogues
73	3-Hydroxypropionic acid	C3H6O3	503-66-2	HMDB0000700	C01013	Organic acids and derivatives
74	3-Indoleacrylic Acid	C11H9NO2	29953-71-7	HMDB0000734		Indoles and derivatives
75	3-Indoleglyoxylic acid	C10H7NO3	1477-49-2	HMDB0242143		Indoles and derivatives
76	3-Indolepropionic acid	C11H11NO2	830-96-6	HMDB0002302		Indoles and derivatives
77	3-Iodo-L-tyrosine	C9H10INO3	70-78-0	HMDB0000021	C02515	Amino acids, peptides, and analogues
78	3-Methoxybenzoic acid	C8H8O3	586-38-9	HMDB0032606		Benzenoids
79	3-Methoxytyrosine	C10H13NO4	7636-26-2	HMDB0001434		Amino acids, peptides, and analogues
80	3-Methyl-2-oxovaleric acid	C6H10O3	24809-08-3	HMDB0000491	C00671	Organic acids and derivatives
81	3-Methyladenine	C6H7N5	5142-23-4	HMDB0011600	C00913	Purines and purine derivatives
82	3-Methyladipic acid	C7H12O4	3058-01-3	HMDB0000555		Organic acids and derivatives
83	3-Methylhistidine	C7H11N3O2	368-16-1	HMDB0000479	C01152	Amino acids, peptides, and analogues
84	3-Methylphenylacetic acid	C9H10O2	621-36-3	HMDB0002222		Benzenoids
85	3-Methyluric acid	C6H6N4O3	605-99-2	HMDB0001970		Purines and purine derivatives
86	3-Methylxanthine	C6H6N4O2	1076-22-8	HMDB0001886	C16357	Purines and purine derivatives
87	3-Nitrotyrosine	C9H10N2O5	621-44-3	HMDB0001904		Amino acids, peptides, and analogues
88	3-Oxochoolic acid	C24H38O5	2304-89-4	HMDB0000502		Steroids and steroid derivatives
89	3-Ureidopropionic acid	C4H8N2O3	462-88-4	HMDB0000026	C02642	Organic acids and derivatives
90	4-(Methylthio)benzoic acid	C8H8O2S	13205-48-6			Benzenoids
91	4-(Trimethylammonio)butanoate	C7H15NO2	407-64-7	HMDB0001161	C01181	Biogenic Amines
92	4,4'-Methylenediphenol	C13H12O2	620-92-8	HMDB0240711	C14298	Benzenoids
93	4,6-Dihydroxyquinoline	C9H7NO2	3517-61-1	HMDB0004077	C05639	Pyridines and derivatives
94	4-Acetamidobutyric acid	C6H11NO3	3025-96-5	HMDB0003681		Amino acids, peptides, and analogues
95	4-Aminobutyric acid	C4H9NO2	56-12-2	HMDB0000112	C00334	Amino acids, peptides, and analogues
96	4-Hydroxy-3-methylbenzoic acid	C8H8O3	499-76-3	HMDB0004815	C21167	Benzenoids
97	4-Hydroxybenzoic acid	C7H6O3	99-96-7	HMDB0000500	C00156	Benzenoids

98	4-Hydroxyhippuric acid	C9H9NO4	2482-25-9	HMDB0013678		Amino acids, peptides, and analogues
99	4-Hydroxyphenylpyruvic acid	C9H8O4	156-39-8	HMDB0000707	C01179	Benzenoids
100	4-Methoxybenzaldehyde	C8H8O2	123-11-5	HMDB0029686		Benzenoids
101	4-Methyl-2-oxopentanoic acid	C6H10O3	816-66-0	HMDB0000695	C00233	Organic acids and derivatives
102	4-Methylhippuric acid	C10H11NO3	27115-50-0	HMDB0013292		Amino acids, peptides, and analogues
103	4-Pyridoxic acid	C8H9NO4	82-82-6	HMDB0000017	C00847	Pyridines and derivatives
104	5alpha-Pregnane-3,20-dione	C21H32O2	566-65-4	HMDB0003759	C03681	Steroids and steroid derivatives
105	5-Aminoimidazole-4-carboxamide	C4H6N4O	360-97-4	HMDB0003192	C04051	Azacyclic compounds
106	5-Aminovaleric acid	C5H11NO2	660-88-8	HMDB0003355	C00431	Amino acids, peptides, and analogues
107	5'-Deoxy-5'-methylthioadenosine	C11H15N5O3S	2457-80-9	HMDB0001173	C00170	Nucleosides, nucleotides, and analogues
108	5-Hydroxyindole	C8H7NO	1953-54-4	HMDB0059805		Indoles and derivatives
109	5-Hydroxyindole-3-acetic acid	C10H9NO3	54-16-0	HMDB0000763	C05635	Indoles and derivatives
110	5-Hydroxylysine	C6H14N2O3	1190-94-9	HMDB0000450	C16741	Amino acids, peptides, and analogues
111	5-hydroxymethyl-2'-deoxycytidine	C10H15N3O5	7226-77-9			Nucleosides, nucleotides, and analogues
112	5-Hydroxymethylcytidine	C10H15N3O6	19235-17-7			Nucleosides, nucleotides, and analogues
113	5-Hydroxytryptophan	C11H12N2O3	4350-09--8	HMDB0000472	C00643	Amino acids, peptides, and analogues
114	5-Methoxyindoleacetate	C11H11NO3	3471-31-6	HMDB0004096	C05660	Indoles and derivatives
115	5-Methoxytryptamine	C11H14N2O	608-07-1	HMDB0004095	C05659	Biogenic Amines
116	5-Methoxytryptophan	C12H14N2O3	25197-96-0	HMDB0002339		Amino acids, peptides, and analogues
117	5-methyl-2'-deoxycytidine	C10H15N3O4	838-07-3	HMDB0002224	C03592	Nucleosides, nucleotides, and analogues
118	5-Methylcytidine	C10H15N3O5	2140-61-6	HMDB0000982		Nucleosides, nucleotides, and analogues
119	5-Methylcytosine	C5H7N3O	554-01-8	HMDB0002894	C02376	Pyrimidines and pyrimidine derivatives
120	5-Methyltetrahydrofolic acid	C20H25N7O6	134-35-0	HMDB0001396	C00440	Pyridines and derivatives
121	6-Hydroxymelatonin	C13H16N2O3	2208-415	HMDB0004081	C05643	Indoles and derivatives
122	6-Hydroxynicotinic acid	C6H5NO3	5006-66-6	HMDB0002658	C01020	Pyridines and derivatives
123	6-Methyladenine	C6H7N5	443-72-1	HMDB0002099	C08434	Purines and purine derivatives
124	6-Phosphogluconic acid	C6H13O10P	921-62-0	HMDB0001316	C00345	Carbohydrates and carbohydrate conjugates
125	7-Dehydrocholesterol	C27H44O	434-16-2	HMDB0000032	C01164	Steroids and steroid derivatives
126	7-Ketcholesterol	C27H44O2	566-28-9	HMDB0000501		Steroids and steroid derivatives
127	7-Ketodeoxycholic acid	C24H38O5	911-40-0	HMDB0000391		Steroids and steroid derivatives
128	7-Methylguanine	C6H7N5O	578-76-7	HMDB0000897	C02242	Purines and purine derivatives
129	7-Methylguanosine	C11H16N5O5+	20244-86-4	HMDB0001107	C20674	Nucleosides, nucleotides, and analogues
130	7-Methyluric acid	C6H6N4O3	612-37-3	HMDB0011107	C16355	Purines and purine derivatives

131	7-Methylxanthine	C6H6N4O2	552-62-5	HMDB0001991	C16353	Purines and purine derivatives
132	8-Hydroxy-2'-deoxyguanosine	C10H13N5O5	88847-89-6	HMDB0003333		Nucleosides, nucleotides, and analogues
133	8-Hydroxyguanosine	C10H13N5O6	3868-31-3	HMDB0002044		Nucleosides, nucleotides, and analogues
134	Abrine	C12H14N2O2	526-31-8		C02983	Amino acids, peptides, and analogues
135	Acadesine	C9H14N4O5	2627-69-2	HMDB0062179	C04663	Nucleosides, nucleotides, and analogues
136	Acetaminophen	C8H9NO2	103-90-2	HMDB0001859	C06804	Benzenoids
137	Acetaminophen glucuronide	C14H17NO8	16110-10-4	HMDB0010316		Carbohydrates and carbohydrate conjugates
138	Acetoacetic acid	C4H6O3	541-50-4	HMDB0000060	C00164	Organic acids and derivatives
139	Acetyl coenzyme A	C23H38N7O17P3S	72-89-9	HMDB0001206	C00024	Nucleosides, nucleotides, and analogues
140	Acetylcholine	C7H16NO2+	51-84-3	HMDB0000895	C01996	Biogenic Amines
141	Acetylcysteine	C5H9NO3S	616-91-1	HMDB0001890	C06809	Amino acids, peptides, and analogues
142	Acetylucine	C8H15NO3	99-15-0	HMDB0011756	C02710	Amino acids, peptides, and analogues
143	Adenine	C5H5N5	73-24-5	HMDB0000034	C00147	Purines and purine derivatives
144	Adenosine	C10H13N5O4	58-61-7	HMDB0000050	C00212	Nucleosides, nucleotides, and analogues
145	Adenosine monophosphate	C10H14N5O7P	61-19-8	HMDB0000045	C00020	Nucleosides, nucleotides, and analogues
146	Adenosine-5'-diphosphate	C10H15N5O10P2	58-64-0	HMDB0001341	C00008	Nucleosides, nucleotides, and analogues
147	Adenosine-5'-triphosphate	C10H16N5O13P3	56-65-5	HMDB0000538	C00002	Nucleosides, nucleotides, and analogues
148	Adipic acid	C6H10O4	124-04-9	HMDB0000448	C06104	Organic acids and derivatives
149	Adrenosterone	C19H24O3	382-45-6	HMDB0006772	C05285	Steroids and steroid derivatives
150	Allantoic acid	C4H8N4O4	99-16-1	HMDB0001209	C00499	Amino acids, peptides, and analogues
151	Alloxan	C4H2N2O4	50-71-5	HMDB0002818		Pyrimidines and pyrimidine derivatives
152	alpha-Linolenic acid	C18H30O2	463-40-1	HMDB0001388	C06427	Lipids
153	Aminocaproic acid	C6H13NO2	60-32-2	HMDB0001901	C02378	Amino acids, peptides, and analogues
154	Aminohippuric acid	C9H10N2O3	61-78-9	HMDB0001867		Amino acids, peptides, and analogues
155	Aminolevulinic acid	C5H9NO3	106-60-5	HMDB0001149	C00430	Amino acids, peptides, and analogues
156	Anserine	C10H16N4O3	584-85-0	HMDB0000194	C01262	Amino acids, peptides, and analogues
157	Anthranilic acid	C7H7NO2	118-92-3	HMDB0001123	C00108	Benzenoids
158	Atrolactic acid	C9H10O3	515-30-0			Benzenoids
159	Azelaic acid	C9H16O4	123-99-9	HMDB0000784	C08261	Organic acids and derivatives
160	Benzoic acid	C7H6O2	65-85-0	HMDB0001870	C00180	Benzenoids
161	Benzophenone	C13H10O	119-61-9	HMDB0032049	C06354	Benzenoids
162	Benzoylformic acid	C8H6O3	611-73-4	HMDB0001587	C02137	Benzenoids
163	beta-Alanine	C3H7NO2	107-95-9	HMDB0000056	C00099	Amino acids, peptides, and analogues

164	beta-Hydroxyisovaleric acid	C5H10O3	625-08-1	HMDB0000754	C20827	Organic acids and derivatives
165	Betaine	C5H11NO2	107-43-7	HMDB0000043	C00719	Biogenic Amines
166	Bilirubin	C33H36N4O6	635-65-4	HMDB0000054	C00486	Azacyclic compounds
167	Biopterin	C9H11N5O3	22150-76-1	HMDB0000468	C06313	Azacyclic compounds
168	Biotin	C10H16N2O3S	58-85-5	HMDB0000030	C00120	Azacyclic compounds
169	Butylamine	C4H11N	109-73-9	HMDB0031321		Biogenic Amines
170	Caffeic acid	C9H8O4	331-39-5	HMDB0001964	C01197	Benzenoids
171	Caffeine	C8H10N4O2	58-08-2	HMDB0001847	C07481	Purines and purine derivatives
172	Calcifediol	C27H44O2	19356-17-3	HMDB0003550	C01561	Steroids and steroid derivatives
173	Carbocysteine	C5H9NO4S	638-23-3	HMDB0029415		Amino acids, peptides, and analogues
174	Cellobiose	C12H22O11	528-50-7	HMDB0000055	C00185	Carbohydrates and carbohydrate conjugates
175	Chenodeoxycholic acid	C24H40O4	474-25-9	HMDB0000518	C02528	Steroids and steroid derivatives
176	Cholecalciferol	C27H44O	67-97-0	HMDB0014315	C05443	Steroids and steroid derivatives
177	Cholic acid	C24H40O5	81-25-4	HMDB0000619	C00695	Steroids and steroid derivatives
178	Choline	C5H14NO+	62-49-7	HMDB0000097	C00114	Biogenic Amines
179	Ciliatine	C2H8NO3P	2041-14-7	HMDB0011747	C03557	Others
180	Cinnamoylglycine	C11H11NO3	16534-24-0	HMDB0011621		Amino acids, peptides, and analogues
181	cis-Aconitic acid	C6H6O6	585-84-2	HMDB0000072	C00417	Organic acids and derivatives
182	Citicoline	C14H26N4O11P2	987-78-0	HMDB0001413	C00307	Nucleosides, nucleotides, and analogues
183	Citraconic acid	C5H6O4	498-23-7	HMDB0000634	C02226	Organic acids and derivatives
184	Citramalic acid	C5H8O5	597-44-4	HMDB0000426	C00815	Organic acids and derivatives
185	Citric Acid	C6H8O7	77-92-9	HMDB0000094	C00158	Organic acids and derivatives
186	Corticosterone	C21H30O4	50-22-6	HMDB0001547	C02140	Steroids and steroid derivatives
187	Cortisone	C21H28O5	53-06-5	HMDB0002802	C00762	Steroids and steroid derivatives
188	Cotinine	C10H12N2O	486-56-6	HMDB0001046		Pyridines and derivatives
189	Creatine	C4H9N3O2	57-00-1	HMDB0000064	C00300	Amino acids, peptides, and analogues
190	Creatinine	C4H7N3O	60-27-5	HMDB0000562	C00791	Amino acids, peptides, and analogues
191	Curcumin	C21H20O6	458-37-7	HMDB0002269	C10443	Benzenoids
192	Cyanocobalamin	C63H88CoN14O14P	68-19-9	HMDB0000607	C02823	Azacyclic compounds
193	Cyclamic acid	C6H13NO3S	100-88-9	HMDB0031340	C02824	Others
194	Cyclic AMP	C10H12N5O6P	60-92-4	HMDB0000058	C00575	Nucleosides, nucleotides, and analogues
195	Cyclic guanosine monophosphate	C10H12N5O7P	7665-99-8	HMDB0001314	C00942	Nucleosides, nucleotides, and analogues
196	Cyclohexylamine	C6H13N	108-91-8	HMDB0031404	C00571	Biogenic Amines

197	Cystamine	C4H12N2S2	51-85-4	HMDB0250701		Biogenic Amines
198	Cysteic Acid	C3H7NO5S	13100-82-8	HMDB0002757	C00506	Amino acids, peptides, and analogues
199	Cysteine-S-sulfate	C3H7NO5S2	1637-71-4	HMDB0000731		Amino acids, peptides, and analogues
200	Cytidine 5'-monophosphate-N-acetylneuraminic acid	C20H31N4O16P	3063-71-6	HMDB0001176	C00128	Nucleosides, nucleotides, and analogues
201	Cytidine-5'-diphosphate	C9H15N3O11P2	63-38-7	HMDB0001546	C00112	Nucleosides, nucleotides, and analogues
202	Cytidine-5'-monophosphate	C9H14N3O8P	63-37-6	HMDB0000095	C00055	Nucleosides, nucleotides, and analogues
203	Cytosine	C4H5N3O	71-30-7	HMDB0000630	C00380	Pyrimidines and pyrimidine derivatives
204	D-(+)-Glucose	C6H12O6	54-17-1	HMDB0000122	C00031	Carbohydrates and carbohydrate conjugates
205	D-(+)-Mannose	C6H12O6	530-26-7	HMDB0000169	C00159	Carbohydrates and carbohydrate conjugates
206	D-(+)-Melibiose	C12H22O11	585-99-9	HMDB0000048		Carbohydrates and carbohydrate conjugates
207	D-Alanyl-D-alanine	C6H12N2O3	923-16-0	HMDB0003459	C00993	Amino acids, peptides, and analogues
208	Dehydroandrosterone	C19H28O2	2283-82-1	HMDB0005962		Steroids and steroid derivatives
209	Dehydroepiandrosterone	C19H28O2	53-43-0	HMDB0000077	C01227	Steroids and steroid derivatives
210	Deoxycholic acid	C24H40O4	83-44-3	HMDB0000626	C04483	Steroids and steroid derivatives
211	Deoxyguanosine	C10H13N5O4	961-07-9	HMDB0000085	C00330	Nucleosides, nucleotides, and analogues
212	Deoxyinosine	C10H12N4O4	890-38-0	HMDB0000071	C05512	Nucleosides, nucleotides, and analogues
213	Deoxyuridine monophosphate (dUMP)	C9H13N2O8P	964-26-1	HMDB0001409	C00365	Nucleosides, nucleotides, and analogues
214	Dephospho coenzyme A	C21H35N7O13P2S	3633-59-8	HMDB0001373	C00882	Nucleosides, nucleotides, and analogues
215	Desaminotyrosine	C9H10O3	501-97-3	HMDB0002199	C01744	Benzenoids
216	Desoxycortone	C21H30O3	64-85-7	HMDB0000016	C03205	Steroids and steroid derivatives
217	D-Galactose	C6H12O6	59-23-4	HMDB0000143	C00984	Carbohydrates and carbohydrate conjugates
218	D-Glucosamine	C6H13NO5	3416-24-8	HMDB0001514	C00329	Carbohydrates and carbohydrate conjugates
219	D-Glucosamine-6-sulfate	C6H13NO8S	91674-26-9	HMDB0000592		Carbohydrates and carbohydrate conjugates
220	D-Glucuronate	C6H10O7	70021-34-0/6556-12-3	HMDB0000127	C00191	Carbohydrates and carbohydrate conjugates
221	Diethanolamine	C4H11NO2	111-42-2	HMDB0004437	C06772	Biogenic Amines
222	Dihydrofolic acid	C19H21N7O6	4033-27-6	HMDB0001056	C00415	Amino acids, peptides, and analogues
223	Dihydrothymine	C5H8N2O2	696-04-8	HMDB0000079	C00906	Pyrimidines and pyrimidine derivatives
224	Dihydrouracil	C4H6N2O2	504-07-4	HMDB0000076	C00429	Pyrimidines and pyrimidine derivatives
225	Dihydroxyacetone phosphate	C3H7O6P	57-04-5	HMDB0001473	C00111	Carbohydrates and carbohydrate conjugates
226	Dimethylallyl pyrophosphate	C5H12O7P2	358-72-5	HMDB0001120	C00235	Lipids
227	Diphenylamine	C12H11N	122-39-4	HMDB0032562	C11016	Benzenoids
228	D-Mannosamine	C6H13NO5	2636-92-2	HMDB0250773	C03570	Carbohydrates and carbohydrate conjugates

229	Dodecanedioic acid	C12H22O4	693-23-2	HMDB0000623	C02678	Lipids
230	Dopamine	C8H11NO2	62-31-7	HMDB0000073	C03758	Biogenic Amines
231	Dopamine 3-O-sulfate	C8H11NO5S	51317-41-0	HMDB0006275	C13690	Biogenic Amines
232	Dopamine 4-O-sulfate	C8H11NO5S	38339-02-5	HMDB0004148	C13691	Biogenic Amines
233	D-Ribulose 5-phosphate	C5H11O8P	4151-19-3	HMDB0000618	C00199	Carbohydrates and carbohydrate conjugates
234	D-Sedoheptulose 7-phosphate	C7H13O10P-2	89927-08-2	HMDB0001068	C05382	Carbohydrates and carbohydrate conjugates
235	D-Tagatose	C6H12O6	17598-81-1	HMDB0003418	C00795	Carbohydrates and carbohydrate conjugates
236	Ectoine	C6H10N2O2	96702-03-3	HMDB0240650	C06231	Amino acids, peptides, and analogues
237	Enoxolone	C30H46O4	471-53-4	HMDB0011628	C02283	Lipids
238	Epinephrine	C9H13NO3	51-43-4	HMDB0000068	C00788	Biogenic Amines
239	Ergosterol	C28H44O	57-87-4	HMDB0000878	C01694	Steroids and steroid derivatives
240	Erucamide	C22H43NO	112-84-5	HMDB0244507		Organic acids and derivatives
241	Erythritol	C4H10O4	149-32-6	HMDB0002994	C00503	Carbohydrates and carbohydrate conjugates
242	Estrone	C18H22O2	53-16-7	HMDB0000145	C00468	Steroids and steroid derivatives
243	Ethanolamine	C2H7NO	141-43-5	HMDB0000149	C00189	Biogenic Amines
244	Ethylmalonic acid	C5H8O4	601-75-2	HMDB0000622		Organic acids and derivatives
245	Flavin adenine dinucleotide	C27H33N9O15P2	146-14-5	HMDB0001248	C00016	Nucleosides, nucleotides, and analogues
246	Flavin mononucleotide	C17H21N4O9P	146-17-8	HMDB0001520	C00061	Nucleosides, nucleotides, and analogues
247	Folic Acid	C19H19N7O6	59-30-3	HMDB0000121	C00504	Amino acids, peptides, and analogues
248	Fructose-6-phosphate	C6H13O9P	643-13-0	HMDB0000124	C00085	Carbohydrates and carbohydrate conjugates
249	Fumaric acid	C4H4O4	110-17-8	HMDB0000134	C00122	Organic acids and derivatives
250	Galactaric acid	C6H10O8	526-99-8	HMDB0000639	C00879	Carbohydrates and carbohydrate conjugates
251	Galactitol	C6H14O6	608-66-2	HMDB0000107	C01697	Carbohydrates and carbohydrate conjugates
252	Gallic acid	C7H6O5	149-91-7	HMDB0005807	C01424	Benzenoids
253	gamma-Carboxyglutamic acid	C6H9NO6	53861-57-7	HMDB0041900		Amino acids, peptides, and analogues
254	gamma-Glutamylalanine	C8H14N2O5	5875-41-2	HMDB0006248		Amino acids, peptides, and analogues
255	gamma-Glutamylmethionine	C10H18N2O5S	17663-87-5	HMDB0034367		Amino acids, peptides, and analogues
256	gamma-L-Glutamyl-L-valine	C10H18N2O5	2746-34-1	HMDB0011172		Amino acids, peptides, and analogues
257	GDP-L-Fucose	C16H25N5O15P2	15839-70-0	HMDB0001095	C00325	Nucleosides, nucleotides, and analogues
258	Gentiobiose	C12H22O11	5996-00-9		C08240	Carbohydrates and carbohydrate conjugates
259	Gentisaldehyde	C7H6O3	1194-98-5	HMDB0004062	C05585	Benzenoids
260	Gentisic acid	C7H6O4	490-79-9	HMDB0000152	C00628	Benzenoids
261	Glucaric acid	C6H10O8	87-73-0	HMDB0000663	C00818	Carbohydrates and carbohydrate conjugates

262	Gluconic acid	C6H12O7	526-95-4	HMDB0000625	C00257	Carbohydrates and carbohydrate conjugates
263	Gluconolactone	C6H10O6	90-80-2	HMDB0000150	C00198	Carbohydrates and carbohydrate conjugates
264	Glucose 1-phosphate	C6H13O9P	59-56-3	HMDB0001586	C00103	Carbohydrates and carbohydrate conjugates
265	Glucose 6-phosphate	C6H13O9P	299-31-0	HMDB0001401	C00092	Carbohydrates and carbohydrate conjugates
266	Glutaric acid	C5H8O4	110-94-1	HMDB0000661	C00489	Organic acids and derivatives
267	Glutathione Disulfide	C20H32N6O12S2	27025-41-8	HMDB0003337	C00127	Amino acids, peptides, and analogues
268	Glyceraldehyde	C3H6O3	56-82-6	HMDB0001051	C00577	Carbohydrates and carbohydrate conjugates
269	Glyceric acid	C3H6O4	473-81-4	HMDB0000139	C00258	Carbohydrates and carbohydrate conjugates
270	Glycerophosphocholine	C8H20NO6P	28319-77-9	HMDB0000086	C00670	Biogenic Amines
271	Glycerophosphoric acid	C3H9O6P	57-03-4	HMDB0000126	C00093	Carbohydrates and carbohydrate conjugates
272	Glycochenodeoxycholic acid	C26H43NO5	640-79-9	HMDB0000637	C05466	Steroids and steroid derivatives
273	Glycocholic acid	C26H43NO6	475-31-0	HMDB0000138	C01921	Steroids and steroid derivatives
274	Glycodeoxycholic acid	C26H43NO5	360-65-6	HMDB0000631	C05464	Steroids and steroid derivatives
275	Glycolic Acid	C2H4O3	79-14-1	HMDB0000115	C00160	Organic acids and derivatives
276	Glycyl-L-Leucine	C8H16N2O3	869-19-2	HMDB0028929	C02155	Amino acids, peptides, and analogues
277	Glycylphenylalanine	C11H14N2O3	3321-03-7	HMDB0028848		Amino acids, peptides, and analogues
278	Glycylproline	C7H12N2O3	704-15-4	HMDB0000721		Amino acids, peptides, and analogues
279	Guanidine	CH5N3	113-00-8	HMDB0001842	C17349	Biogenic Amines
280	Guanidinosuccinic acid	C5H9N3O4	6133-30-8	HMDB0003157	C03139	Amino acids, peptides, and analogues
281	Guanidoacetic acid	C3H7N3O2	352-97-6	HMDB0000128	C00581	Amino acids, peptides, and analogues
282	Guanine	C5H5N5O	73-40-5	HMDB0000132	C00242	Purines and purine derivatives
283	Guanosine	C10H13N5O5	118-00-3	HMDB0000133	C00387	Nucleosides, nucleotides, and analogues
284	Guanosine-5'-diphosphate	C10H15N5O11P2	146-91-8	HMDB0001201	C00035	Nucleosides, nucleotides, and analogues
285	Guanosine-5'-monophosphate	C10H14N5O8P	85-32-5	HMDB0001397	C00144	Nucleosides, nucleotides, and analogues
286	Guanosine-5'-triphosphate	C10H16N5O14P3	86-01--1	HMDB0001273	C00044	Nucleosides, nucleotides, and analogues
287	Hexadecanedioic acid	C16H30O4	505-54-4	HMDB0000672	C19615	Lipids
288	Hippuric acid	C9H9NO3	495-69-2	HMDB0000714	C01586	Amino acids, peptides, and analogues
289	Histamine	C5H9N3	51-45-6	HMDB0000870	C00388	Biogenic Amines
290	Histidinol	C6H11N3O	4836-52-6	HMDB0003431	C00860	Biogenic Amines
291	Homocitrulline	C7H15N3O3	1190-49-4	HMDB0000679	C02427	Amino acids, peptides, and analogues
292	Homocysteine thiolactone	C4H7NOS	3622-59-1	HMDB0002287		Amino acids, peptides, and analogues
293	Homogentisic acid	C8H8O4	451-13-8	HMDB0000130	C00544	Benzenoids
294	Homovanillic acid	C9H10O4	306-08-1	HMDB0000118	C05582	Benzenoids

295	Hordenine	C10H15NO	539-15-1	HMDB0004366	C06199	Biogenic Amines
296	Hydrocortisone	C21H30O5	50-23-7	HMDB0000063	C00735	Steroids and steroid derivatives
297	Hydroxyisocaproic acid	C6H12O3	13748-90-8	HMDB0000746		Organic acids and derivatives
298	Hydroxyphenyllactic acid	C9H10O4	306-23-0	HMDB0000755	C03672	Organic acids and derivatives
299	Hydroxyproline	C5H9NO3	51-35-4	HMDB0000725	C01157	Amino acids, peptides, and analogues
300	Hydroxytyrosol	C8H10O3	10597-60-1	HMDB0005784		Benzenoids
301	Hypotaurine	C2H7NO2S	300-84-5	HMDB0000965	C00519	Organic acids and derivatives
302	Hypoxanthine	C5H4N4O	68-94-0	HMDB0000157	C00262	Purines and purine derivatives
303	Imidazole	C3H4N2	288-32-4	HMDB0001525	C01589	Azacyclic compounds
304	Imidazoleacetic acid	C5H6N2O2	645-65-8	HMDB0002024	C02835	Azacyclic compounds
305	Indole	C8H7N	120-72-9	HMDB0000738	C00463	Indoles and derivatives
306	Indole-3-acetic acid	C10H9NO2	87-51-4	HMDB0000197	C00954	Indoles and derivatives
307	Indole-3-butyric acid	C12H13NO2	133-32-4	HMDB0002096	C11284	Indoles and derivatives
308	Indole-3-carboxylic acid	C9H7NO2	771-50-6	HMDB0003320	C19837	Indoles and derivatives
309	Indole-3-lactic acid	C11H11NO3	1821-52-9	HMDB0000671	C02043	Indoles and derivatives
310	Inosine	C10H12N4O5	58-63-9	HMDB0000195	C00294	Nucleosides, nucleotides, and analogues
311	Inosine diphosphate	C10H14N4O11P2	86-04-4	HMDB0003335	C00104	Nucleosides, nucleotides, and analogues
312	Inosinic acid	C10H13N4O8P	131-99-7	HMDB0000175	C00130	Nucleosides, nucleotides, and analogues
313	Inositol	C6H12O6	87-89-8	HMDB0000211	C00137	Others
314	Isoamylamine	C5H13N	107-85-7	HMDB0031659	C02640	Biogenic Amines
315	Isobutyrylglycine	C6H11NO3	15926-18-8	HMDB0000730		Amino acids, peptides, and analogues
316	Isocitric acid	C6H8O7	320-77-4	HMDB0000193	C00311	Organic acids and derivatives
317	Isoguanine	C5H5N5O	3373-53-3	HMDB0000403		Purines and purine derivatives
318	Isomaltose	C12H22O11	37169-69-0	HMDB0002923	C00252	Carbohydrates and carbohydrate conjugates
319	Isonicotinic acid	C6H5NO2	55-22-1	HMDB0060665	C07446	Pyridines and derivatives
320	Itaconic acid	C5H6O4	97-65-4	HMDB0002092	C00490	Organic acids and derivatives
321	Kynurenic acid	C10H7NO3	492-27-3	HMDB0000715	C01717	Azacyclic compounds
322	Kynurenine	C10H12N2O3	343-65-7	HMDB0000684	C00328	Amino acids, peptides, and analogues
323	L-(+)-rhamnose	C6H12O5	3615-41-6	HMDB0000849	C00507	Carbohydrates and carbohydrate conjugates
324	L-2-Amino adipic Acid	C6H11NO4	1118-90-7	HMDB0000510	C00956	Amino acids, peptides, and analogues
325	L-2-Aminobutyric acid	C4H9NO2	1492-24-6	HMDB0000452	C02356	Amino acids, peptides, and analogues
326	Lactic acid	C3H6O3	79-33-4	HMDB0000190	C00186	Organic acids and derivatives
327	L-Alanine	C3H7NO2	56-41-7	HMDB0000161	C00041	Amino acids, peptides, and analogues

328	L-alpha-Aspartyl-L-phenylalanine	C13H16N2O5	13433-09-5	HMDB0000706		Amino acids, peptides, and analogues
329	L-Arabinose	C5H10O5	87-72-9	HMDB0000646	C00259	Carbohydrates and carbohydrate conjugates
330	L-Arabitol	C5H12O5	7643-75-6	HMDB0001851	C00532	Carbohydrates and carbohydrate conjugates
331	L-Arginine	C6H14N4O2	74-79-3	HMDB0000517	C00062	Amino acids, peptides, and analogues
332	L-Argininosuccinic Acid	C10H18N4O6	2387-71-5	HMDB0000052	C03406	Amino acids, peptides, and analogues
333	L-Asparagine	C4H8N2O3	70-47-3	HMDB0000168	C00152	Amino acids, peptides, and analogues
334	L-Aspartic acid	C4H7NO4	56-84-8	HMDB0000191	C00049	Amino acids, peptides, and analogues
335	L-Carnitine	C7H15NO3	541-15-1	HMDB0000062	C00318	Biogenic Amines
336	L-Carnosine	C9H14N4O3	305-84-0	HMDB0000033	C00386	Amino acids, peptides, and analogues
337	L-Citrulline	C6H13N3O3	372-75-8	HMDB0000904	C00327	Amino acids, peptides, and analogues
338	L-Cystathionine	C7H14N2O4S	56-88-2	HMDB0000099	C02291	Amino acids, peptides, and analogues
339	L-Cysteinesulfinic acid	C3H7NO4S	1115-65-7	HMDB0000996	C00606	Amino acids, peptides, and analogues
340	L-Cystine	C6H12N2O4S2	56-89-3	HMDB0000192	C00491	Amino acids, peptides, and analogues
341	L-Dopa	C9H11NO4	59-92-7	HMDB0000181	C00355	Amino acids, peptides, and analogues
342	Levulinic-acid	C5H8O3	123-76-2	HMDB0000720		Organic acids and derivatives
343	L-Fucose-1-phosphate	C6H13O8P	16562-58-6	HMDB0001265	C02985	Carbohydrates and carbohydrate conjugates
344	L-Glutamic acid	C5H9NO4	56-86-0	HMDB0000148	C00025	Amino acids, peptides, and analogues
345	L-Glutamine	C5H10N2O3	56-85-9	HMDB0000641	C00064	Amino acids, peptides, and analogues
346	L-Glycine	C2H5NO2	56-40-6	HMDB0000123	C00037	Amino acids, peptides, and analogues
347	L-Histidine	C6H9N3O2	71-00-1	HMDB0000177	C00135	Amino acids, peptides, and analogues
348	L-Homoarginine	C7H16N4O2	156-86-5	HMDB0000670	C01924	Amino acids, peptides, and analogues
349	L-Homocysteic Acid	C4H9NO5S	14857-77-3	HMDB0002205	C16511	Amino acids, peptides, and analogues
350	L-Homocystine	C8H16N2O4S2	626-72-2	HMDB0000676	C01817	Amino acids, peptides, and analogues
351	L-Homoserine	C4H9NO3	672-15-1	HMDB0000719	C00263	Amino acids, peptides, and analogues
352	Lithocholic acid	C24H40O3	434-13-9	HMDB0000761	C03990	Steroids and steroid derivatives
353	L-Leucine	C6H13NO2	61-90-5	HMDB0000687	C00123	Amino acids, peptides, and analogues
354	L-Lysine	C6H14N2O2	56-87-1	HMDB0000182	C00047	Amino acids, peptides, and analogues
355	L-Malic acid	C4H6O5	97-67-6	HMDB0000156	C00149	Organic acids and derivatives
356	L-Methionine	C5H11NO2S	63-68-3	HMDB0000696	C00073	Amino acids, peptides, and analogues
357	L-Methionine sulfone	C5H11NO4S	7314-32-1	HMDB0062174		Amino acids, peptides, and analogues
358	L-Nicotine	C10H14N2	54-11-5	HMDB0001934	C00745	Pyridines and derivatives
359	L-Norleucine	C6H13NO2	327-57-1	HMDB0001645	C01933	Amino acids, peptides, and analogues
360	L-Ornithine	C5H12N2O2	70-26-8	HMDB0000214	C00077	Amino acids, peptides, and analogues

361	L-Phenylalanine	C9H11NO2	63-91-2	HMDB0000159	C00079	Amino acids, peptides, and analogues
362	L-Pipecolic acid	C6H11NO2	3105-95-1	HMDB0000716	C00408	Amino acids, peptides, and analogues
363	L-Proline	C5H9NO2	147-85-3	HMDB0000162	C00148	Amino acids, peptides, and analogues
364	L-Pyroglutamic acid	C5H7NO3	98-79-3	HMDB0000267	C01879	Amino acids, peptides, and analogues
365	L-Saccharopine	C11H20N2O6	997-68-2	HMDB0000279	C00449	Amino acids, peptides, and analogues
366	L-Serine	C3H7NO3	56-45-1	HMDB0000187	C00065	Amino acids, peptides, and analogues
367	L-Sorbose	C6H12O6	470-15-5	HMDB0001266	C00247	Carbohydrates and carbohydrate conjugates
368	L-Tartaric acid	C4H6O6	87-69-4	HMDB0000956	C00898	Carbohydrates and carbohydrate conjugates
369	L-Threonic acid	C4H8O5	7306-96-9	HMDB0000943	C01620	Carbohydrates and carbohydrate conjugates
370	L-Threonine	C4H9NO3	72-19-5	HMDB0000167	C00188	Amino acids, peptides, and analogues
371	L-Tryptophan	C11H12N2O2	73-22-3	HMDB0000929	C00078	Amino acids, peptides, and analogues
372	L-Tyrosine	C9H11NO3	60-18-4	HMDB0000158	C00082	Amino acids, peptides, and analogues
373	L-Valine	C5H11NO2	72-18-4	HMDB0000883	C00183	Amino acids, peptides, and analogues
374	Maleic acid	C4H4O4	110-16-7	HMDB0000176	C01384	Organic acids and derivatives
375	Malonaldehyde	C3H4O2	542-78-9	HMDB0006112	C19440	Organic acids and derivatives
376	Malonic acid	C3H4O4	141-82-2	HMDB0000691	C04025	Organic acids and derivatives
377	Maltol	C6H6O3	118-71-8	HMDB0030776	C11918	Others
378	Mannitol	C6H14O6	69-65-8	HMDB0000765	C00392	Carbohydrates and carbohydrate conjugates
379	Mannose 6-phosphate	C6H13O9P	3672-15-9	HMDB0304312	C00275	Carbohydrates and carbohydrate conjugates
380	Melatonin	C13H16N2O2	73-31-4	HMDB0001389	C01598	Biogenic Amines
381	Menatetrenone	C31H40O2	863-61-6	HMDB0030017	C00828	Lipids
382	Mercaptopurine	C5H4N4S	50-44-2	HMDB0015167	C02380	Purines and purine derivatives
383	Mesaconic acid	C5H6O4	498-24-8	HMDB0000749	C01732	Organic acids and derivatives
384	Metformin	C4H11N5	657-24-9	HMDB0001921	C07151	Biogenic Amines
385	Methionine sulfoxide	C5H11NO3S	3226-65-1	HMDB0002005	C02989	Amino acids, peptides, and analogues
386	Methionine sulfoximine	C5H12N2O3S	1982-67-8	HMDB0029430	C03510	Amino acids, peptides, and analogues
387	Methyl hippurate	C10H11NO3	1205-08-9	HMDB0000859		Amino acids, peptides, and analogues
388	Methylmalonic acid	C4H6O4	516-05-2	HMDB0000202	C02170	Organic acids and derivatives
389	Methylsuccinic acid	C5H8O4	498-21-5	HMDB0001844		Organic acids and derivatives
390	Mevalonic acid	C6H12O4	17817-88-8	HMDB0000227	C00418	Organic acids and derivatives
391	Mevalonolactone	C6H10O3	674-26-0	HMDB0006024		Others
392	Morpholine	C4H9NO	110-91-8	HMDB0031581	C14452	Azacyclic compounds
393	Myristic acid	C14H28O2	544-63-8	HMDB0000806	C06424	Lipids

394	N-(2-Furoyl)glycine	C7H7NO4	5657-19-2	HMDB0000439		Amino acids, peptides, and analogues
395	N,N-Dimethylarginine	C8H18N4O2	30315-93-6	HMDB0001539		Amino acids, peptides, and analogues
396	N,N-Dimethylglycine	C4H9NO2	1118-68-9	HMDB0000092	C01026	Amino acids, peptides, and analogues
397	N,N-Dimethylguanosine	C12H17N5O5	2140-67-2	HMDB0004824		Nucleosides, nucleotides, and analogues
398	N4-Acetylcytidine	C11H15N3O6	3768-18-1	HMDB0005923	C22293	Nucleosides, nucleotides, and analogues
399	N6,N6,N6-Trimethyl-L-lysine	C9H20N2O2	19253-88-4	HMDB0001325	C03793	Amino acids, peptides, and analogues
400	N6-Acetyl-L-lysine	C8H16N2O3	692-04-6	HMDB0000206	C02727	Amino acids, peptides, and analogues
401	N-Acetyl-5-hydroxytryptamine	C12H14N2O2	1210-83-9	HMDB0001238	C00978	Biogenic Amines
402	N-Acetylcadaverine	C7H16N2O	32343-73-0	HMDB0002284		Biogenic Amines
403	N-Acetylcarnosine	C11H16N4O4	56353-15-2	HMDB0012881		Amino acids, peptides, and analogues
404	N-Acetyl-D-Glucosamine	C8H15NO6	7512-17-6	HMDB0000215	C00140	Carbohydrates and carbohydrate conjugates
405	N-Acetyl-D-Glucosamine 6-Phosphate	C8H16NO9P	18191-20-3	HMDB0001062	C00357	Carbohydrates and carbohydrate conjugates
406	N-Acetylglucosaminylasparagine	C12H21N3O8	2776-93-4	HMDB0000489	C04540	Carbohydrates and carbohydrate conjugates
407	N-Acetylglutamine	C7H12N2O4	2490-97-3	HMDB0006029		Amino acids, peptides, and analogues
408	N-Acetyl glycine	C4H7NO3	543-24-8	HMDB0000532		Amino acids, peptides, and analogues
409	N-Acetyl-L-alanine	C5H9NO3	97-69-8	HMDB0000766		Amino acids, peptides, and analogues
410	N-Acetyl-L-aspartic acid	C6H9NO5	997-55-7	HMDB0000812	C01042	Amino acids, peptides, and analogues
411	N-Acetyl-L-glutamic acid	C7H11NO5	1188-37-0	HMDB0001138	C00624	Amino acids, peptides, and analogues
412	N-Acetyl-L-phenylalanine	C11H13NO3	2018-61-3	HMDB0000512	C03519	Amino acids, peptides, and analogues
413	N-Acetyl-L-tyrosine	C11H13NO4	537-55-3	HMDB0000866		Amino acids, peptides, and analogues
414	N-Acetyl-Neuraminic Acid	C11H19NO9	131-48-6	HMDB0000230	C19910	Carbohydrates and carbohydrate conjugates
415	N-Acetylornithine	C7H14N2O3	6205-08--9	HMDB0003357	C00437	Amino acids, peptides, and analogues
416	N-Acetylputrescine	C6H14N2O	5699-41-2	HMDB0002064	C02714	Biogenic Amines
417	N-Acetylserine	C5H9NO4	16354-58-8	HMDB0002931		Amino acids, peptides, and analogues
418	NAD+	C21H27N7O14P2	53-84-9	HMDB0000902	C00003	Nucleosides, nucleotides, and analogues
419	Naringin	C27H32O14	10236-47-2	HMDB0002927	C09789	Benzenoids
420	N-carbamoyl-L-aspartate	C5H8N2O5	13184-27-5	HMDB0000828	C00438	Amino acids, peptides, and analogues
421	N-Formylglycine	C3H5NO3	2491-15-8	HMDB0255145		Amino acids, peptides, and analogues
422	N-Formyl-L-methionine	C6H11NO3S	4289-98-9	HMDB0001015	C03145	Amino acids, peptides, and analogues
423	Nicotinamide	C6H6N2O	98-92-0	HMDB0001406	C00153	Pyridines and derivatives
424	Nicotinamide adenine dinucleotide phosphate	C21H28N7O17P3	53-59-8	HMDB0000217	C00006	Nucleosides, nucleotides, and analogues
425	Nicotinamide mononucleotide	C11H15N2O8P	1094-61-7	HMDB0000229	C00455	Nucleosides, nucleotides, and analogues
426	Nicotinamide N-oxide	C6H6N2O2	1986-81-8	HMDB0002730		Pyridines and derivatives

427	Nicotinic acid	C6H5NO2	59-67-6	HMDB0001488	C00253	Pyridines and derivatives
428	Nicotinuric acid	C8H8N2O3	583-08-4	HMDB0003269	C05380	Amino acids, peptides, and analogues
429	N-Isovaleroylglycine	C7H13NO3	16284-60-9	HMDB0000678		Amino acids, peptides, and analogues
430	N-Methylnicotinamide	C7H8N2O	114-33-0	HMDB0003152		Pyridines and derivatives
431	Normetanephrine	C9H13NO3	97-31-4	HMDB0000819	C05589	Biogenic Amines
432	Norvaline	C5H11NO2	6600-40-4	HMDB0251527	C01826	Amino acids, peptides, and analogues
433	N-Propionylglycine	C5H9NO3	21709-90-0	HMDB0000783		Amino acids, peptides, and analogues
434	Nutriacholic acid	C24H38O4	4651-67-6	HMDB0000467		Steroids and steroid derivatives
435	O-Acetylcarnitine	C9H18NO4+	3040-38-8	HMDB0000201	C02571	Lipids
436	O-Acetyl-L-serine	C5H9NO4	5147-00-2	HMDB0003011	C00979	Amino acids, peptides, and analogues
437	O-Benzyl-L-serine	C10H13NO3	4726-96-9			Amino acids, peptides, and analogues
438	Octopamine	C8H11NO2	104-14-3	HMDB0004825	C04227	Biogenic Amines
439	o-Methoxyphenyl sulfate	C7H8O5S	3233-59-8	HMDB0060013		Benzenoids
440	O-Phosphoethanolamine	C2H8NO4P	1071-23-4	HMDB0000224	C00346	Organic acids and derivatives
441	O-Phospho-L-Serine	C3H8NO6P	407-41-0	HMDB0000272	C01005	Amino acids, peptides, and analogues
442	Orotic acid	C5H4N2O4	65-86-1	HMDB0000226	C00295	Pyrimidines and pyrimidine derivatives
443	Orsellinic acid	C8H8O4	480-64-8		C01839	Benzenoids
444	O-Succinyl-L-homoserine	C8H13NO6	1492-23-5	HMDB0255868	C01118	Amino acids, peptides, and analogues
445	Oxypurinol	C5H4N4O2	2465-59-0	HMDB0000786	C07599	Purines and purine derivatives
446	Pantothenic acid	C9H17NO5	79-83-4	HMDB0000210	C00864	Organic acids and derivatives
447	Pantothenol	C9H19NO4	16485-10-2	HMDB0304820		Lipids
448	Paraxanthine	C7H8N4O2	611-59-6	HMDB0001860	C13747	Purines and purine derivatives
449	p-Cresol glucuronide	C13H16O7	17680-99-8	HMDB0011686		Carbohydrates and carbohydrate conjugates
450	Pelargonin	C27H31O15+	58751-31-8	HMDB0033681	C08725	Azacyclic compounds
451	Phenoxyacetic acid	C8H8O3	122-59-8	HMDB0031609	C02181	Benzenoids
452	Phenylacetylglutamine	C13H16N2O4	28047-15-6	HMDB0006344	C04148	Amino acids, peptides, and analogues
453	Phenylethylamine	C8H11N	64-04-0	HMDB0012275	C05332	Biogenic amines
454	Phenylpyruvic acid	C9H8O3	156-06-9	HMDB0000205	C00166	Benzenoids
455	Phloroglucinol	C6H6O3	108-73-6	HMDB0013675	C02183	Benzenoids
456	Phosphocreatine	C4H10N3O5P	67-07--2	HMDB0001511	C02305	Amino acids, peptides, and analogues
457	Phosphoenolpyruvic acid	C3H5O6P	138-08-9	HMDB0000263	C00074	Organic acids and derivatives
458	Phosphorylcholine Chloride	C5H15ClNO4P	3616-04-4	HMDB0001565	C00588	Biogenic Amines
459	Phthalic acid	C8H6O4	88-99-3	HMDB0002107	C01606	Benzenoids

460	p-Hydroxymandelic acid	C8H8O4	1198-84-1	HMDB0000822	C11527	Benzenoids
461	Picolinic acid	C6H5NO2	98-98-6	HMDB0002243	C10164	Pyridines and derivatives
462	Pimelic acid	C7H12O4	111-16-0	HMDB0000857	C02656	Organic acids and derivatives
463	Piperine	C17H19NO3	94-62-2	HMDB0029377	C03882	Benzenoids
464	Progesterone	C21H30O2	57-83-0	HMDB0001830	C00410	Steroids and steroid derivatives
465	Propionylcarnitine	C10H19NO4	20064-19-1	HMDB0000824	C03017	Lipids
466	Propionyl-CoA	C24H40N7O17P3S	317-66-8	HMDB0001275	C00100	Nucleosides, nucleotides, and analogues
467	Prostaglandin B2	C20H30O4	13367-85-6	HMDB0004236	C05954	Lipids
468	Prostaglandin D1	C20H34O5	17968-82-0	HMDB0005102	C06438	Lipids
469	Prostaglandin E2	C20H32O5	363-24-6	HMDB0001220	C00584	Lipids
470	Pseudouridine	C9H12N2O6	1445-07-4	HMDB0000767	C02067	Nucleosides, nucleotides, and analogues
471	Purine	C5H4N4	120-73-0	HMDB0001366	C15587	Purines and purine derivatives
472	Pyridoxal	C8H9NO3	66-72-8	HMDB0001545	C00250	Pyridines and derivatives
473	Pyridoxamine	C8H12N2O2	85-87-0	HMDB0001431	C00534	Pyridines and derivatives
474	Pyridoxine	C8H11NO3	65-23-6	HMDB0000239	C00314	Pyridines and derivatives
475	Pyrophosphoric acid	H4O7P2	2466-09-3	HMDB0000250	C00013	Others
476	Pyrrolidine	C4H9N	123-75-1	HMDB0031641		Azacyclic compounds
477	Pyruvic acid	C3H4O3	127-17-3	HMDB0000243	C00022	Organic acids and derivatives
478	Quinic acid	C7H12O6	77-95-2	HMDB0003072	C00296	Organic acids and derivatives
479	Quinoline-2-carboxylic acid	C10H7NO2	93-10-7	HMDB0000842	C06325	Azacyclic compounds
480	Quinolinic acid	C7H5NO4	89-00-9	HMDB0000232	C03722	Pyridines and derivatives
481	Raffinose	C18H32O16	512-69-6	HMDB0003213	C00492	Carbohydrates and carbohydrate conjugates
482	Resveratrol	C14H12O3	501-36-0	HMDB0003747	C03582	Benzenoids
483	Retinal	C20H28O	116-31-4	HMDB0001358	C00376	Lipids
484	Ribose	C5H10O5	10257-32-6	HMDB0000283	C00121	Carbohydrates and carbohydrate conjugates
485	Ribose-5-Phosphate	C5H11O8P	34980-65-9	HMDB0001548	C00117	Carbohydrates and carbohydrate conjugates
486	S-Adenosyl-L-homocysteine	C14H20N6O5S	979-92-0	HMDB0000939	C00021	Nucleosides, nucleotides, and analogues
487	S-Adosylmethionine	C15H22N6O5S	29908-03-0	HMDB0001185	C00019	Nucleosides, nucleotides, and analogues
488	Salicylic acid	C7H6O3	69-72-7	HMDB0001895	C00805	Benzenoids
489	Salicyluric acid	C9H9NO4	487-54-7	HMDB0000840	C07588	Amino acids, peptides, and analogues
490	Sarcosine	C3H7NO2	107-97-1	HMDB0000271	C00213	Amino acids, peptides, and analogues
491	Scopoletin	C10H8O4	92-61-5	HMDB0034344	C01752	Benzenoids
492	Sebacic acid	C10H18O4	111-20-6	HMDB0000792	C08277	Organic acids and derivatives

493	Selenomethionine	C5H11NO2Se	3211-76-5	HMDB0003966	C05335	Amino acids, peptides, and analogues
494	Shikimic acid	C7H10O5	138-59-0	HMDB0003070	C00493	Organic acids and derivatives
495	S-Methyl-L-cysteine	C4H9NO2S	1187-84-4	HMDB0002108	C22040	Amino acids, peptides, and analogues
496	Sorbitol	C6H14O6	50-70-4	HMDB0000247	C00794	Carbohydrates and carbohydrate conjugates
497	Spaglumic acid	C11H16N2O8	3106-85-2	HMDB0001067	C12270	Amino acids, peptides, and analogues
498	Suberic acid	C8H14O4	505-48-6	HMDB0000893	C08278	Organic acids and derivatives
499	Succinic acid	C4H6O4	110-15-6	HMDB0000254	C00042	Organic acids and derivatives
500	Sucrose	C12H22O11	57-50-1	HMDB0000258	C00089	Carbohydrates and carbohydrate conjugates
501	Taurine	C2H7NO3S	107-35-7	HMDB0000251	C00245	Organic acids and derivatives
502	Taurochenodeoxycholate	C26H45NO6S	516-35-8	HMDB0000951	C05465	Steroids and steroid derivatives
503	Taurocholic acid	C26H45NO7S	81-24-3	HMDB0000036	C05122	Steroids and steroid derivatives
504	Taurodeoxycholic acid	C26H45NO6S	516-50-7	HMDB0000896	C05463	Steroids and steroid derivatives
505	Tauroursodeoxycholic acid	C26H45NO6S	14605-22-2	HMDB0000874	C16868	Steroids and steroid derivatives
506	Terephthalic acid	C8H6O4	100-21-0	HMDB0002428	C06337	Benzenoids
507	Testosterone	C19H28O2	58-22-0	HMDB0000234	C00535	Steroids and steroid derivatives
508	Theophylline	C7H8N4O2	58-55-9	HMDB0001889	C07130	Purines and purine derivatives
509	Thiamine	C12H17N4OS+	70-16-6	HMDB0000235	C00378	Pyrimidines and pyrimidine derivatives
510	Thiamine monophosphate	C12H17N4O4PS	495-23-8	HMDB0002666	C01081	Pyrimidines and pyrimidine derivatives
511	Thiamine pyrophosphate	C12H19N4O7P2S+	154-87-0	HMDB0001372	C00068	Pyrimidines and pyrimidine derivatives
512	Thioctic acid	C8H14O2S2	1200-22-2	HMDB0001451	C16241	Lipids
513	Threonylphenylalanine	C13H18N2O4	16875-27-7	HMDB0029068		Amino acids, peptides, and analogues
514	Thymidine	C10H14N2O5	50-89-5	HMDB0000273	C00214	Nucleosides, nucleotides, and analogues
515	Thymidine-5'-diphosphate	C10H16N2O11P2	491-97-4	HMDB0001274	C00363	Nucleosides, nucleotides, and analogues
516	Thymidine-5'-phosphate	C10H15N2O8P	365-07-1	HMDB0001227	C00364	Nucleosides, nucleotides, and analogues
517	Thymine	C5H6N2O2	65-71-4	HMDB0000262	C00178	Pyrimidines and pyrimidine derivatives
518	Tiglic acid	C5H8O2	80-59-1	HMDB0001470	C08279	Organic acids and derivatives
519	Tiglylglycine	C7H11NO3	35842-45-6	HMDB0000959		Amino acids, peptides, and analogues
520	trans-Cinnamic acid	C9H8O2	140-10-3	HMDB0000930	C00423	Benzenoids
521	Trehalose	C12H22O11	99-20-7	HMDB0000975	C01083	Carbohydrates and carbohydrate conjugates
522	Trehalose-6-phosphate	C12H23O14P	4484-88-2	HMDB0001124	C00689	Carbohydrates and carbohydrate conjugates
523	Tretinoin	C20H28O2	302-79-4	HMDB0002369	C15493	Lipids
524	Trigonelline	C7H7NO2	535-83-1	HMDB0000875	C01004	Pyridines and derivatives
525	Trimethoprim	C14H18N4O3	738-70-5	HMDB0014583	C01965	Benzenoids

526	Trimethylamine	C3H9N	75-50-3	HMDB0000906	C00565	Biogenic Amines
527	Trimethylamine N-oxide	C3H9NO	1184-78-7	HMDB0000925	C01104	Biogenic Amines
528	Tryptamine	C10H12N2	61-54-1	HMDB0000303	C00398	Biogenic Amines
529	Tryptophol	C10H11NO	526-55-6	HMDB0003447	C00955	Indoles and derivatives
530	Tyramine	C8H11NO	51-67-2	HMDB0000306	C00483	Biogenic Amines
531	UDP-D-glucose	C15H24N2O17P2	133-89-1	HMDB0000286	C00029	Nucleosides, nucleotides, and analogues
532	Uracil	C4H4N2O2	66-22-8	HMDB0000300	C00106	Pyrimidines and pyrimidine derivatives
533	Urea	CH4N2O	57-13-6	HMDB0000294	C00086	Organic acids and derivatives
534	Uric acid	C5H4N4O3	69-93-2	HMDB0000289	C00366	Purines and purine derivatives
535	Uridine	C9H12N2O6	58-96-8	HMDB0000296	C00299	Nucleosides, nucleotides, and analogues
536	Uridine diphosphate glucuronic acid	C15H22N2O18P2	2616-64-0	HMDB0000935	C00167	Nucleosides, nucleotides, and analogues
537	Uridine-5'-diphosphate	C9H14N2O12P2	58-98-0	HMDB0000295	C00015	Nucleosides, nucleotides, and analogues
538	Urocanic acid	C6H6N2O2	104-98-3	HMDB0000301	C00785	Azacyclic compounds
539	Vanillylmandelic acid	C9H10O5	55-10-7	HMDB0000291	C05584	Benzenoids
540	Xanthosine	C10H12N4O6	146-80-5	HMDB0000299	C01762	Nucleosides, nucleotides, and analogues
541	Xanthurenic acid	C10H7NO4	59-00-7	HMDB0000881	C02470	Azacyclic compounds
542	Xanthylic acid (XMP)	C10H13N4O9P	523-98-8	HMDB0001554	C00655	Nucleosides, nucleotides, and analogues
543	Xylitol	C5H12O5	488-81-3	HMDB0002917	C00379	Carbohydrates and carbohydrate conjugates
544	Xylulose 5-phosphate	C5H11O8P	4212-65-1	HMDB0000868	C00231	Carbohydrates and carbohydrate conjugates
545	Acetic acid	CH3COOH	64-19-7	HMDB0000042	C00033	Short-chain fatty acids
546	Butyric acid	C4H8O2	107-92-6	HMDB0000039	C00246	Short-chain fatty acids
547	Decanoic acid	C10H20O2	334-48-5	HMDB0000511	C01571	Medium-chain fatty acids
548	Heptanoic acid	C7H14O2	111-14-8	HMDB0000666	C17714	Medium-chain fatty acids
549	Hexanoic acid	C6H12O2	142-62-1	HMDB0000535	C01585	Short-chain fatty acids
550	Isobutyric acid	C4H8O2	79-31-2	HMDB0001873	C02632	Short-chain fatty acids
551	Isovaleric acid	C5H10O2	503-74-2	HMDB0000718	C08262	Short-chain fatty acids
552	Nonanoic acid	C9H18O2	112-05-0	HMDB0000847	C01601	Medium-chain fatty acids
553	Octanoic acid	C8H16O2	124-07-2	HMDB0000482	C06423	Medium-chain fatty acids
554	Propionic acid	CH3CH2COOH	79-09-4	HMDB0000237	C00163	Short-chain fatty acids
555	Valeric acid	C5H10O2	109-52-4	HMDB0000892	C00803	Short-chain fatty acids

REVIEWERS' COMMENTS

Reviewer #1 (Remarks to the Author):

The authors answered all of my questions perfectly. I recommend the publication of the manuscript.

Reviewer #2 (Remarks to the Author):

The authors have done a nice job in responding to the reviewers' comments and the manuscript can be published without further modification in my opinion.

Reviewer #4 (Remarks to the Author):

The authors have addressed all my concerns and comments. I appreciate their efforts. I believe it can be accepted for publication.

De-Chen Lin, PhD
University of Southern California

RESPONSE TO REVIEWERS' COMMENTS

Reviewer #1 (Remarks to the Author):

The authors answered all of my questions perfectly. I recommend the publication of the manuscript.

Response: Thank you very much for your strong support. Wishing you a Happy New Year!

Reviewer #2 (Remarks to the Author):

The authors have done a nice job in responding to the reviewers' comments and the manuscript can be published without further modification in my opinion.

Response: Thank you very much for your valuable feedback. Your rigorous academic approach has indeed elevated our article. Wishing you a prosperous New Year!

Reviewer #4 (Remarks to the Author):

The authors have addressed all my concerns and comments. I appreciate their efforts. I believe it can be accepted for publication.

De-Chen Lin, PhD

University of Southern California

Response: Thank you very much for your insightful suggestions, and we hope to have more opportunities to learn from you in the future. Happy Chinese New Year of the Loong!